# Positive Matrix Factorization of Large Real-Time Atmospheric Mass Spectrometry Datasets Using Error-Weighted Randomized Hierarchical Alternating Least Squares

Benjamin Sapper[1,†], Sean Youn[1], Daven Henze[1], Manjula Canagaratna[2], Harald Stark[3,4], and Jose L. Jimenez[4]

[1]University of Colorado Boulder, 11 Engineering Dr, Boulder, CO 80309, United States
[2]Aerodyne Research, 45 Manning Road, Billerica, MA 01821, United States
[3]Center for Aerosol and Cloud Chemistry, Aerodyne Research Inc., 45 Manning Road, Billerica, MA 01821, United States
[4]Department of Chemistry and Cooperative Institute for Research in Environmental Sciences (CIRES), University of Colorado, Boulder, Colorado 80309, United States
[†]deceased

**Correspondence:** Sean Youn (sean.youn@colorado.edu)

**Abstract.** Weighted positive matrix factorization (PMF) has been used by scientists to find small sets of underlying factors in environmental data. However, as the size of the data has grown, increasing computational costs have made it impractical to use traditional methods for this factorization. In this paper, we present a new external weighting method to dramatically decrease computational costs for these traditional algorithms. The external weighting scheme, along with the randomized hierarchical alternating least squares (RHALS) algorithm, was applied to the Southern Oxidant and Aerosol Study (SOAS 2013) dataset of gaseous highly oxidized multifunctional molecules (HOMs). The modified RHALS algorithm successfully reproduced six previously identified, interpretable factors with the total computation time of the nonoptimized code showing potential improvements on the order of one to two orders of magnitude compared to competing algorithms. We also investigate rotational ambiguity in the solution and present a simple "pulling" method to rotate a set of factors. This method is shown to find alternative solutions, and in some cases, lower the weighted residual error of the algorithm.

## 1 Introduction

### 1.1 Problem Statement

Low rank matrix factorization has been widely used in data science to explain underlying factors in large datasets (Xie et al. (1998); Kim and Hopke (2007); Wei et al. (2016)). The process considers a data matrix, $\mathbf{A}$, of size $m \times n$, that is decomposed into two smaller matrices, $\mathbf{W}$ of size $m \times k$ and $\mathbf{H}$ of size $k \times n$, where $k \ll \min(m,n)$ and $\mathbf{A} \approx \mathbf{WH}$. Traditionally, principal component analysis (PCA) and singular value decomposition (SVD) have been used to find these factors (Kumar (2017); Wei et al. (2016)). PCA finds the eigenvectors of the covariance matrix $\mathbf{A}^T\mathbf{A}$, which are called the principal components, representing the directions of maximum variance in the data. The vectors are ordered by how much variance they explain, and only the most important vectors are kept, which are identified as the underlying factors in the data. Closely related, the SVD

finds the factorization $\mathbf{A} = \mathbf{U}\mathbf{S}\mathbf{V}^T$, where $\mathbf{U}$ and $\mathbf{V}$ contain the left and right singular vectors, respectively, and $\mathbf{S}$ is a diagonal matrix containing the singular values of $\mathbf{A}$ in decreasing order. Thus, to find a low rank approximation of $\mathbf{A}$, one could keep only the $k$ most significant singular values and vectors to form the truncated SVD, $\mathbf{U}_k\mathbf{S}_k\mathbf{V}_k^T$. Mathematically, the truncated SVD is the most optimal rank $k$ factorization of $\mathbf{A}$ for minimizing squared error (Eckart and Young (1936)). However, the SVD is not appropriate for all factorization problems for several reasons (Paatero and Tapper (1994)):

1. The SVD produces factors with negative values: for some factor analysis problems, such as finding chemical sources for air pollution data, SVD results can be difficult to interpret as chemical concentrations can only be nonnegative.

   2. The SVD produces orthogonal factors: many factor analysis problems are not constrained by requirements of orthogonality between factors.

   3. The SVD is not fit to solve the following weighted least squares problem: suppose that accompanying the dataset $\mathbf{A}$ is

30 an equally sized ($m \times n$) matrix $\mathbf{\Sigma}$ with $\mathbf{\Sigma}_{ij} = \sigma_{ij}$ representing the uncertainty of the measurement for $\mathbf{A}_{ij}$. If $\mathrm{rank}(\mathbf{\Sigma}) > 1$, the SVD can't be scaled to find a solution minimizing the weighted residual error, which is defined as $\sum_{i=1}^{m}\sum_{j=1}^{n}\frac{(\mathbf{A}_{ij} - \sum_{l=1}^{k}\mathbf{W}_{il}\mathbf{H}_{lj})^2}{\sigma_{ij}^2}$ (Paatero and Tapper (1994)).

   Positive matrix factorization (PMF) was introduced by Paatero and Tapper (1993) to address these concerns. Weighted PMF attempts to find two factor matrices $\mathbf{W}$ and $\mathbf{H}$ by minimizing the equation

$$||(\mathbf{A} - \mathbf{W}\mathbf{H}) \oslash \mathbf{\Sigma}||_F^2 \text{ with } \mathbf{W} \geq 0, \mathbf{H} \geq 0 \tag{1}$$

In Eq. (1), $\oslash$ represents elementwise division, the norm $||\cdot||_F$ is the Frobenius norm, and all elements of $\mathbf{W}$ and $\mathbf{H}$ are constrained to be nonnegative. Further, we note that for consistency with nomenclature in the literature related to use of this algorithm for factorization of real-time atmospheric mass spectrometry datasets, we refer to this approach as "positive" matrix factorization (i.e., PMF) while recognizing that a more precise name would be nonnegative matrix factorization (NMF).

Traditional factor analysis methods are known to be computationally expensive. Steps to speed up factor analysis have been explored, such as randomization and the use of graphical processing units (GPUs) (Halko et al. (2011); Tan et al. (2018)). Developing efficient algorithms is especially critical in atmospheric mass spectrometry, as improvements in instrumentation and increases in the duration of their use in field campaigns has led to intractably large datasets. Currently, analysis of these datasets requires sacrificing data resolution or extensive manual preprocessing to operate within existing PMF software tools,

and full analysis can routinely take days or weeks of computation time (Hopke et al. (2023)). As a result, a variety of approaches have emerged for efficient source apportionment of atmospheric mass spectrometry data. Algorithms to solve the nonconvex optimization posed by PMF range from gradient descent, block coordinate descent, and projected gradient methods (Guo et al. (2024)). Attempts at using supervised, ensemble machine learning approaches have been shown to be capable of replicating results from traditional (unsupervised) factorization methods while reducing computation time (Zhang et al. (2025)). Recently,

Erichson et al. (2018) applied randomization to PMF and introduced a new method, randomized hierarchical alternating least squares (RHALS), to solve the unweighted PMF problem. In this paper, we test the application of RHALS to atmospheric concentration data that contain uncertainties. Accounting for these uncertainties as regression weights, we introduce a method of externally weighting and unweighting the data, which to our knowledge is novel in its application to RHALS. We consider the

accuracy and the reduced computational costs compared to other PMF algorithms commonly used in the field of atmospheric science.

## 1.2 Background

### 1.2.1 PMF2 and Paatero

The first widely accepted algorithm for PMF was derived in Paatero (1997) using the Gauss-Newton Method. This algorithm, called PMF2, is commonly used with environmental data (Kim and Hopke (2007); Ulbrich et al. (2009); Massoli et al. (2018)). Paatero (1997) defines an enhanced objective function, and attempts to find factor matrices $\mathbf{W}$ and $\mathbf{H}$ that minimize the cost function $Q$ in

$$Q = \sum_{i=1}^{m}\sum_{j=1}^{n}\frac{(\mathbf{A}_{ij}-\sum_{l=1}^{k}\mathbf{W}_{il}\mathbf{H}_{lj})^2}{\sigma_{ij}^2} - \alpha\sum_{i=1}^{m}\sum_{l=1}^{k}\log(\mathbf{W}_{il}) - \beta\sum_{l=1}^{k}\sum_{j=1}^{n}\log(\mathbf{H}_{lj}) + \gamma\sum_{i=1}^{m}\sum_{l=1}^{k}\mathbf{W}_{il}^2 + \delta\sum_{l=1}^{k}\sum_{j=1}^{n}\mathbf{H}_{lj}^2 \qquad (2)$$

In Eq. (2), logarithmic penalty terms are added to penalize factor values that become too close to zero (and therefore potentially negative), and L2 regularization is added to smooth out the factors and avoid overfitting (Paatero (1997)). $\alpha$ and $\beta$ control the strength of the penalty terms, while $\gamma$ and $\delta$ control the strength of L2 regularization. To date, Paatero's exact algorithmic approach to solving Eq. (2) remains unpublished. However, pseudocode for using the Gauss-Newton method to solve Eq. (2) is detailed in Lu and Wu (2004).

### 1.2.2 Multiplicative Update

An alternative method for PMF was developed in Lee and Seung (1999). The multiplicative update (MU) method utilizes a special case of gradient descent where the learning rates are chosen to avoid subtraction in the gradient (Gillis (2020)). A multiplicative update estimate of a parameter $\boldsymbol{\theta}$ (either $\mathbf{W}$ or $\mathbf{H}$) is found by updates of the form (Gillis (2020))

$$\boldsymbol{\theta} = \boldsymbol{\theta} \odot (\nabla_{\boldsymbol{\theta}}^{-} Q(\boldsymbol{\theta}) \oslash \nabla_{\boldsymbol{\theta}}^{+} Q(\boldsymbol{\theta})) \qquad (3)$$

where $Q(\boldsymbol{\theta})$ is a cost function to be minimized, $\nabla_{\boldsymbol{\theta}}^{-}$ consists of the negative terms of the gradient of the cost function, $\nabla_{\boldsymbol{\theta}}^{+}$ consists of the positive terms of the gradient, and $\odot$ denotes elementwise multiplication. $\boldsymbol{\theta}$ is initialized with all positive entries as the MU cannot update an entry $\boldsymbol{\theta}_{ij}$ if it is equal to zero (Gillis (2020)). As the data matrix $\mathbf{A}$, the uncertainties ($\sigma_{ij}$), and the factor matrices are all nonnegative at each step of this algorithm, the factor matrices in the subsequent step are guaranteed to be positive because Eq. (3) only deals with the multiplication and division of positive numbers.

It is possible to perform PMF using other forms of gradient descent - for example, the projected gradient method (PGM) sets the step size to the inverse of the maximum eigenvalue of the Hessian of the cost function, and may lead to faster convergence than MU (Gillis (2020)). However, we choose to only test MU due to its widespread use and flexibility (Gillis (2020)).

### 1.2.3 Alternating Least Squares and Hierarchical ALS

Alternating least squares (ALS) methods solve for the factor matrices $\mathbf{W}$ and $\mathbf{H}$ by iteratively updating each matrix until convergence is reached (Cichocki et al. (2009)). The cost function $Q$, containing $||(\mathbf{A} - \mathbf{WH}) \oslash \mathbf{\Sigma}||_F^2$, is minimized by setting the partial derivatives $\frac{\partial Q}{\partial \mathbf{W}}$ and $\frac{\partial Q}{\partial \mathbf{H}}$ to zero and solving for $\mathbf{W}$ and $\mathbf{H}$. To satisfy the positivity constraint, negative elements in the factors are set to zero.

Nonnegative ALS has no theoretical convergence guarantee and, in some problems, may fail to converge to a feasible solution (Gillis (2020)). For this reason, alternating nonnegative least squares (ANLS) and alternating direction method of multipliers (ADMM) are interesting alternatives. In ANLS, indices of an "active set" are set to zero, and the rest are updated via an unconstrained optimization (Kim and Park (2011)). The active set is then updated to contain the indices with the new negative factor elements. In ADMM, an auxiliary factor matrix $\mathbf{Y}$ is formed, and an additional term is added to the cost function which penalizes the distance between the target factor matrix ($\mathbf{W}$ or $\mathbf{H}$) and $\mathbf{Y}$ (Gillis (2020)). Both of these methods may lead to faster and better convergence than nonnegative ALS (Gillis (2020)). However, we find that the simple nonnegative ALS almost always converges to feasible solutions for our dataset, and we do not explore these alternative methods.

In recent years, hierarchical alternating least squares (HALS) has become increasingly popular as an efficient method for PMF (Cichocki and Phan (2009)). Instead of minimizing with respect to the entire factor matrices $\mathbf{W}$ and $\mathbf{H}$, HALS minimizes the cost function with respect to one block, or an outer product, of individual factors at a time. The main component of the cost function is redefined as $Q_j = ||(\mathbf{R}_j - \mathbf{W}_{(:,j)}\mathbf{H}_{(j,:)}) \oslash \mathbf{\Sigma}||_F^2$, where $\mathbf{W}_{(:,j)}$ and $\mathbf{H}_{(j,:)}$ are the $j^{\text{th}}$ factors. $Q_j$ can be minimized for each factor $j$ by setting the partial derivatives $\frac{\partial Q_j}{\partial \mathbf{W}_{(:,j)}}$ and $\frac{\partial Q_j}{\partial \mathbf{H}_{(j,:)}}$ to zero and solving for $\mathbf{W}_{(:,j)}$, and $\mathbf{H}_{(j,:)}$.

The derivation of the ALS update rules is detailed in section A of the Appendix, while the derivation of HALS is detailed in Section 2.

### 1.3 Random Projections

To reduce the computational costs of a matrix factorization algorithm for large datasets, randomization methods have been used as a dimension reduction technique (Erichson et al. (2018); Halko et al. (2011); Kaloorazi and Chen (2019)). Below, we present a brief overview of the theory and results laid out in Halko et al. (2011).

When performing randomization techniques, we hope that much of the relevant information about the column space of the data matrix $\mathbf{A}$ can be stored in a much smaller subset of vectors that we can sample. This is only true if the "effective rank" of $\mathbf{A}$ is low ($\mathbf{A}$ only has a few nonnegligible singular values), but that is generally assumed to be the case in any PMF problem (Erichson et al. (2018)). Mathematically, we seek the approximation

$$\mathbf{A} \approx \mathbf{PP}^T\mathbf{A} \tag{4}$$

where the relatively few number of columns in the matrix $\mathbf{P}$ are orthonormal and form an approximate basis of $\mathbf{A}$. Choosing the columns of $\mathbf{P}$ to be the left singular vectors of A would minimize L2 error: if the first $k$ singular vectors were chosen, the error term $||\mathbf{A} - \mathbf{PP}^T\mathbf{A}||_2 = \sigma_{k+1}$, with $\sigma_{k+1}$ being the $(k+1)^{\text{st}}$ largest singular value of $\mathbf{A}$ (Halko et al. (2011)). However, random sampling from the column space of $\mathbf{A}$ can also produce a suitable basis.

The assumption that there are $k$ underlying factors within $\mathbf{A}$ implies that the effective rank of $\mathbf{A}$ is $k$. It then appears reasonable to use $k$ random samples from the column space of $\mathbf{A}$ to form a basis. However, with underlying uncertainties, we can write $\mathbf{A} = \mathbf{B} + \mathbf{E}$, where $\mathbf{B}$ is the rank $k$ matrix spanned by the factors for which we wish to find a basis, and $\mathbf{E}$ is a perturbation matrix filled with the noise in $\mathbf{A}$ (Halko et al. (2011)). Suppose we were to sample from the column space of A – that is, form the vector $\mathbf{y} = \mathbf{A}\boldsymbol{\omega} = \mathbf{B}\boldsymbol{\omega} + \mathbf{E}\boldsymbol{\omega}$. Each vector $\mathbf{y}$ is slightly pushed out of the column space of $\mathbf{B}$ by the term $\mathbf{E}\boldsymbol{\omega}$. Thus, to increase the likelihood of spanning the column space of $\mathbf{B}$, an additional $p$ vectors are sampled from $\mathbf{A}$. In practice, choosing $p$ to be 10 or 20 is sufficient (Erichson et al. (2018)).

To construct this low rank approximation, $k + p$ random normal samples, stored as columns of the matrix $\boldsymbol{\Omega}$ (dimensions $n \times (k + p)$) of the column space of $\mathbf{A}$ are taken and stored in $\mathbf{Y}$:

$$\underset{m \times (k+p)}{\mathbf{Y}} = \underset{m \times n}{\mathbf{A}} \cdot \underset{n \times (k+p)}{\boldsymbol{\Omega}} \tag{5}$$

Next, the columns of $\mathbf{Y}$ are orthonormalized using a QR decomposition to form our projection matrix $\mathbf{P}$. The algorithm can now be run on the lower dimensional matrix $\mathbf{B} = \mathbf{P}^T \mathbf{A}$.

## 1.4 Nonuniqueness of Solutions

Unlike the SVD, there is no guarantee of uniqueness for the factor matrices $\mathbf{W}$ and $\mathbf{H}$ in PMF. That is, the factorization $\mathbf{A} = \mathbf{W}\mathbf{H}$ can also be expressed as $\mathbf{A} = \mathbf{W}\mathbf{T}\mathbf{T}^{-1}\mathbf{H}$, where $\mathbf{T}$ is a "rotational" matrix and $\hat{\mathbf{W}} = \mathbf{W}\mathbf{T}$ and $\hat{\mathbf{H}} = \mathbf{T}^{-1}\mathbf{H}$ are the new rotated factors. We note that $\mathbf{T}$ does not necessarily represent a true rotation in a mathematical form, which would require $\mathbf{T}$ to be orthogonal. To span the space of feasible solutions, previous approaches such as PMF2 have aimed to find new solutions $\hat{\mathbf{W}}$ and $\hat{\mathbf{H}}$ by varying $\mathbf{T}$ (Paatero (1997)), or varying initializations (Ulbrich et al. (2009)).

The rotational matrix $\mathbf{T}$ is a $k \times k$ matrix where $t_{ii}$, a diagonal element of $\mathbf{T}$, represents a scaling of the $i^{\text{th}}$ factor, and $t_{ij}$ represents a rotation of the $j^{\text{th}}$ factor towards the $i^{\text{th}}$ factor in $\mathbf{W}$, and a rotation of the $i^{\text{th}}$ factor away from the $j^{\text{th}}$ factor in $\mathbf{H}$. For example, consider the elementary rotation matrices (Paatero et al. (2002))

$$\mathbf{T}_E = \begin{pmatrix} 1 & 0 & 0 & \cdots & 0 \\ 0 & 1 & r & \cdots & 0 \\ 0 & 0 & 1 & \cdots & 0 \\ \vdots & \vdots & \vdots & \ddots & \vdots \\ 0 & 0 & 0 & \cdots & 1 \end{pmatrix}, \mathbf{T}_E^{-1} = \begin{pmatrix} 1 & 0 & 0 & \cdots & 0 \\ 0 & 1 & -r & \cdots & 0 \\ 0 & 0 & 1 & \cdots & 0 \\ \vdots & \vdots & \vdots & \ddots & \vdots \\ 0 & 0 & 0 & \cdots & 1 \end{pmatrix} \tag{6}$$

All factors remain the same, except $\hat{\mathbf{W}}_{(:,3)} = \mathbf{W}_{(:,3)} + r\mathbf{W}_{(:,2)}$, and $\hat{\mathbf{H}}_{(2,:)} = \mathbf{H}_{(2,:)} - r\mathbf{H}_{(3,:)}$.

Regardless of whether $r$ is positive or negative, values in either $\mathbf{W}$ or $\mathbf{H}$ will be pulled towards negative values. Thus if a large proportion of the factors are filled with zeros, there may be little to no pure rotations, or rotations that do not change the residual $||\mathbf{A} - \mathbf{W}\mathbf{H}||_F^2$. Algorithms such as those developed by Paatero, where a logarithmic penalty term is added to push factor values to be more positive, will have few zeros in the factor matrices, and thus will have more rotational ambiguity (Paatero et al. (2005)). However, RHALS enforces nonnegativity merely by setting negative elements to zero, and thus many values in

the factor matrices may end up being zero. Thus for RHALS, a perfect rotational matrix $\mathbf{T}$ will almost certainly not exist, and only "approximate" rotations can be studied, in which the rotation will alter the value of the weighted error.

It is not feasible to span all possible variants that $\mathbf{T}$ can take. Thus, the problem is often simplified to considering only positive rotations (values of $\mathbf{T}$ greater than zero) and negative rotations (values of $\mathbf{T}$ less than zero). A rotational program in PMF2 called FPEAK uses the parameter $\phi$ to denote the rotation strength, with positive values leading to positive rotations in $\mathbf{W}$ (Paatero (1997)). Paatero further improved this method in the Multilinear Engine (ME) algorithm, where the strength of rotation is allowed to vary between factors (Paatero and Hopke (2009)). The pulling algorithm presented in Paatero and Hopke (2009) is a sophisticated rotational method; more rudimentary pulling methods that mimic varying the regularization of the factor matrices are presented in Paatero (1997) and Paatero et al. (2002). Recent attempts at controlling for rotational ambiguity have involved additional factorization of the time-series matrix $\mathbf{W}$ into a matrix incorporating shape regularization to reflect known diurnal patterns of factors and a diagonal scaling matrix (Nanra et al. (2024)).

## 1.5 Scaling with Uncertainties

Recall that to account for inaccuracies in real data, we measure the squared error $Q$ of the algorithm by dividing the residual by the standard deviation of the uncertainty of each measurement:

$$Q = \sum_{i=1}^{m} \sum_{j=1}^{n} \left( \frac{\mathbf{A}_{ij} - \sum_{l=1}^{k} \mathbf{W}_{il} \mathbf{H}_{lj}}{\sigma_{ij}} \right)^2 \tag{7}$$

To account for these uncertainties, one could incorporate them into each update rule of the factor matrices, as is done in PMF2 (Paatero (1997)). In this paper, we refer to this as "internal weighting". However, this is computationally expensive due to repeated elementwise operations with the uncertainty matrix $\mathbf{\Sigma}$. Elementwise operations of large arrays are inefficient processes compared to other operations of the same computational complexity, such as matrix-vector multiplication, due to the large allocation of memory towards intermediary results (Jia et al. (2020)). We thus introduce an alternate approach to weighted PMF where the data is prescaled by the uncertainties matrix, the unweighted algorithm is applied to the scaled data, and the converged factors are scaled by the uncertainties. This approach, which we will refer to as "external weighting," dramatically reduces computational costs and allows for dimensionality reduction as weights are not included in the update rules.

It is noted in Paatero and Tapper (1993) that if the weighting matrix is rank one, that is if $\frac{1}{\sigma_{ij}} = \mathbf{B}_{(i,:)} \mathbf{C}_{(:,j)}$ for vectors of $\mathbf{B}$ and $\mathbf{C}$, then an optimal scaling can be found. By forming the diagonal matrices $\mathbf{D}_L$ (with the elements of $\mathbf{B}_{(i,:)}$) and $\mathbf{D}_R$ (with the elements of $\mathbf{C}_{(:,j)}$), if $\mathbf{A} = \mathbf{WH} + \mathbf{E}$, then $\mathbf{D}_L \mathbf{A} \mathbf{D}_R = \mathbf{D}_L \mathbf{WH} \mathbf{D}_R + \mathbf{D}_L \mathbf{E} \mathbf{D}_R = \hat{\mathbf{W}} \hat{\mathbf{H}} + \hat{\mathbf{E}}$. Thus, by first finding $\hat{\mathbf{A}} = \mathbf{D}_L \mathbf{A} \mathbf{D}_R$ and then running a PMF algorithm without weights on $\hat{\mathbf{A}}$, one can produce estimated factor matrices $\hat{\mathbf{W}}$ and $\hat{\mathbf{H}}$, where the unscaled estimates are $\mathbf{W} = \mathbf{D}_L^{-1} \hat{\mathbf{W}}$ and $\mathbf{H} = \hat{\mathbf{H}} \mathbf{D}_R^{-1}$.

If the weighting matrix $\mathbf{\Sigma}$ isn't rank one – which is likely for environmental data – and $\frac{1}{\sigma_{i,j}}$ cannot be estimated as the outer product of two vectors, there is no scaling of the previous form that can be applied to the data matrix $\mathbf{A}$ (Paatero and Tapper (1993)). To address this, Paatero and Tapper (1993) presented a simple algorithm to find an approximate rank 1 factorization

of $\Sigma$. We present a different method where the data is scaled by the full rank matrix $\Sigma$, and then unscaled after the algorithm is complete. This algorithm is described in detail in Section 2.3.

### 1.5.1 Expectation Maximization

The expectation maximization (EM) approach was first designed for matrix factorization problems associated with missing entries (Zhang et al. (2006)). Specifically, if $A^o$ is the observed data and $A^u$ is the unknown data within $\mathbf{A}$, then the EM approach seeks to find factors $\mathbf{W}$ and $\mathbf{H}$ that satisfy the following (Zhang et al. (2006))

$$\mathrm{argmax}_{\mathbf{WH}}\mathbb{E}(\log(\mathbb{P}(A^o, A^u|\mathbf{WH}))|A^o, \mathbf{WH}^{(t-1)}) \tag{8}$$

where $\mathbf{WH}^{(t-1)}$ is the product of the previous estimates of $\mathbf{W}$ and $\mathbf{H}$ and $\mathbb{P}$ is a probability measure. This problem is equivalent to running a PMF algorithm on the following adjusted matrix (Zhang et al. (2006))

$$\mathbf{A}_1 = \mathbf{C} \odot \mathbf{A} + (\mathbf{1} - \mathbf{C}) \odot \mathbf{WH}^{(t-1)} \tag{9}$$

with $\mathbf{C}_{ij} = 1$ if $\mathbf{A}_{ij}$ is known and $\mathbf{C}_{ij} = 0$ if $\mathbf{A}_{ij}$ is unknown, and $\mathbf{1}$ is a matrix of ones.

Recent work has looked into expanding on this approach to continuous weights as seen in most PMF problems of real-time atmospheric mass spectrometry data (Yahaya et al. (2019); Yahaya (2021); Yahaya et al. (2021)). To handle the continuous case, a variation of Eq. (8) is maximized (Yahaya et al. (2019))

$$\mathrm{argmax}_{\mathbf{WH}}\mathbb{E}(\log(\mathbb{P}(\mathbf{C} \odot \mathbf{A}, (\mathbf{1} - \mathbf{C}) \odot \mathbf{A}_{\mathrm{theo}}|\mathbf{WH}))|\mathbf{C} \odot \mathbf{A}, \mathbf{WH}^{(t-1)}) \tag{10}$$

where $\mathbf{C}$ is now a weight matrix containing estimates of confidence as a value between $0$ and $1$ in a given data point, and $\mathbf{A}_{\mathrm{theo}}$ is the theoretical true data. Maximizing Eq. (10) is equivalent to running any PMF algorithm on the matrix $\mathbf{A}_1$ formed in Eq. (9).

To form the confidence matrix $\mathbf{C}$ from the uncertainties matrix $\Sigma$, Yahaya (2021) suggests scaling the weights (in this case $\frac{1}{\sigma_{ij}}$) so that the maximum value is 1. However, previous testing has primarily focused on problems with binary weights (Yahaya et al. (2019); Yahaya (2021); Yahaya et al. (2021)).

## 1.6 Determining the Number of Factors

The squared weighted residual error, $Q = \sum_{i=1}^{m}\sum_{j=1}^{n}\left(\frac{\mathbf{A}_{ij} - \sum_{l=1}^{k}\mathbf{W}_{il}\mathbf{H}_{lj}}{\sigma_{ij}}\right)^2$, can be used to determine whether a given solution either overfits or underfits the data (Ulbrich et al. (2009)). If measurement errors are normally distributed, then $Q$ will follow a $\chi^2$ distribution with $mn - k(m+n)$ degrees of freedom. Thus, to avoid overfitting, the number of factors in a solution are chosen such that $Q \approx mn - k(m+n)$ (Paatero et al. (2002)). When additional error is present, measuring the convergence of $Q$ or the weighted residual error with additional factors can determine if these additional factors add much information to the model.

Another method used in determining the number of factors is the lack of rotational ambiguity of a solution (Ulbrich et al. (2009)). Consider a simple case where the data matrix $\mathbf{A}$ is the product of two rank-two matrices $\mathbf{W} = [\boldsymbol{a}, \boldsymbol{b}]$, and $\mathbf{H} = [\boldsymbol{y}, \boldsymbol{z}]^T$,

with $a, b, y$, and $z$ being column vectors. An exact solution can also be obtained by finding $\mathbf{W} = [c, d, b]$, and $\mathbf{H} = [y, y, z]$, where $c + d = a$, in a process known as factor splitting (Ulbrich et al. (2009)). Thus, a three factor solution introduces rotational ambiguity, as any two factors $c$ and $d$ can be chosen as long as they add up to $a$. The same analysis can be seen by analyzing solutions with four or more factors, and we can conclude that a large amount of rotational ambiguity is a potential sign of overfitting.

## 1.7 Data

In this paper, we use the data from "Ambient Measurements of Highly Oxidized Gas-Phase Molecules during the Southern Oxidant and Aerosol Study (SOAS) 2013," measuring highly oxidized multifunctional molecules (HOMs) over a forest site in Alabama from June 22nd 2013 to July 7th 2013 (Massoli et al. (2018)). The dataset contains mass spectra concentrations of 1,059 different ions over 27,336 different time stamps. Additionally, initial uncertainties associated with each measurement are also included. PMF was applied to the data using the PMF2 algorithm described in Paatero (1997), checking solutions from two to 10 factors, where a six factor solution was obtained (Massoli et al. (2018)). We note that some of the uncertainties were artificially increased for this PMF2 analysis. The authors concluded that a significant portion of the secondary organic aerosol (SOA) was the result of interactions between biogenic and anthropogenic emissions (Massoli et al. (2018)).

We use this six factor solution as a reference solution and test whether the RHALS algorithm can recreate formulated factors as well as those found from PMF2. Analysis of results for different numbers of factors (other than the original six identified in Massoli et al. (2018)) were not considered in order to maintain interpretability of the algorithm output. For reference, the PMF2 factor mass spectra and the time trends over all of the data are shown in Figure 1. The factor time series, as well as the time series of the total mass concentration is also shown in Figure 2. Both plots show total concentration amounts over the entire time series and mass spectra respectively. Discussion of the chemical interpretations of the data are not presented and the scope of this effort is limited to the mathematical results from implementing the RHALS algorithm.

## 2 Methods

In this section, we present the derivation of the basic weighted HALS algorithm in Section 2.1, a simple rotation algorithm in Section 2.2, our new external weighting algorithm in Section 2.3, and inclusion of L1 and L2 regularization in Section 2.4.

## 2.1 HALS Algorithm

The derivation of the unweighted algorithm for HALS is detailed in Erichson et al. (2018). Here we present a similar derivation, taking into account uncertainties in the data that act as weights. Another derivation of weighted HALS is given in Ho (2008).

The HALS algorithm applies block coordinate descent methods in order to minimize the cost function $Q_j$ by minimizing a "block," or outer product of individual factors, of $\mathbf{W}$ and $\mathbf{H}$ at a time while keeping the other factors fixed. (Erichson et al. (2018)).

$$Q_j = ||(\mathbf{R}_j - \mathbf{W}_{(:,j)}\mathbf{H}_{(j,:)}) \oslash \mathbf{\Sigma}||_F^2 \tag{11}$$

In Eq. (11), $\mathbf{R}_j$ is the $j^{\text{th}}$ residual, with

$$\mathbf{R}_j = \mathbf{A} - \sum_{i \neq j}^{k} \mathbf{W}_{(:,i)}\mathbf{H}_{(i,:)} = \mathbf{A} - \mathbf{WH} + \mathbf{W}_{(:,j)}\mathbf{H}_{(j,:)} \tag{12}$$

$\mathbf{W}_{(:,j)}$ is the $j^{\text{th}}$ column, or $j^{\text{th}}$ factor, of $\mathbf{W}$, $\mathbf{H}_{(j,:)}$ is the $j^{\text{th}}$ row/factor of $\mathbf{H}$, and $k$ is the number of factors that the algorithm is aiming to find. As defined previously, $\boldsymbol{\Sigma}$ contains the uncertainties associated with each element in the data matrix $\mathbf{A}$.

To derive update rules for HALS, partial derivatives of Eq. (11) are taken with respect to the factors $\mathbf{W}_{(:,j)}$ and $\mathbf{H}_{(j,:)}$. In order to incorporate the uncertainties ($\boldsymbol{\Sigma}$), we present a variation on the derivation presented in Erichson et al. (2018) by considering just a row ($i$) and column ($p$) of the weighted residual.

$$Q_j^i = ||(\mathbf{R}_{j(i,:)} - \mathbf{W}_{ij}\mathbf{H}_{(j,:)})\boldsymbol{\Sigma}_i^{-1}||_F^2 \tag{13}$$

$$Q_j^p = ||\boldsymbol{\Sigma}_p^{-1}(\mathbf{R}_{j(:,p)} - \mathbf{H}_{jp}\mathbf{W}_{(:,j)})||_F^2 \tag{14}$$

In Eq. (13) and Eq. (14), $\boldsymbol{\Sigma}_i$ are $\boldsymbol{\Sigma}_p$ are diagonal matrices (of size $m$ and $n$ respectively) with the diagonal elements corresponding to the elements of the $i^{\text{th}}$ row (for $\boldsymbol{\Sigma}_i$) and $p^{\text{th}}$ column (for $\boldsymbol{\Sigma}_p$) of $\boldsymbol{\Sigma}$. Expanding Eq. (13) and Eq. (14) using the fact that $||\mathbf{X}||_F^2 = \text{Tr}(\mathbf{X}^T\mathbf{X})$, where $\text{Tr}(\mathbf{A})$ denotes the trace of the matrix $\mathbf{A}$, we get

$$Q_j^i = \text{Tr}(\boldsymbol{\Sigma}_i^{-1}\mathbf{R}_{j(i,:)}^T\mathbf{R}_{j(i,:)}\boldsymbol{\Sigma}_i^{-1} - 2\mathbf{W}_{ij}\boldsymbol{\Sigma}_i^{-1}\mathbf{R}_{j(i,:)}^T\mathbf{H}_{(j,:)}\boldsymbol{\Sigma}_i^{-1} + \mathbf{W}_{ij}^2\boldsymbol{\Sigma}_i^{-1}\mathbf{H}_{(j,:)}^T\mathbf{H}_{(j,:)}\boldsymbol{\Sigma}_i^{-1}) \tag{15}$$

$$Q_j^p = \text{Tr}(\mathbf{R}_{j(:,p)}^T\boldsymbol{\Sigma}_p^{-1}\boldsymbol{\Sigma}_p^{-1}\mathbf{R}_{j(:,p)} - 2\mathbf{H}_{jp}\mathbf{R}_{j(:,p)}^T\boldsymbol{\Sigma}_p^{-1}\boldsymbol{\Sigma}_p^{-1}\mathbf{W}_{(:,j)} + \mathbf{H}_{jp}^2\mathbf{W}_{(:,j)}^T\boldsymbol{\Sigma}_p^{-1}\boldsymbol{\Sigma}_p^{-1}\mathbf{W}_{(:,j)}) \tag{16}$$

Differentiating with respect to $\mathbf{W}_{ij}$ and $\mathbf{H}_{jp}$,

$$\frac{\partial Q_j^i}{\partial \mathbf{W}_{ij}} = -2\text{Tr}(\boldsymbol{\Sigma}_i^{-1}\mathbf{R}_{j(i,:)}^T\mathbf{H}_{(j,:)}\boldsymbol{\Sigma}_i^{-1}) + 2\mathbf{W}_{ij}\text{Tr}(\boldsymbol{\Sigma}_i^{-1}\mathbf{H}_{(j,:)}^T\mathbf{H}_{(j,:)}\boldsymbol{\Sigma}_i^{-1}) \tag{17}$$

$$\frac{\partial Q_j^p}{\partial \mathbf{H}_{jp}} = -2\text{Tr}(\mathbf{R}_{j(:,p)}^T\boldsymbol{\Sigma}_p^{-1}\boldsymbol{\Sigma}_p^{-1}\mathbf{W}_{(:,j)}) + 2\mathbf{H}_{jp}\text{Tr}(\mathbf{W}_{(:,j)}^T\boldsymbol{\Sigma}_p^{-1}\boldsymbol{\Sigma}_p^{-1}\mathbf{W}_{(:,j)}) \tag{18}$$

To eliminate the matrix traces, it is easy to show that Eq. (17) and Eq. (18) can be rewritten as

$$\frac{\partial Q_j^i}{\partial \mathbf{W}_{ij}} = -2\mathbf{R}_{j(i,:)}(\mathbf{H}_{(j,:)}^T \oslash (\boldsymbol{\Sigma}_{(i,:)}^T \odot \boldsymbol{\Sigma}_{(i,:)}^T)) + 2\mathbf{W}_{ij}\mathbf{H}_{(j,:)}(\mathbf{H}_{(j,:)}^T \oslash (\boldsymbol{\Sigma}_{(i,:)}^T \odot \boldsymbol{\Sigma}_{(i,:)}^T)) \tag{19}$$

$$\frac{\partial Q_j^p}{\partial \mathbf{H}_{jp}} = -2\mathbf{R}_{j(:,p)}^T (\mathbf{W}_{(:,j)} \oslash (\boldsymbol{\Sigma}_{(:,p)} \odot \boldsymbol{\Sigma}_{(:,p)})) + 2\mathbf{H}_{jp}\mathbf{W}_{(:,j)}^T (\mathbf{W}_{(:,j)} \oslash (\boldsymbol{\Sigma}_{(:,p)} \odot \boldsymbol{\Sigma}_{(:,p)})) \tag{20}$$

Setting Eq. (19) and Eq. (20) to zero and solving for the factor values yields

$$\mathbf{W}_{ij} = \frac{\mathbf{R}_{j(i,:)}(\mathbf{H}_{(j,:)}^T \oslash (\boldsymbol{\Sigma}_{(i,:)}^T \odot \boldsymbol{\Sigma}_{(i,:)}^T))}{\mathbf{H}_{(j,:)}(\mathbf{H}_{(j,:)}^T \oslash (\boldsymbol{\Sigma}_{(i,:)}^T \odot \boldsymbol{\Sigma}_{(i,:)}^T))} \tag{21}$$

$$\mathbf{H}_{jp} = \frac{\mathbf{R}_{j(:,p)}^T (\mathbf{W}_{(:,j)} \oslash (\boldsymbol{\Sigma}_{(:,p)} \odot \boldsymbol{\Sigma}_{(:,p)}))}{\mathbf{W}_{(:,j)}^T (\mathbf{W}_{(:,j)} \oslash (\boldsymbol{\Sigma}_{(:,p)} \odot \boldsymbol{\Sigma}_{(:,p)}))} \tag{22}$$

Substituting Eq. (12) into Eq. (19) and Eq. (20) yields the update rules:

$$\mathbf{W}_{ij} \leftarrow \left[ \mathbf{W}_{ij} + \frac{\mathbf{A}_{(i,:)}(\mathbf{H}_{(j,:)}^T \oslash (\boldsymbol{\Sigma}_{(i,:)}^T \odot \boldsymbol{\Sigma}_{(i,:)}^T)) - \mathbf{W}_{(i,:)}\mathbf{H}(\mathbf{H}_{(j,:)}^T \oslash (\boldsymbol{\Sigma}_{(i,:)}^T \odot \boldsymbol{\Sigma}_{(i,:)}^T))}{\mathbf{H}_{(j,:)}(\mathbf{H}_{(j,:)}^T \oslash (\boldsymbol{\Sigma}_{(i,:)}^T \odot \boldsymbol{\Sigma}_{(i,:)}^T))} \right]_+ \tag{23}$$

$$\mathbf{H}_{jp} \leftarrow \left[ \mathbf{H}_{jp} + \frac{\mathbf{A}_{(:,p)}^T (\mathbf{W}_{(:,j)} \oslash (\boldsymbol{\Sigma}_{(:,p)} \odot \boldsymbol{\Sigma}_{(:,p)})) - \mathbf{H}_{(:,p)}^T \mathbf{W}^T (\mathbf{W}_{(:,j)} \oslash (\boldsymbol{\Sigma}_{(:,p)} \odot \boldsymbol{\Sigma}_{(:,p)}))}{\mathbf{W}_{(:,j)}^T (\mathbf{W}_{(:,j)} \oslash (\boldsymbol{\Sigma}_{(:,p)} \odot \boldsymbol{\Sigma}_{(:,p)}))} \right]_+ \tag{24}$$

where the operator $[\cdot]_+$ projects all negative update values to 0. In practice, the authors in Erichson et al. (2018) utilize the following simplified form of Eq. (23) and Eq. (24):

$$\mathbf{W}_{ij} \leftarrow \left[ \mathbf{W}_{ij} - \frac{\nabla_{\mathbf{W}_{ij}} Q_j^i}{\nabla_{\mathbf{W}_{ij}}^2 Q_j^i} \right]_+ \tag{25}$$

$$\mathbf{H}_{jp} \leftarrow \left[ \mathbf{H}_{jp} - \frac{\nabla_{\mathbf{H}_{jp}} Q_j^p}{\nabla_{\mathbf{H}_{jp}}^2 Q_j^p} \right]_+ \tag{26}$$

Thus, one can add additional auxiliary functions to the cost function $Q$, such as regularization and rotation terms, and add them to the update rules based on the new gradient and Hessian values. Furthermore, writing the update rules as in Eq. (25) and Eq. (26) includes the calculation of the projected gradient. This can be used as a stopping condition criterion that avoids the computational costs of calculating other convergence statistics (Erichson et al. (2018)).

We note that when the uncertainties are equal to one, $\boldsymbol{\Sigma}$ can be disregarded in Eq. (23) and Eq. (24), and the update rules are identical to those in Erichson et al. (2018). In Eq. (23) and Eq. (24), the Hessians $\mathbf{H}(\mathbf{H}_{(j,:)}^T \oslash (\boldsymbol{\Sigma}_{(i,:)}^T \odot \boldsymbol{\Sigma}_{(i,:)}^T))$ and $\mathbf{W}^T (\mathbf{W}_{(:,j)} \oslash (\boldsymbol{\Sigma}_{(:,p)} \odot \boldsymbol{\Sigma}_{(:,p)}))$ should be found prior to multiplication by $\mathbf{W}_{(i,:)}$ and $\mathbf{H}_{(:,p)}^T$ respectively to minimize computational costs. We do not preallocate the products $\mathbf{WH}$ and $\mathbf{H}^T\mathbf{W}^T$, which is also not done in Erichson et al. (2018).

## 2.2 Rotational Considerations

Rotating solutions to induce slight variations in the output of a PMF algorithm may be necessary, especially in cases where the interpretation of the solutions yields some unrealistic results (e.g. a factor is zero during a period in which it is expected to be present, or two factors appear mixed in their time series and/or spectra). As mentioned in Section 1.4, we do not attempt to constrain these "rotations" to be norm preserving. However, it is possible to find approximate rotations. As detailed in Paatero and Hopke (2009), for a specific factor value $\mathbf{W}_{ij}$ or $\mathbf{H}_{jp}$, an auxiliary term can be added to the cost function to pull the component towards a set value $\mathbf{W}_{ij}^*$ or $\mathbf{H}_{ij}^*$. Defining

$$Q_{\mathrm{aux}_W}^{ij} = s(\mathbf{W}_{ij} - \mathbf{W}_{ij}^*)^2 \text{ or } Q_{\mathrm{aux}_H}^{pj} = s(\mathbf{H}_{jp} - \mathbf{H}_{jp}^*)^2 \tag{27}$$

with $s$ determining the strength of the pull, we find $\nabla_{\mathbf{W}_{ij}} Q_{\mathrm{aux}_W}^{ij} = 2s(\mathbf{W}_{ij} - \mathbf{W}_{ij}^*)$, $\nabla_{\mathbf{W}_{ij}}^2 Q_{\mathrm{aux}_W}^{ij} = 2s$, $\nabla_{\mathbf{H}_{j,p}} Q_{\mathrm{aux}_H}^{pj} = 2s(\mathbf{H}_{jp} - \mathbf{H}_{jp}^*)$, and $\nabla_{\mathbf{H}_{jp}}^2 Q_{\mathrm{aux}_H}^{jp} = 2s$. Adding these terms to the gradient and Hessian in Eq. (25) and Eq. (26), we can derive new update rules for $\mathbf{W}_{ij}$ and $\mathbf{H}_{jp}$

To probe possible rotations of entire factors, we introduce "pulling equations" to the cost function, which pull the elements of $\mathbf{W}$ and $\mathbf{H}$ to the desired rotations:

$$Q_{\mathrm{aux}_H}^{(j)} = a(\sum_{i=1}^{n}(1 - \mathbf{H}_{ji}))^2 \tag{28}$$

$$Q_{\mathrm{aux}_W}^{(j)} = b(\sum_{i=1}^{m}\mathbf{W}_{ij})^2 \tag{29}$$

These equations correspond to $r < 0$, with both $a$ and $b$ positive. For $r > 0$, $\mathbf{W}$ and $\mathbf{H}$ would be interchanged. These pulling equations are similar to those introduced in Paatero et al. (2002), although their meanings are slightly changed.

Taking derivatives of Eq. (28) and Eq. (29)

$$\nabla_{\mathbf{H}_{jp}} Q_{\mathrm{aux}_H}^{jp} = -a(1 - \mathbf{H}_{jp}) \tag{30}$$

$$\nabla_{\mathbf{H}_{jp}}^2 Q_{\mathrm{aux}_H}^{jp} = a \tag{31}$$

$$\nabla_{\mathbf{W}_{ij}} Q_{\mathrm{aux}_W}^{ji} = b\mathbf{W}_{ij} \tag{32}$$

$$\nabla_{\mathbf{W}_{ij}}^2 Q_{\mathrm{aux}_W}^{ji} = b \tag{33}$$

we add these terms to the gradient and Hessian in Eq. (25) and Eq. (26) to receive new update rules. Optimally, we would expect these equations to pull the values of $\mathbf{H}$ towards more positive values and the values of $\mathbf{W}$ towards zero, which mirrors the effect of setting $r$ to be less than zero in Eq. (6). The end result may be different solutions which are more realistic and interpretable.

## 2.3 External Weighting

To perform external weighting, we first find $\hat{\mathbf{A}} = \mathbf{A} \oslash \mathbf{\Sigma}$ (i.e. we divide the data elementwise by the uncertainties). Note that the uncertainty matrix $\mathbf{\Sigma}$ must only contain nonzero real entries, and therefore the algorithm cannot handle problems with binary weights (such as PMF problems with missing entries). One approach to PMF with binary weights is the expectation maximization (EM) approach detailed in Section 1.5.1 (Yahaya et al. (2021); Zhang et al. (2006)).

After elementwise division, RHALS is applied to $\hat{\mathbf{A}}$ to form the estimated scaled factor matrices $\hat{\mathbf{W}}$ and $\hat{\mathbf{H}}$. The unscaled estimates $\mathbf{W}$ and $\mathbf{H}$ are found by the relation

$$\mathbf{WH} = (\hat{\mathbf{W}}\hat{\mathbf{H}}) \odot \mathbf{\Sigma} \tag{34}$$

$\mathbf{W}$ and $\mathbf{H}$ can then be found iteratively via alternating least squares:

$$\mathbf{W} = ((\hat{\mathbf{W}}\hat{\mathbf{H}}) \odot \mathbf{\Sigma})\mathbf{H}^T(\mathbf{HH}^T)^{-1} \text{ or } \mathbf{W} = ((\hat{\mathbf{W}}\hat{\mathbf{H}}) \odot \mathbf{\Sigma})\mathbf{H}^\dagger \tag{35}$$

$$\mathbf{H} = (\mathbf{W}^T\mathbf{W})^{-1}\mathbf{W}^T((\hat{\mathbf{W}}\hat{\mathbf{H}}) \odot \mathbf{\Sigma}) \text{ or } \mathbf{H} = \mathbf{W}^\dagger((\hat{\mathbf{W}}\hat{\mathbf{H}}) \odot \mathbf{\Sigma}) \tag{36}$$

Here, $\mathbf{W}^\dagger$ and $\mathbf{H}^\dagger$ denote the pseudoinverses of $\mathbf{W}$ and $\mathbf{H}$, and all negative elements are set to zero after the factor matrix is updated. Mathematically, the two methods for calculating $\mathbf{W}$ and $\mathbf{H}$ detailed in Eq. (35) and Eq. (36), respectively, are identical, as long as the ranks of the factor matrices are equal to $k$. A short proof of this is detailed in Section B of the Appendix. Of course, if the factor matrices were of lower rank (or contained very small singular values), then a lower rank factorization should first be found instead. Since both algorithms are identical mathematically, one could be favorable if it provided a speed advantage, depending on the computational efficiency of the pseudoinverse algorithm called. Both update rules are $\mathcal{O}(mnk)$ (Feng et al. (2018)), and running this code in MATLAB on a single CPU, we found the update rules using the pseudoinverses were faster.

To begin the iteration, we have to initialize either $\mathbf{W}$ or $\mathbf{H}$. The most intuitive way to do this is by assuming the unweighted factors are similar to the weighted factors, and setting $\mathbf{H}$ (or $\mathbf{W}$) to the weighted values, but scaled to the magnitude of the original data. Since each entry of the data matrix is approximated as $\mathbf{A}_{ij} = \sum_{l=1}^{k} \mathbf{W}_{il}\mathbf{H}_{lj}$, and assuming relatively equal magnitudes for $\mathbf{W}$ and $\mathbf{H}$, the appropriate scale factor should be $\sqrt{\frac{\bar{\mathbf{A}}}{k}}$ where $\bar{\mathbf{A}}$ denotes the element-mean of $\mathbf{A}$. When $\mathbf{W}$ and $\mathbf{H}$ contain values with greatly differing magnitudes, perhaps a different scaling factor should be used, although this has not been extensively tested. Thus, the algorithm should be initialized by

$$\mathbf{H}_0 = \left( \sqrt{\frac{\bar{\mathbf{A}}}{k\bar{\hat{\mathbf{H}}}^2}} \right) \hat{\mathbf{H}} \tag{37}$$

In Eq. (37), $\bar{\hat{\mathbf{H}}}$ denotes the element-mean of $\hat{\mathbf{H}}$. Measuring the change in $\mathbf{W}$ and $\mathbf{H}$ between iterations, the algorithm typically converges relatively quickly - within 20 to 40 iterations. In practice, one can use L2 regularization in the external weighting

steps, equal to 0.01 for our data, as the least squares method may become increasingly ill-posed as the number of factors increases. This value may need to be altered based on the magnitude of the values in the data as well as the number of factors. However, we found that adding L2 regularization lowered the similarity of the factors to the given factors from the solution using PMF2.

We used this method for all externally weighted algorithms tested in Section 4. Theoretically, any PMF algorithm could be used for the post-processing step. This may become relevant, since as noted in Section 1.2.3, the nonnegative ALS method described above may have convergence issues for certain factorization problems (Gillis (2020)).

## 2.4  Regularization

As in the RHALS algorithm presented in Erichson et al. (2018), we add L1 and L2 regularization. L1 regularization is added to
the cost function through the L1 norm, $||\cdot||_1$, which is the sum of the absolute value of the components in a given row/column in the factor matrices. L2 regularization is added through the L2, or Euclidean, norm, $||\cdot||_2^2$. L1 regularization is typically added to reduce sparsity, while L2 regularization is added to control the Euclidean norms of the factors as well as avoid overfitting ill posed problems (Erichson et al. (2018)). The cost function now becomes

$$||(\mathbf{R}_j - \mathbf{W}_{(:,j)}\mathbf{H}_{(j,:)}) \oslash \mathbf{\Sigma}||_F^2 + \alpha||\mathbf{W}_{(:,j)}||_1 + \beta||\mathbf{H}_{(j,:)}||_1 + \gamma||\mathbf{W}_{(:,j)}||_2^2 + \delta||\mathbf{H}_{(j,:)}||_2^2 \tag{38}$$

In Eq. (38), $\alpha, \beta, \gamma$, and $\delta$ control the amount of regularization that is added. Typically $\alpha$ and $\beta$, as well as $\gamma$ and $\delta$, are set equal. $\nabla_{\mathbf{W}_{ij}}Q_j^i = \beta + 2\gamma\mathbf{W}_{ij}$, $\nabla_{\mathbf{W}_{ij}}^2 Q_j^i = 2\gamma$, and $\nabla_{\mathbf{H}_{jp}}Q_j^p = \alpha + 2\delta\mathbf{H}_{jp}$, $\nabla_{\mathbf{H}_{jp}}^2 Q_j^p = 2\delta$. These terms are added to the update rules in Eq. (25) and Eq. (26). Optimal parameter values for $\alpha, \beta, \gamma$, and $\delta$ in Eq. (38) are often found using an L curve analysis, which measures the tradeoff between minimizing a norm and minimizing the residual error (Hansen and O'Leary (1993)).

    In the SOAS dataset, the average data value is $1.9969 \times 10^{-5}$ for the mass-to-charge ratio. When external weighting is
applied, the average value of $\mathbf{A} \oslash \mathbf{\Sigma}$ is 1.4851. L curves below are thus plotted using two different scales: one with a scale of $10^{-5}$, and one with a scale of one. The deterministic, unweighted algorithm is first applied to a small dataset consisting of the first 100 rows and columns of the SOAS dataset, scaled to contain different average values, to produce the L curves shown in Figure 3. Values of the regularization parameters are listed on the graphs, with "e" representing the exponential function with base 10, and the number to the right is the exponent (e.g. $e03 = 10^3$). The optimal regularization value is one that is located in
the bend of the graph, meaning it minimizes a solution's norm while having low residual error. In our results, we put preference on minimizing the latter, so we choose regularization values towards the left side of the bend. Analyzing the figures, the L1 regularization parameters $\alpha$ and $\beta$ are chosen between $1 \times 10^{-6}$ and $1 \times 10^{-5}$ for the scale of $10^{-5}$, and between 0.1 and one for the scale of one. The L2 regularization parameters $\gamma$ and $\delta$ are chosen around 0.001 for the scale of $10^{-5}$, and 100 for the scale of one. We note that choosing much smaller regularization values does not drastically increase the norms of the solution,
suggesting that regularization is not especially necessary for this problem. However, for more ill-posed problems, the L1 and L2 norms may become extremely large as the amount of regularization tends to zero (Hansen and O'Leary (1993)).

    As a rule of thumb, the L1 regularization parameters should be chosen to be on the same order of magnitude as the data, and the L2 regularization parameters should be chosen to be around 100 times that. Note that neither the data matrix size nor

the number of factors affect these optimal regularization parameter values. External and internal weighting will also not affect these values, as the magnitudes of the gradient and the Hessian will be around the same value.

An increasingly popular alternative to traditional regularization is nuclear norm regularization, which can be applied to matrix factorization without the need for rank constraints (Hu et al. (2013); Sun and Mazumder (2013); Fornasier et al. (2011)). The nuclear norm is defined as

$$||\mathbf{A}||_* = \sum_{i=1}^{n} \sigma_i(\mathbf{A}) \tag{39}$$

where $\sigma_i(\mathbf{A})$ is the $i^{\text{th}}$ singular value of $\mathbf{A}$. It is the convex envelope to the rank function, and thus finding a PMF solution that minimizes the nuclear norm also has a minimal rank (Hu et al. (2013)). Two possible ways to apply nuclear norm regularization to weighted PMF are by the alternating direction method of multipliers (ADMM), as demonstrated in Sun and Mazumder (2013), and by reconstruction of the nuclear norm into a Frobenius norm, which can then be treated as L2 regularization (Fornasier et al. (2011)). However, both these approaches involve key computations with the large, low-rank product $\mathbf{WH}$, and may be computationally expensive to implement. Furthermore, both algorithms still involve some arbitrary choice of rank by requiring a present amount of nuclear norm regularization. In traditional PMF, factor profiles are optimized from a predetermined number of factors. Our implementation and methdology are consistent with this approach.

## 3  Computational Details

The computer used for these calculations has an 11th Gen Intel® Core™ i7-1165G7 dual quad-core CPU with a speed of 2.80GHz. MATLAB™ R2021a is used for all calculations. While no algorithms detailed in this paper explicitly utilize multi-core processing, they may rely on built in MATLAB functions (such as svd()) that utilize multicore processing for increased efficiency.

## 4  Results

We measure the accuracy of RHALS using the weighted residual error of the final algorithm again defined as $\sqrt{\sum_{i=1}^{m} \sum_{j=1}^{n} \left( \frac{\mathbf{A}_{ij} - \sum_{l=1}^{k} \mathbf{W}_{il} \mathbf{H}_{lj}}{\sigma_{ij}} \right)^2}$, the correlation coefficients between the time series factors produced by RHALS and those of the reference solution, and the cosine similarity between the mass spectra profiles produced by RHALS and the reference solution.

In the MU algorithm, the elements of $\mathbf{W}$ and $\mathbf{H}$ are initialized by taking the absolute value of random numbers drawn from a standard normal distribution. These values are then scaled by a factor of $\bar{\mathbf{A}}^2$. The ALS and HALS algorithms are initialized by the nonnegative double SVD (NNDSVD) approach detailed in Boutsidis and Gallopoulos (2008). Note that the latter approach will still vary between random seeds as the nonnegative NNDSVD is also found from utilizing the randomization technique in Section 1.3.

All algorithms cease updates when a stopping criterion is met or the maximum number of iterations is reached. For the MU and ALS algorithms, the value of the cost function is calculated after each iteration, and the algorithm halts when the percent change in this value, divided by the initial cost function value, is less than a set tolerance. The HALS algorithms differs by using the percent change in the projected gradient in its stopping criterion (detailed in Lin (2007)). The projected gradient is defined as (Erichson et al. (2018))

$$
\nabla^P_{\mathbf{W}_{ij}} \begin{cases} \nabla_{\mathbf{W}_{ij}} \text{ if } \mathbf{W}_{ij} > 0 \\ \min(0, \nabla_{\mathbf{W}_{ij}}) \text{ if } \mathbf{W}_{ij} = 0 \end{cases} . \tag{40}
$$

and similarly for $\nabla^P_{\mathbf{H}_{jp}}$. When the percentage change in $\sum_{i=1}^m \sum_{j=1}^k (\nabla^P_{\mathbf{W}_{ij}})^2 + \sum_{i=1}^k \sum_{j=1}^n (\nabla^P_{\mathbf{H}_{ij}})^2$ (divided by its initial value) is less than a specified tolerance, the algorithm halts. Below, the maximum number of iterations allowed is set to be 100 (which is rarely reached by any algorithm), and the tolerance is set to be $1 \times 10^{-4}$ for the HALS and ALS algorithms. A tolerance of $10^{-5}$ is used for the MU algorithm to account for a larger initial cost from the random initialization.

When comparing factors from different algorithms, it is important to note that the ratio of values between a time series factor and a mass spectra factor might be slightly different. Consider two algorithms that compute the identical factorizations

$$
\mathbf{W}_1 \mathbf{H}_1 = \sum_{l=1}^k \mathbf{W}^{(1)}_{(:,l)} \mathbf{H}^{(1)}_{(l,:)} \text{ and } \mathbf{W}_2 \mathbf{H}_2 = \sum_{l=1}^k \mathbf{W}^{(2)}_{(:,l)} \mathbf{H}^{(2)}_{(l,:)} = \sum_{l=1}^k \left( c_l \mathbf{W}^{(1)}_{(:,l)} \right) \left( \frac{1}{c_l} \mathbf{H}^{(1)}_{(l,:)} \right)
$$

with $c_l$ being an arbitrary positive constant. It can be seen above that $\mathbf{W}^{(2)}_{(:,l)}$ is identical to $\mathbf{W}^{(1)}_{(:,l)}$ scaled in magnitude by $c_l$. It would appear that one factorization yields a stronger signal for factor $l$, despite the fact that the factorizations are identical. Thus, when comparing individual factors between algorithms, the time series concentrations are cumulative over the entire mass spectra, and the mass spectra concentrations are cumulative over the entire time series (as also presented with the PMF2 results). Specifically, we plot the time series of factor $l$ as the sum over the columns of the outer product $\mathbf{W}_{(:,l)} \mathbf{H}_{(l,:)}$, and we plot the mass spectra of factor $l$ as the sum over the rows. To avoid this issue, one can add scaling coefficients to rescale the factors at each iteration so that they will approximately have the same magnitude (Lu and Wu (2004)). This has not yet been implemented in our code.

### 4.1 Computational Efficiency

Table 1 shows the number of operations required at each step (update of $\mathbf{W}$ and $\mathbf{H}$). Preprocessing steps such as finding $\mathbf{A} \oslash \Sigma$ and $\Sigma \odot \Sigma$ are excluded from the table. External weighting eliminates almost all elementwise operations needed, which is a slow memory bound operation (Jia et al. (2020)). However, systems with an abundance of free memory and/or GPUs may find that internal weighting methods are sufficiently quick for small or medium size problems. Furthermore, the performance of matrix operations within these algorithms may vary between programming languages and libraries, such as between the numpy, numexpr, and Theano libraries in Python (Bergstra et al. (2010)).

Analyzing Table 1 allows us to see why, in practice, it takes longer for internally weighted ALS and HALS to converge than internally weighted MU, as the former will have more matrix multiplication operations and elementwise operations for our

target rank, $k = 6$. However, when the algorithms are internally weighted, ALS and HALS become much more computationally efficient, allowing runtime to diminish as well.

To show numerically how the cost of RHALS scales with the size of the system, we consider the size of the data matrix vs the run time of the RHALS algorithm, as well as the deterministic internally weighted HALS and MU algorithms. Figure 4 shows how the computational costs of the different algorithms vary with the number of rows (and columns) in a square data matrix. The number of factors is set to five, and the data are formed by multiplying two rank-five matrices filled with sampled values from the PMF2 solution of the SOAS dataset. Uncertainty values are also randomly sampled from the given uncertainty

data, and noise is added to the data matrix through random normal values centered around 0 with standard deviations equal to the uncertainties. L1 regularization is set to one for all factors, and L2 regularization is set to $50$.

    As seen in Figure 4, the logarithms of size and run time are approximately linear for all three algorithms. RHALS clearly outperforms MU and HALS in terms of efficiency, requiring roughly $5\%$ and $1\%$ of the run time of the MU and deterministic HALS algorithms respectively. It should also be noted that the randomization step, which accounts for about $5\%$ of the

435 algorithm computational costs, can be parallelized using GPUs (Erichson et al. (2018)). The post algorithm external weighting step can also be parallelized as it only involves matrix multiplication and an SVD (Lahabar and Narayanan (2009)). This post weighting step accounts for around $60\%$ of the algorithm's computational costs.

## 4.2   Simple Case

We now present the RHALS algorithm for a small test case. To do this, we will form a data matrix by combining underlying

"true" factors and random noise. We form a 5,952 by 400 matrix using the first three factors from the PMF2 solution to the SOAS dataset. We choose the time stamps between 9,385 and 15,336 from the PMF2 time series factor matrix $\mathbf{W}_{\mathrm{PMF2}}$ as the PMF2 factors are clearly distinguishable from each other in this time interval. We choose the first 400 mass spectra profiles from the mass spectra factor matrix $\mathbf{H}_{\mathrm{PMF2}}$ as the bulk of the mass spectra concentrations lie within this range.

    A data matrix is then formed by combining the factors and adding random normal noise with a standard deviation equal to

445 that of the uncertainty of the data:

$$\mathbf{A}_{ij}^{\mathrm{test}} = \max(0, \mathbf{W}_{(i,:)}^{\mathrm{test}} \mathbf{H}_{(:,j)}^{\mathrm{test}} + N(0, \mathbf{\Sigma}_{ij})) \tag{41}$$

Next, we run RHALS over 20 trials, each with a different initialization, in order to obtain many different solutions that exist in the solution space. Here, initializations are formed using the randomized SVD method described earlier, which generated sufficiently different solutions. Figure 5 shows the weighted residual error of the RHALS algorithm over 20 trials.

The average weighted error over 20 trials is $1.4749 \times 10^3$, and the algorithm took an average time of 0.0561 seconds to converge with an average of 48.15 block coordinate descent steps per trial. As one can see, solutions can vary over different trials, and a smaller weighted error could be used as justification of one solution over another.

    Next, we compare the similarities of the converged factors and the original factors that formed the test case. The averages of the similarities between the three factors are shown in Figure 6. A cosine similarity is used for the mass spectra, while the

455 correlation coefficient is used for the time series.

As one can see, the RHALS algorithm recreates the mass spectra and time series factors almost perfectly for all trials tested, regardless of the weighted residual error. In practice, a cosine similarity over $0.95$ or a correlation coefficient over $0.90$ between factor profiles represent almost identically interpretable solutions. Any trial is viable to be chosen as a "good" solution. Each solution yields a mass spectra similarity over $0.994$, and a time series correlation over $0.974$. There is low variance in the similarity metrics among trials and higher weighted error corresponds with less important parts of the factor.

## 4.3   Large Dataset

### 4.3.1   Comparing Different Algorithms

Next, we analyze the complete dataset, and compare the RHALS factors to all PMF2 factors. Table 2 shows a table of diagnostics for the HALS, RHALS, ALS, and MU algorithms applied to the complete SOAS dataset, averaged over three trials. As one can see, algorithms with external weighting demonstrate a dramatic reduction in computational costs, albeit at a larger error. The ALS algorithms were by far the slowest, and the internally weighted HALS algorithm yielded the lowest average weighted error. The internally weighted MU algorithm produced comparable results to the internally weighted HALS algorithm in less than two thirds of the time; however, both the internally and externally weighted cases took the most steps to converge. The two fastest algorithms were the externally weighted HALS and RHALS algorithms, taking 1.01 and 0.50 seconds respectively.

PMF factor profiles are often visually analyzed and compared to known candidate profiles for identification. Thus, the question arises as to whether or not an algorithm utilizing external weighting still contains interpretable factors in light of the decreased accuracy. Figure 7 shows the cumulative time series from one solution of the externally and internally weighted HALS algorithm, with the factors produced by the externally weighted algorithm and the internally weighted algorithm overlaid. Figure 8 shows the cumulative mass spectra of this solution, with the factors from the externally weighted algorithm and the internally weighted algorithm laid out side by side. Similar graphs for the MU and ALS algorithms are detailed in the Appendix.

Upon visual inspection, the time series produced by the externally weighted factors have peaks and troughs at almost identical times, but the magnitudes of these peaks and troughs can vary throughout the factors. Interestingly, there exists a consistent difference in the magnitude of the time series between internal and external weighting. Specifically, the external weighting algorithms seem to consistently overpredict the concentrations of the time series of the second factor compared to the internally weighted algorithms. The difference is small compared to the concentrations of the factors, but this may lead to an over interpretation of the importance of the second factor with an externally weighted algorithm. It should be noted that there exists a (similarly) large variation in relative factor signals between trials of internally weighted algorithms, although caution should be taken with regards to the possibility that external weighting introduces extra error in this analysis. More research is needed to understand the overestimation or underestimation of factor magnitudes in external weighting.

Comparing the mass spectra yields a similar analysis. Most factors of externally weighted algorithms share the same spikes of key ions as in the internally weighted ions, although sometimes at different magnitudes. Occasionally, some externally weighted mass spectra factors will look quite dissimilar to the corresponding internally weighted factor, or contain noticeable

spikes and divots in key ions, as seen in the second factor for the MU and HALS algorithm. It should also be noted that the solutions of the externally weighted algorithms could be rotated away from each other. These differences may be extreme enough to encourage users to utilize multiple trials to search for multiple solutions.

### 4.3.2 Comparison Between Expectation Maximization and External Weighting

To test the EM approach to uncertainties weighting as mentioned in Section 1.5.1, the weights $\sigma_{ij}$ in the uncertainties matrix $\mathbf{\Sigma}$ are scaled so that $\max_{ij}\left(\frac{1}{\sigma_{ij}}\right) = 1$. Since the bulk computational component of the algorithm is constructing the matrix $\mathbf{A}_1$ in Eq. 9, the authors in Yahaya et al. (2019) and Yahaya et al. (2021) recommend updating $\mathbf{A}_1$ only after convergence or a maximum number of iterations of 20 or 50. They also note that applying the expectation step too early in the algorithm led to poorer performance due to the amount of error in the estimates of $\mathbf{W}$ and $\mathbf{H}$. In order to apply the projected gradient stopping criterion discussed in Section 4, we chose to reconstruct $\mathbf{A}_1$ at fixed iterations - after 10 and 20 PMF steps for different experiments. The first construction of $\mathbf{A}_1$ is evaluated at the first, fifth, and $10^{\text{th}}$ step. We used the NNDSVD initialization in Boutsidis and Gallopoulos (2008), and also varied the tolerance of the stopping condition between $10^{-4}$ and $10^{-6}$. We summarize these results by presenting the range of average values across the different variations.

We present a comparison of external weighting and the EM algorithm in Table 3, using the ranges of values from the different experiments listed above. Each value is an average over 20 trials. We compare the convergence times of the algorithms, the number of steps, the weighted errors, and the similarity to the PMF2 solution for both $\mathbf{W}$ (correlation) and $\mathbf{H}$ (cosine similarity).

As demonstrated in Table 3, some variations of the EM algorithm were able to outperform externally weighted HALS and RHALS in total time, as well as in weighted error. For instance, running the expectation maximization algorithm with the first calculation of $\mathbf{A}_1$ taking place at the $10^{\text{th}}$ step, and recalculating $\mathbf{A}_1$ after 20 additional steps, until the convergence criterion with a tolerance of $10^{-5}$ was reached, yielded an average weighted error of $6.92 \times 10^3$ in 0.6703 seconds. This computational speed compares to RHALS, while the accuracy bests both RHALS and externally weighted HALS. However, no expectation maximization algorithm was as successful at recreating the PMF2 time series factors, as seen in column 5 of Table 3. Externally weighted RHALS and HALS also provided mass spectra factors with a higher similarity to the PMF2 factors, with the exception of two runs of the expectation maximization algorithm, one recalculating $\mathbf{A}_1$ after 20 steps, with the first calculation at the fifth step, and the other after 10 steps, with the first calculation at the first step. These yielded average similarities of 0.8856 and 0.8861 respectively, although they both took over 2.90 seconds to run, much slower than any other algorithm tested besides internally weighted HALS.

We also tested how well each algorithm produced factors that were within a rotation of the PMF2 factors. As detailed in Section 1.4, the factor profiles of $\hat{\mathbf{W}}\hat{\mathbf{H}} = \mathbf{W}\mathbf{T}^{-1}\mathbf{T}\mathbf{H}$ for a square matrix $\mathbf{T}$ may be closer to the desired solution than the original factors $\mathbf{W}$ and $\mathbf{H}$. To see the extent that $\mathbf{W}$ can be rotated towards $\mathbf{W}_{\text{PMF2}}$, we find the matrix $\mathbf{T}$ that minimizes the total squared differences between $\mathbf{H}_{\text{PMF2}}$ and $\mathbf{T}\mathbf{H}$, and then find the average correlation between the rows of $\mathbf{W}_{\text{PMF2}}$ and $\mathbf{W}\mathbf{T}^{-1}$. A symmetrical approach can be made to find the cosine similarity for $\mathbf{H}$. For internally weighted HALS, externally weighted HALS, and RHALS, we found average post rotation time series correlations to be 0.9391, 0.9021, and 0.8902, respectively, and post rotation mass spectra similarities to be 0.9602, 0.9433, and 0.9518, respectively. When this approach was tested

on HALS and RHALS using the EM algorithm, average post rotation time series correlations varied within $0.8528 - 0.8819$ and $0.8314 - 0.8544$, respectively, and post rotation mass spectra similarities within $0.9230 - 0.9363$ and $0.9147 - 0.9287$, respectively.

Ultimately, we found that external weighting recreated the PMF2 factors more consistently than EM, both before and after rotation. This may be due to the fact that the scaling of the weights used for the EM step is not perfectly analogous to creating a set of weights that represent the confidence of each data point. Thus, the EM method using this scaling may not capture the key error weighted patterns in the data as well as external weighting.

### 4.3.3    Complete Analysis of RHALS Algorithm

Figure 9 shows the weighted residual error for a six factor solution (equivalent to the number of factors in PMF2) over 20 different nonnegative SVD initializations. The average weighted residual error is $7.2743 \times 10^3$, with a convergence time of $0.5163$ seconds over $38.45$ steps per trial. If the solutions differ significantly, weighted residual error is a useful metric for choosing a solution from the 20 trials. The $11^{\text{th}}$ and $13^{\text{th}}$ trials appear promising, with the lowest weighted residual errors of $7.1474 \times 10^3$ and $7.1026 \times 10^3$, respectively. The average similarities between RHALS factors and PMF2 factors are presented in Figure 10, with the same similarity metrics as listed above.

We see that different initializations can lead to different solutions, in terms of both similarity to the given PMF2 solution and weighted error, suggesting that convergence to a global minimum isn't always achieved. This further emphasizes the importance of using multiple initializations in order to find an optimal solution.

Only the second, $11^{\text{th}}$, $13^{\text{th}}$ and $17^{\text{th}}$ trials have an average cosine similarity between mass spectra factors over $0.95$, along with a correlation over $0.90$ between time series factors. The $13^{\text{th}}$ trial holds the highest similarity scores, with a time series correlation of $0.9468$ and a cosine similarity of $0.9787$. As seen in Figure 9, these solutions have a small weighted residual error, further justifying picking a solution with low weighted error.

As we saw with the small test case, every RHALS solution contained factors similar to the true factors. However, only four out of 20 trials produced solutions surpassing $0.95$ in mass spectra similarity and $0.90$ in time series correlation for the entire dataset. Running 100 trials, it was found that RHALS found a solution matching these criteria $27\%$ of the time. Thus the probability of not finding a good solution in 10 trials would be around $(1 - 0.27)^{10} \approx 4.3\%$, and $\approx 0.2\%$ in 20 trials. This rate could vary depending on the number of factors and the rotational ambiguity of the problem, as in the "bad" trials RHALS may simply be finding a rotated version of the "true" solution.

In the potential case that interest lies in factors that occur most frequently, an alternative approach to picking a solution given multiple trials would be to focus on solutions in which factors are found repeatedly. In Figure 11, we compare the average similarities of a given trial to the other 19 trials. Again, cosine similarity is used for the mass spectra and the correlation coefficient is used for the time series. The bottom plot shows the two graphs averaged to give a total metric of the similarity of a solution to the other solutions.

Analyzing these figures, it appears that most solutions have about the same similarity to each other, with a time series correlation of around $0.87$ and a mass spectra similarity of around $0.93$. A few solutions, such as those in the trials between 6

and 9, can be ruled out as outliers, due to a low similarity to the other solutions. While the $12^{th}$ trial yields the highest similarity between it and the other solutions (0.9148 averaged between the time series correlation and the mass spectra similarity), no solution drastically outperforms any of the other solutions. Note that the solutions from the second, $11^{th}$, $13^{th}$ and $17^{th}$ trials all perform well in this analysis, with average similarities of 0.9028, 0.8993, 0.9016, and 0.9036, respectively.

From visual inspection of figures 9 and 11, the solution from the $13^{th}$ trial is barely outperformed in the above analysis, while holding a clear lead in accuracy. Thus, it seems to be a rational choice to decide that this solution is the "best." The solution from the $13^{th}$ trial is plotted in Figures 12 and 13. Figure 12 shows the cumulative mass spectra profiles from the RHALS solution, while Figure 13 shows the cumulative time series profiles from the RHALS solution compared with those from the PMF2 solution. Again, the profiles are plotted as a sum over the entire time series and mass spectra, respectively.

The similarities of the RHALS factors to the "true" PMF2 factors are very promising; however, the magnitude of some factors differ. Specifically, RHALS underestimates the significance of the first factor, while overestimating the significance of the second and fifth factors. Again, this additional error should be considered when interpreting RHALS factors.

Another potential concern from the RHALS algorithm is that the factors have a general bias towards higher or lower magnitudes. Specifically, since RHALS enforces nonnegativity merely by setting negative elements to zero, in addition to regularization, it could be hypothesized that RHALS would produce factors of lower magnitude than the actual data. In Figure A1, the time series of the six RHALS factors, stacked on top of each other, are plotted against the total time series of the data matrix. We sometimes see larger magnitudes from the RHALS algorithm, along with other times seeing smaller magnitudes, but there seems to be no general pattern of bias towards solutions of greater or lesser magnitude generated by RHALS.

Finally, we determine if a solution with a different number of factors is more optimal with the RHALS algorithm for this dataset. Figure A6 shows the convergence of weighted residual error of RHALS as the number of factors increases, with the random initialization from the $13^{th}$ trial. Error dramatically decreases as the factors increase from one to six, while barely improving for solutions with a larger number of factors. Therefore, a six factor solution is justified by the RHALS algorithm.

## 4.4 Testing Rotations

For rotations of entire factors, we first tested the rotational algorithm detailed in Paatero and Hopke (2009) to determine whether or not it was applicable to RHALS. Unfortunately the pulling equations used in the multilinear engine (ME) do not seem to also work for RHALS.

Implementing the approach laid out in Section 2.2 to some of the solutions, we find that this method can potentially find better factorizations in the solution space. Ranging the values of $a$ and $b$ in Eq. (28) and Eq. (29) between 0 and 500 (which is of similar magnitude of the optimal amount of L2 regularization), we test the rotational method on the RHALS solutions from the $11^{th}$, $13^{th}$ and $17^{th}$ trials. The graphs are plotted in which a positive pulling value corresponds to a "pull up" of $\mathbf{W}$ and a "pull down" of $\mathbf{H}$, and vice versa for negative pulling values. The values of $a$ and $b$ are the magnitude of the pulling parameter. We present the rotation of the $13^{th}$ solution in Figures 14 and 15, with the rotations of the solutions from other trials in the Appendix.

As can be seen by analyzing the rotation of the solution from the three different trials, solutions that have lower weighted residual error and are closer to the target solution can be found through this simple pulling method, regardless of the direction of the pull. Interestingly, weighted residual error was able to be decreased to under 7,000 when rotating the solution from the 13th trial towards larger values of $\mathbf{W}$ with a pulling parameter of 400. Note that the matrix norms do not always increase with larger positive values of the pulling parameter (and vice versa). This is most clearly seen in Figure A11 in the Appendix with

the rotated factors from the 11th trial. As the factors are pulled in one direction, the matrix norms may respond in the opposite direction.

   The magnitude of change in the similarity to the PMF2 factors is small for all solutions, and rotations may only be worthwhile to look at when no good solution exists from the onset. The benefit of the rotations is more apparent when rotating poor solutions.

**5 Conclusions**

As the size of datasets has grown, computational costs have become increasingly expensive for traditional PMF algorithms. Thus, randomized and hierarchical algorithms are attractive alternative methods. Specifically, the RHALS algorithm was shown to provide a reduction in run time compared to the multiplicative update algorithm, as well as the deterministic HALS algorithm. Furthermore, we proposed a novel approach to handling uncertainties in a weighted factorization problem. While this

approach, coupled with randomization, slightly reduced the accuracy of the algorithm, it dramatically decreased the computational cost. Ultimately, we showed that our weighted RHALS algorithm was able to almost completely recreate the factors in both a formed test matrix and a real dataset, and is a useful tool to finding nonnegative factors of large datasets, particularly in the context of real-time atmospheric mass spectrometry.

*Code and data availability.* The source code for the algorithms detailed in this paper are given at https://zenodo.org/badge/latestdoi/537111580.

The datasets used for this analysis are given at https://doi.org/10.25810/0829-t279.

**Appendix A: ALS Derivation**

The cost function $Q$ is defined as

$$Q = ||(\mathbf{A} - \mathbf{W}\mathbf{H}) \oslash \mathbf{\Sigma}||_F^2 + \sum_{i=1}^{m} \alpha ||\mathbf{W}_{(i,:)}||_1 + \sum_{j=1}^{n} \beta ||\mathbf{H}_{(:,j)}||_1 + \gamma ||\mathbf{W}||_F^2 + \delta ||\mathbf{H}||_F^2 \tag{A1}$$

where $\alpha, \beta, \gamma$, and $\delta$ are regularization parameters. As with the derivation of HALS, the elementwise division of $\mathbf{\Sigma}$ is eliminated

by considering a row or column of the residual at a time. Thus we find

$$0 \leftarrow \frac{\partial Q_i}{\partial \mathbf{W}_{(i,:)}} = \frac{\partial}{\partial \mathbf{W}_{(i,:)}} (||(\mathbf{A}_{(i,:)} - \mathbf{W}_{(i,:)}\mathbf{H})\mathbf{\Sigma}_i^{-1}||_F^2 + \alpha ||\mathbf{W}_{(i,:)}||_1 + \sum_{j=1}^{n} \beta ||\mathbf{H}_{(:,j)}||_1 + \gamma ||\mathbf{W}_{(i,:)}||_2^2 + \delta ||\mathbf{H}||_F^2) \tag{A2}$$

$$0 \leftarrow \frac{\partial Q_j}{\partial \mathbf{H}_{(:,j)}} = \frac{\partial}{\partial \mathbf{H}_{(:,j)}}(||\boldsymbol{\Sigma}_j^{-1}(\mathbf{A}_{(:,j)} - \mathbf{W}\mathbf{H}_{(:,j)})||_F^2 + \sum_{i=1}^{m}\alpha||\mathbf{W}_{(i,:)}||_1 + \beta||\mathbf{H}_{(:,j)}||_1 + \gamma||\mathbf{W}||_F^2 + \delta||\mathbf{H}_{(:,j)}||_2^2) \quad \text{(A3)}$$

where $\boldsymbol{\Sigma}_i$ is a diagonal $n \times n$ matrix with the diagonal values equal to the $i^{\text{th}}$ row of $\boldsymbol{\Sigma}$, and $\boldsymbol{\Sigma}_j$ is a diagonal $m \times m$ matrix with the diagonal values equal to the $j^{\text{th}}$ column of $\boldsymbol{\Sigma}$. Using the fact that $||\mathbf{X}||_F^2 = \text{Tr}(\mathbf{X}^T\mathbf{X})$, with Tr being the trace of the matrix, A2 and A3 can be rewritten as

$$0 = \frac{\partial}{\partial \mathbf{W}_i}(\text{Tr}(\boldsymbol{\Sigma}_i^{-1}\mathbf{A}_i^T\mathbf{A}_i\boldsymbol{\Sigma}_i^{-1} - 2\boldsymbol{\Sigma}_i^{-1}\mathbf{A}_i^T\mathbf{W}_i\mathbf{H}\boldsymbol{\Sigma}_i^{-1} + \boldsymbol{\Sigma}_i^{-1}\mathbf{H}^T\mathbf{W}_i^T\mathbf{W}_i\mathbf{H}\boldsymbol{\Sigma}_i^{-1})$$
$$+ \alpha||\mathbf{W}_{(i,:)}||_1 + \sum_{j=1}^{n}\beta||\mathbf{H}_{(:,j)}||_1 + \gamma||\mathbf{W}_{(i,:)}||_2^2 + \delta||\mathbf{H}||_F^2) \quad \text{(A4)}$$

$$0 = \frac{\partial}{\partial \mathbf{H}_j}(\text{Tr}(\mathbf{A}_j^T\boldsymbol{\Sigma}_j^{-1}\boldsymbol{\Sigma}_j^{-1}\mathbf{A}_j - 2\mathbf{H}_j^T\mathbf{W}^T\boldsymbol{\Sigma}_j^{-1}\boldsymbol{\Sigma}_j^{-1}\mathbf{A}_j + \mathbf{H}_j^T\mathbf{W}^T\boldsymbol{\Sigma}_j^{-1}\boldsymbol{\Sigma}_j^{-1}\mathbf{W}\mathbf{H}_j)$$
$$+ \sum_{i=1}^{m}\alpha||\mathbf{W}_{(i,:)}||_1 + \beta||\mathbf{H}_{(:,j)}||_1 + \gamma||\mathbf{W}||_F^2 + \delta||\mathbf{H}_{(:,j)}||_2^2) \quad \text{(A5)}$$

where $\mathbf{A}_{(i,:)} = \mathbf{A}_i$, $\mathbf{A}_{(:,j)} = \mathbf{A}_j$, $\mathbf{W}_{(i,:)} = \mathbf{W}_i$, and $\mathbf{H}_{(:,j)} = \mathbf{H}_j$. Using vector derivative rules and the fact that $\text{Tr}(\mathbf{ABC}) = \text{Tr}(\mathbf{CAB}) = \text{Tr}(\mathbf{BCA})$ when $\mathbf{ABC}$ is square, the following gradients are found.

$$0 \leftarrow -2\mathbf{A}_i\boldsymbol{\Sigma}_i^{-1}\boldsymbol{\Sigma}_i^{-1}\mathbf{H}^T + 2\mathbf{W}_i\mathbf{H}\boldsymbol{\Sigma}_i^{-1}\boldsymbol{\Sigma}_i^{-1}\mathbf{H}^T + \alpha\mathbf{1}_{1\times k} + 2\gamma\mathbf{W}_i \quad \text{(A6)}$$

$$0 \leftarrow -2\mathbf{W}^T\boldsymbol{\Sigma}_j^{-1}\boldsymbol{\Sigma}_j^{-1}\mathbf{A}_j + 2\mathbf{W}^T\boldsymbol{\Sigma}_j^{-1}\boldsymbol{\Sigma}_j^{-1}\mathbf{W}\mathbf{H}_j + \beta\mathbf{1}_{k\times 1} + 2\delta\mathbf{H}_j \quad \text{(A7)}$$

In A6 and A7, $\mathbf{1}$ is a vector of ones. Finally, the following update rules are found.

$$\mathbf{W}_i \leftarrow [(\mathbf{A}_i\boldsymbol{\Sigma}_i^{-1}\boldsymbol{\Sigma}_i^{-1}\mathbf{H}^T - \frac{\alpha}{2}\mathbf{1}_{1\times k})(\mathbf{H}\boldsymbol{\Sigma}_i^{-1}\boldsymbol{\Sigma}_i^{-1}\mathbf{H}^T + \gamma\mathbf{I}_{k\times k})^{-1}]_+ \quad \text{(A8)}$$

$$\mathbf{H}_j \leftarrow [(\mathbf{W}^T\boldsymbol{\Sigma}_j^{-1}\boldsymbol{\Sigma}_j^{-1}\mathbf{W} + \delta\mathbf{I}_{k\times k})^{-1}(\mathbf{W}^T\boldsymbol{\Sigma}_j^{-1}\boldsymbol{\Sigma}_j^{-1}\mathbf{A}_j - \frac{\beta}{2}\mathbf{1}_{k\times 1})]_+ \quad \text{(A9)}$$

where $\mathbf{I}$ is the identity matrix.

## Appendix B: Mathematical Equivalence of Pseudoinverse Update and Ordinary Least Squares

For simplicity, we label $(\widetilde{\mathbf{W}}\widetilde{\mathbf{H}}) \odot \boldsymbol{\Sigma}$ as $\tilde{\mathbf{y}}$. Let's consider the ordinary least squares update for $\mathbf{W}$:

$$\mathbf{W} = \tilde{\mathbf{y}}\mathbf{H}^T(\mathbf{H}\mathbf{H}^T)^{-1}$$

$\mathbf{H}$ is a $k \times n$ matrix that is assumed to be of full row rank, and thus can be decomposed into the exact SVD factorization $\mathbf{H} = \mathbf{U}\mathbf{S}\mathbf{V}^T$, where $\mathbf{U}$ is a $k \times k$ orthogonal matrix, $\mathbf{S}$ is a $k \times k$ diagonal matrix, and $\mathbf{V}$ is a $n \times k$ matrix with orthonormal columns. The pseudoinverse of $\mathbf{H}$ is defined as $\mathbf{V}\mathbf{S}^{-1}\mathbf{U}^T$. $\mathbf{V}^T\mathbf{V} = \mathbf{I}$, and $\mathbf{U}^T = \mathbf{U}^{-1}$. The update for $\mathbf{W}$ can be rewritten as

$$
\begin{aligned}
\mathbf{W} &= \tilde{\mathbf{y}}\mathbf{V}\mathbf{S}\mathbf{U}^T(\mathbf{U}\mathbf{S}\mathbf{I}\mathbf{S}\mathbf{U}^T)^{-1} \\
&= \tilde{\mathbf{y}}\mathbf{V}\mathbf{S}\mathbf{U}^T(\mathbf{U}\mathbf{S}^2\mathbf{U}^T)^{-1} \\
&= \tilde{\mathbf{y}}\mathbf{V}\mathbf{S}\mathbf{U}^T(\mathbf{U}\mathbf{S}^{-2}\mathbf{U}^T) \\
&= \tilde{\mathbf{y}}\mathbf{V}\mathbf{S}\mathbf{I}\mathbf{S}^{-2}\mathbf{U}^T \\
&= \tilde{\mathbf{y}}(\mathbf{V}\mathbf{S}^{-1}\mathbf{U}^T) \\
&= \tilde{\mathbf{y}}\mathbf{H}^{\dagger}
\end{aligned}
\tag{B1}
$$

which is equal to the pseudoinverse update described in Section 2.3. A similar argument can be made for the update for $\mathbf{H}$.

*Author contributions.*  B.C.S. wrote the manuscript, conducted all of the numerical experiments, and derived the equations. D.K.H. advised the numerical and theoretical approach. D.K.H., M.C., H.S., J.L.J, and S.Y. reviewed and edited the manuscript. M.C. and H.S. provided the real-time atmospheric mass spectrometry data set and reference PMF2 solution.

*Competing interests.*  The authors declare that they have no conflict of interest.

*Acknowledgements.*  B.C.S. acknowledges funding from CU SPUR (Summer Program for Undergraduate Research), D.K.H. acknowledges
support from NASA 80NSSC20K0214.

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

**Table 1.** Comparison of the number of operations per step for each algorithm used. We also track the number of elementwise multiplications and divisions (implemented in MATLAB as .* and ./) in the third column

| Comparison of Cost/Step for Different Algorithms | | |
|---|---|---|
| Algorithm | Number of Operations Using Matrix Multiplication | Number of Elementwise Multiplications/Divisions |
| ALS (IW) | $\mathcal{O}(mnk^2)$ | $2mnk$ |
| ALS (EW) | $\mathcal{O}(mnk)$ | $0$ |
| MU (IW) | $\mathcal{O}(mnk)$ | $4mn + 4(m+n)k$ |
| MU (EW) | $\mathcal{O}(mnk)$ | $4(m+n)k$ |
| MU (Randomized) | $\mathcal{O}((k+l)nk)$ | $4((k+l)+n)k$ |
| HALS (IW) | $\mathcal{O}(mnk^2)$ | $2mnk$ |
| HALS (EW) | $\mathcal{O}(mnk)$ | $0$ |
| RHALS | $\mathcal{O}((k+l)nk)$ | $0$ |
| Post-Processing | $\mathcal{O}(mnk)$ | $0$ |

**Table 2.** Average statistics of different algorithms over five different SVD initializations, tolerance=$10^{-4}$ (for MU, tolerance is $10^{-5}$ and a random initialization), L1 Regularization=1, L2 Regularization=50

| Comparison of Algorithms | | | |
|---|---|---|---|
| Algorithm | Total Time (s) | Steps | Weighted Error |
| ALS (IW) | 317.08 | 42.2 | $6.48 \times 10^3$ |
| ALS (EW) | 12.21 | 29.0 | $7.35 \times 10^3$ |
| MU (IW) | 19.31 | 76.4 | $6.52 \times 10^3$ |
| MU (EW) | 4.06 | 51.2 | $7.11 \times 10^3$ |
| MU (Randomized) | 4.02 | 53.6 | $7.76 \times 10^3$ |
| HALS (IW) | 33.87 | 19.4 | $6.45 \times 10^3$ |
| HALS (EW) | 1.01 | 31.2 | $7.08 \times 10^3$ |
| RHALS (EW) | 0.50 | 36.6 | $7.15 \times 10^3$ |

**Table 3.** Average statistics of external weighting (EW) versus expectation maximization (EM) algorithms over 20 trials. Internally weighted (IW) HALS is provided as a reference. Ranges of the values of the algorithms run using different variations of expectation maximization (EM) steps are presented. The correlation of the columns of $\mathbf{W}$ and similarity of the rows of $\mathbf{H}$ to the PMF2 solution are also listed.

| Comparison of Algorithms | | | | | |
|---|---|---|---|---|---|
| Algorithm | Total Time (s) | Steps | Weighted Error | Correlation of $\mathbf{W}$ | Similarity of $\mathbf{H}$ |
| HALS (IW) | 40.70 | 19.65 | $6.46 \times 10^3$ | 0.8756 | 0.9134 |
| HALS (EW) | 1.07 | 31.80 | $7.09 \times 10^3$ | 0.8414 | 0.8828 |
| RHALS (EW) | 0.56 | 38.35 | $7.27 \times 10^3$ | 0.8475 | 0.9034 |
| HALS (EM) | 0.51-2.93 | 13.50-83.90 | $6.71\text{-}7.22 \times 10^3$ | 0.7759-0.8306 | 0.8434-0.8861 |
| RHALS (EM) | 0.33-1.19 | 20.30-61.05 | $7.11\text{-}7.54 \times 10^3$ | 0.7930-0.8221 | 0.8301-0.8694 |

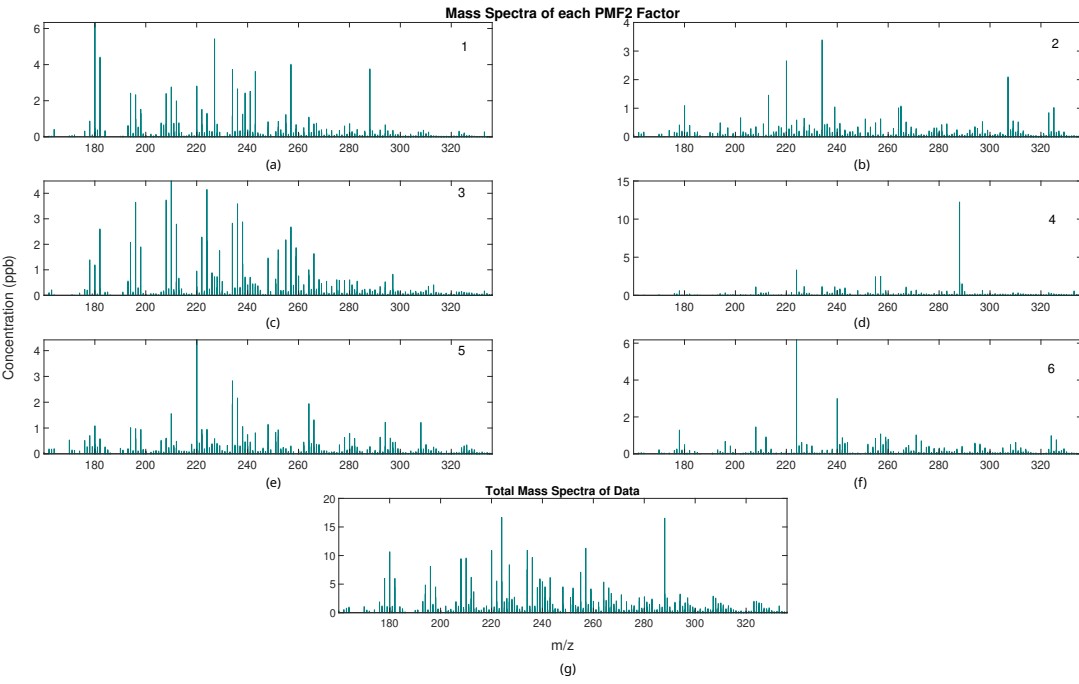

**Figure 1.** (a-f) mass spectra profiles of the six PMF2 factors labeled by order; (g) total mass spectra concentration of the data.

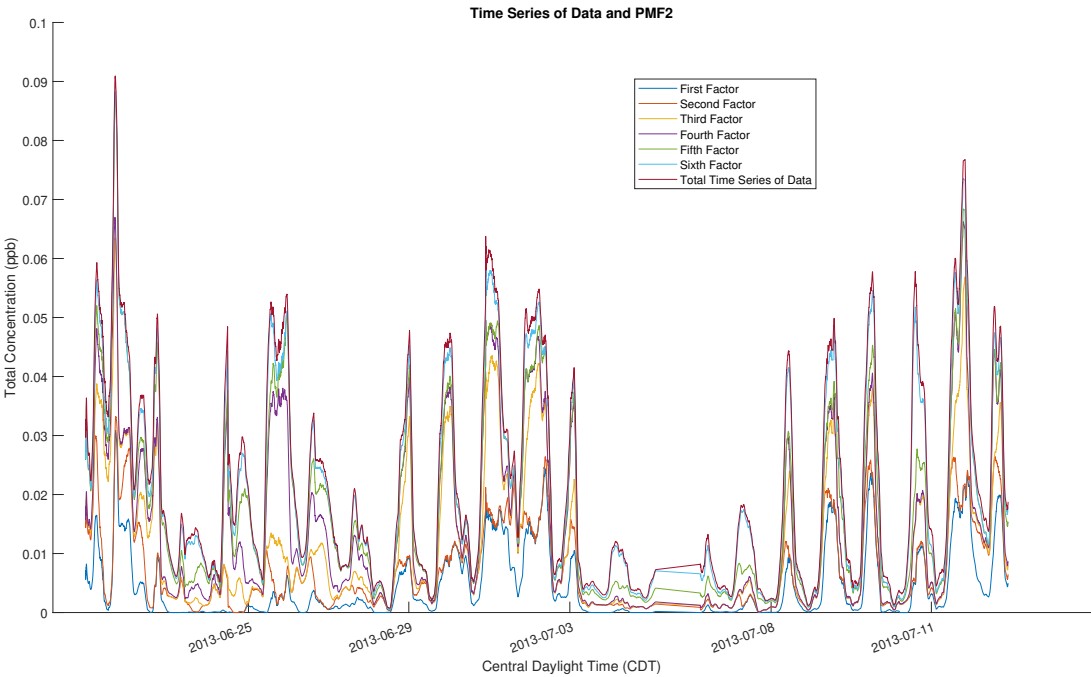

**Figure 2.** Time series profiles of the six PMF2 factors, overlaid with the total time series concentration of the data. A rolling average is used with values representing average concentrations over the previous two hours. The individual time series are stacked on top of each other in order to compare the total PMF2 time series to the overall time series of the data.

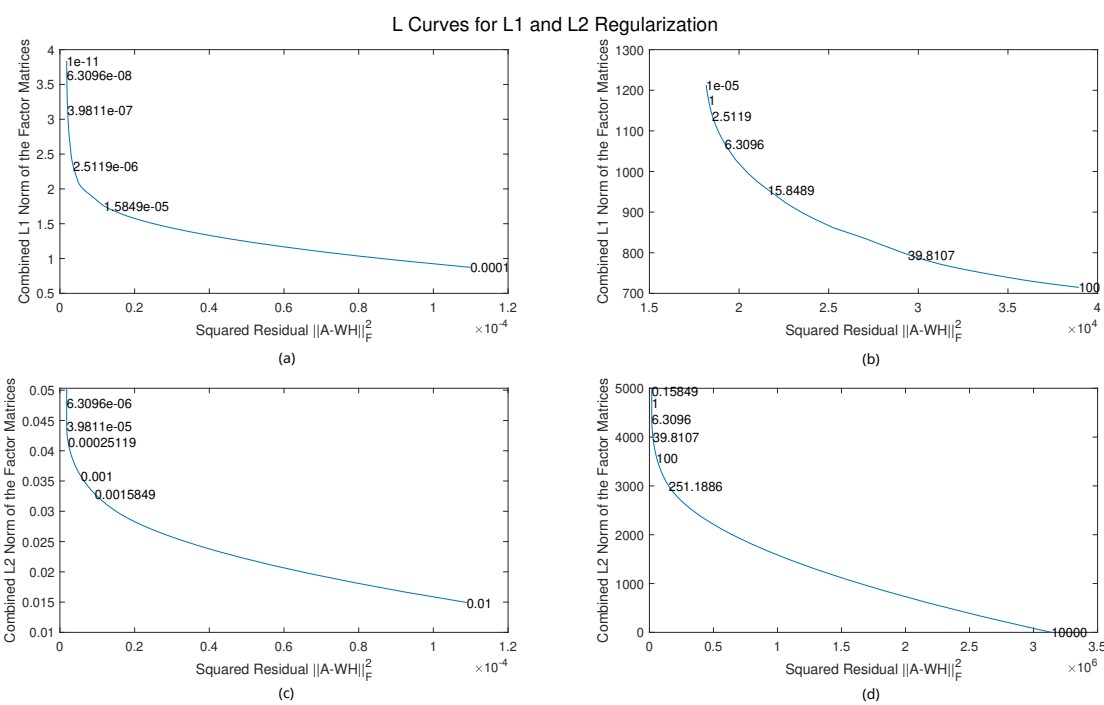

**Figure 3.** L Curves for L1 and L2 Regularization, applied to data with different average values. (a) L1 regularization for data with a magnitude around $10^{-5}$; (b) L1 regularization for data with a magnitude around one; (c) L2 regularization for data with a magnitude around $10^{-5}$; (d) L2 regularization for data with a magnitude around one.

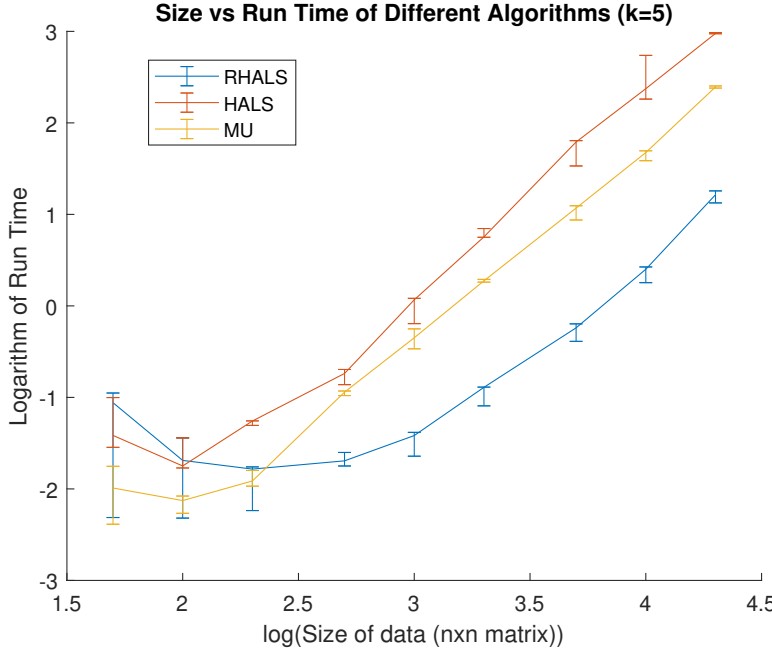

**Figure 4.** Size of data matrix vs run time (in seconds) in RHALS algorithm, performed with three different random seeds (containing the absolute value of random normal variables). The x-axis shows the base 10 log of $n$, and the y-axis plots the base 10 log of run time. The median run time is plotted, with the error bars plotting the maximum and minimum run times. Note that run time initially decreases due to a coincidental reduction in the number of steps to convergence, but problems of environmental interest are generally located on the right of the graph or beyond its right edge.

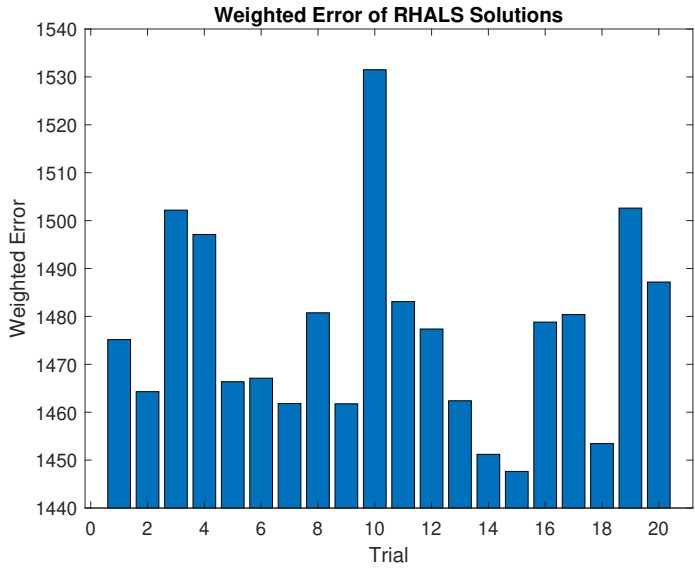

**Figure 5.** RHALS error over 20 trials. Mean: 1476.60, Standard Deviation: 20.47.

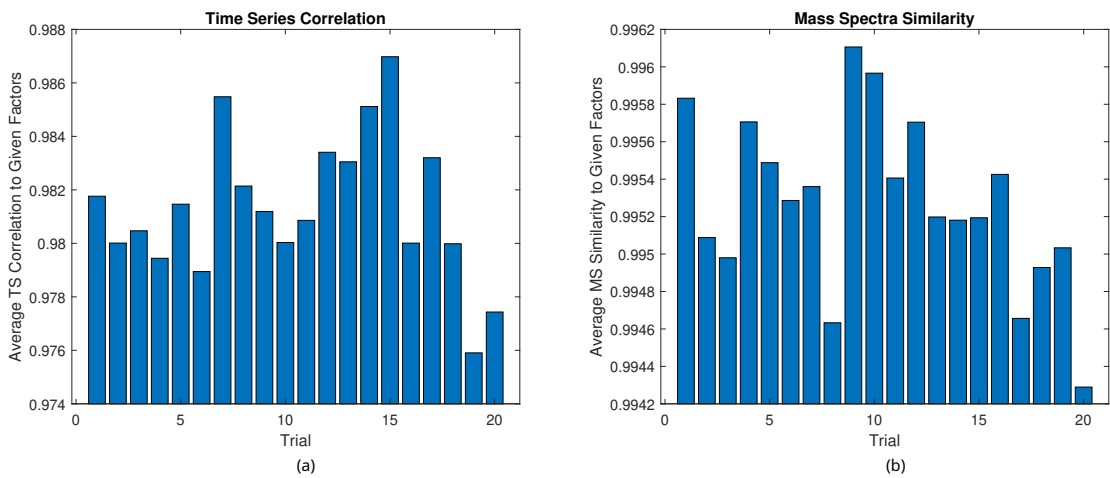

**Figure 6.** (a) average time series similarity and (b) average mass spectra similarity to formed factors in the small test case.

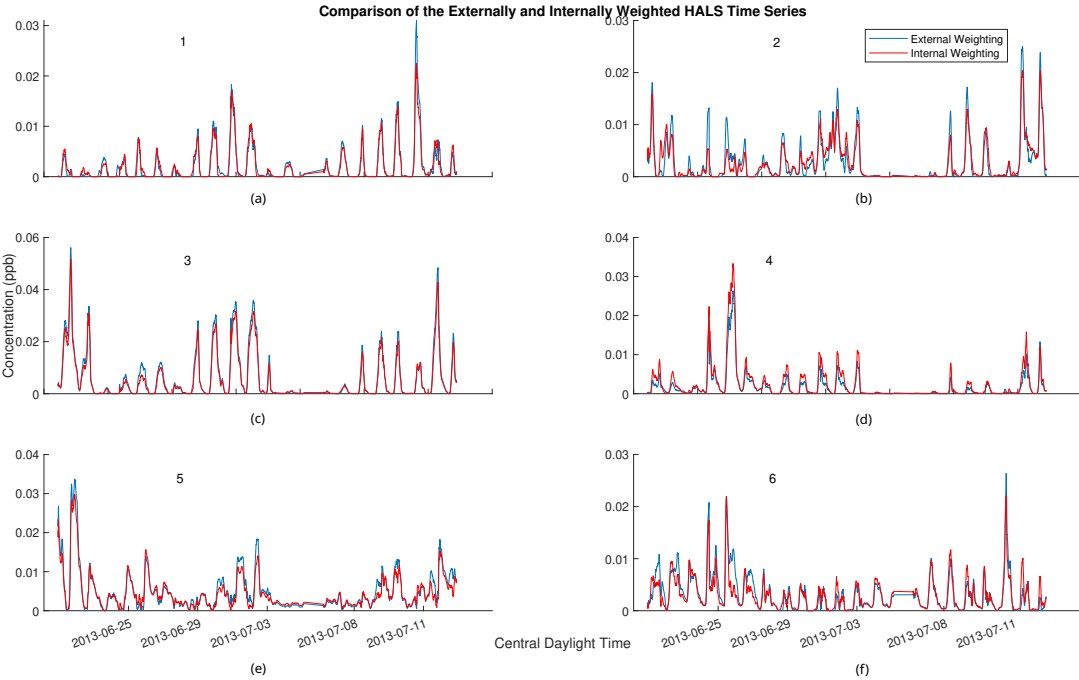

**Figure 7.** Comparison of two hour rolling average of time series for externally weighted and internally weighted HALS factors. Externally weighted error: $7.1321 \times 10^3$. Internally weighted error: $6.4841 \times 10^3$.

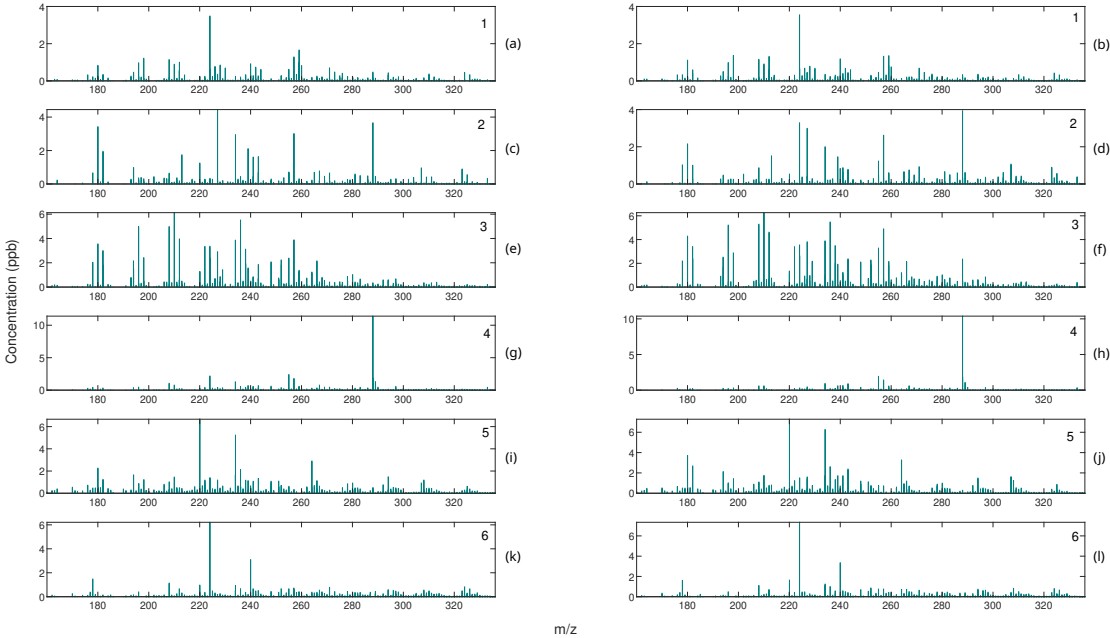

**Figure 8.** Comparison of mass spectra for internally weighted (on the left) and externally weighted (on the right) HALS factors.

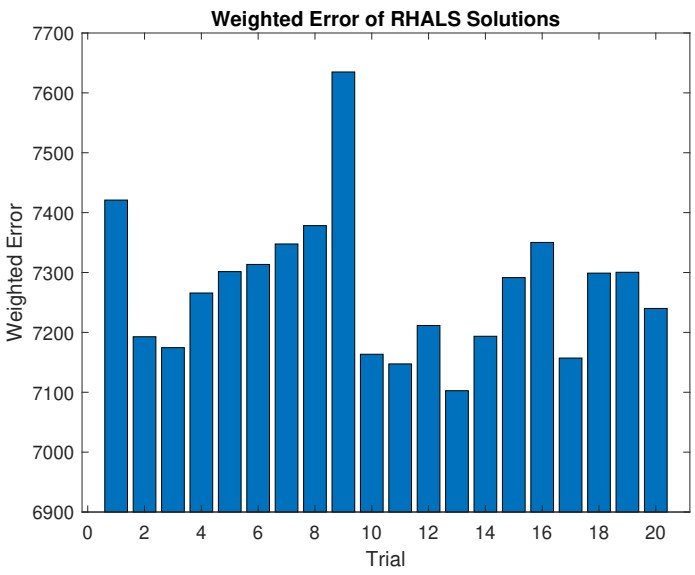

**Figure 9.** RHALS error over 20 trials. Mean Error: $7.2743 \times 10^3$, Standard Deviation of Error: 120.8. Mean Time: 0.5163, Standard Deviation of Time: 0.0292. Mean Steps: 38.45 Standard Deviation of Steps: 6.88.

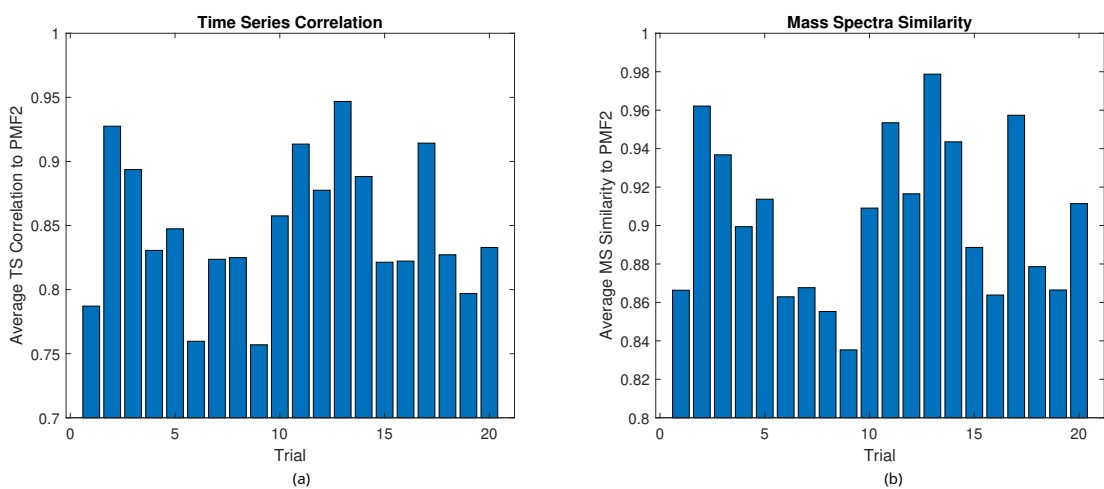

**Figure 10.** (a) average time series and (b) average mass spectra similarities between RHALS and PMF2.

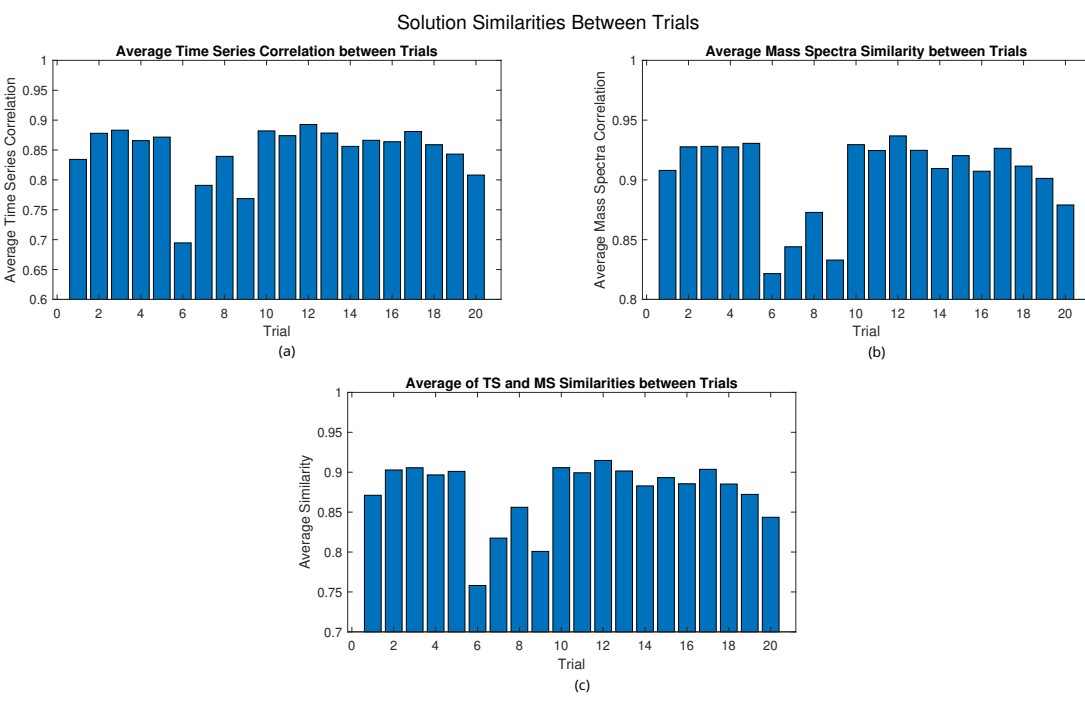

**Figure 11.** (a) average time series correlation (b) average mass spectra similarity and (c) average of time series and mass spectra similarities between the solutions of the different trials for RHALS.

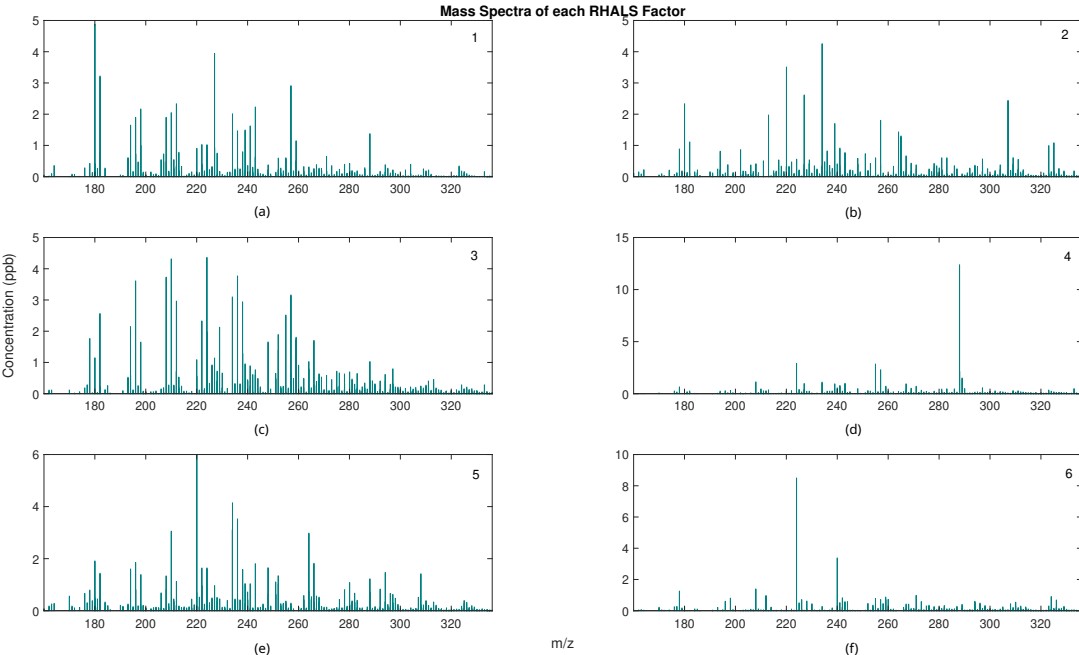

**Figure 12.** RHALS mass spectra series factors; (a) first factor similarity with PMF2 = 0.9631 (b) second factor similarity with PMF2 = 0.9582 (c) third factor similarity with PMF2 = 0.9912; (d) fourth factor similarity with PMF2 = 0.9970 (e) fifth factor similarity with PMF2 = 0.9795 (f) sixth factor similarity with PMF2 = 0.9835.

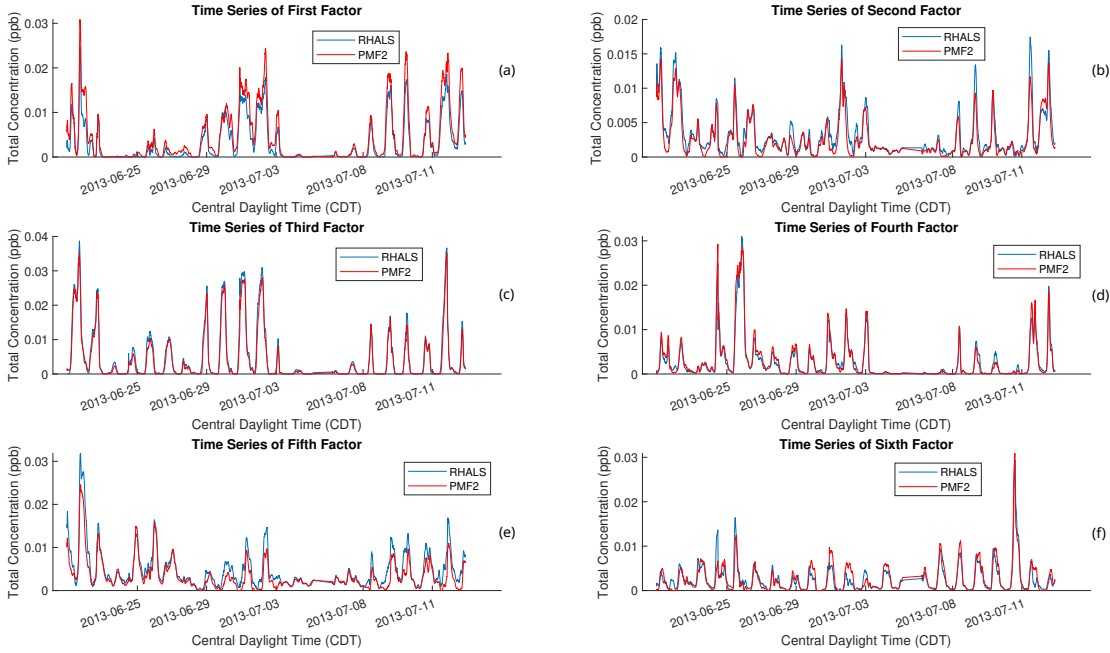

**Figure 13.** Two hour rolling average of RHALS time series factors overlaid with PMF2 factors.

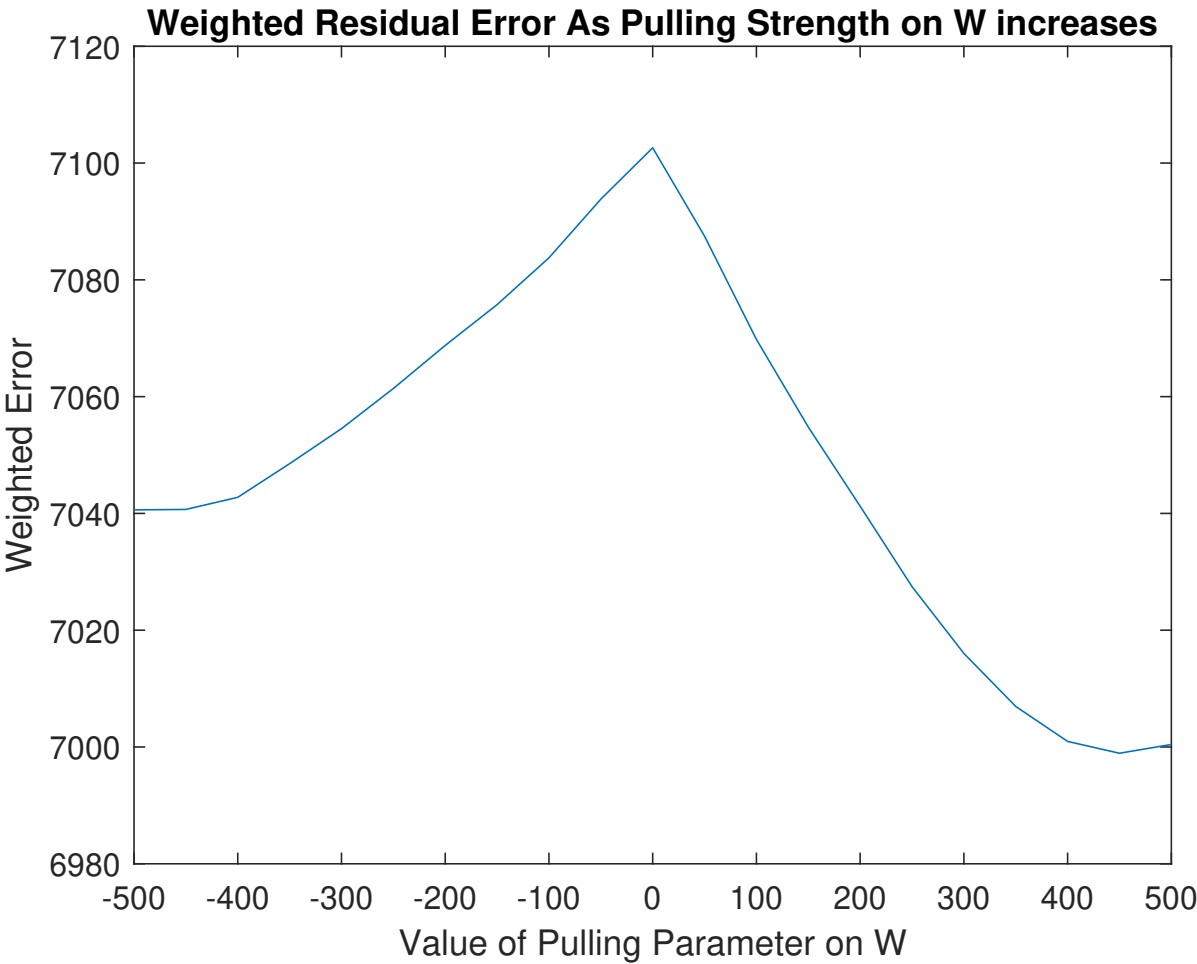

**Figure 14.** Weighted residual error of rotated solutions from the $13^{\text{th}}$ trial of RHALS.

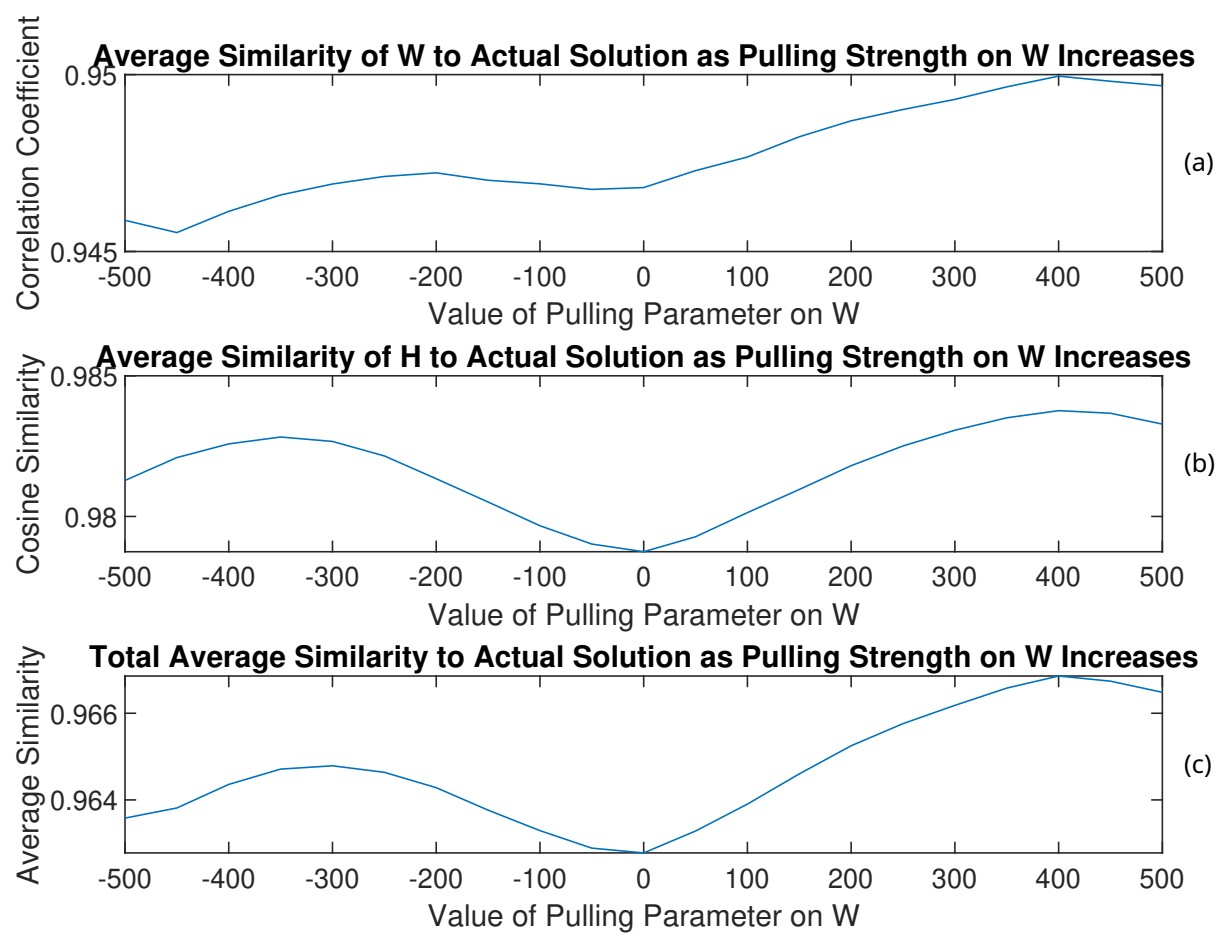

**Figure 15.** (a) **W**, (b) **H**, and (c) total average similarities of the rotated solutions to the PMF2 solution from the $13^{th}$ trial of RHALS. The total similarity refers to an average of the similarity metrics for the time series and mass spectra factors.

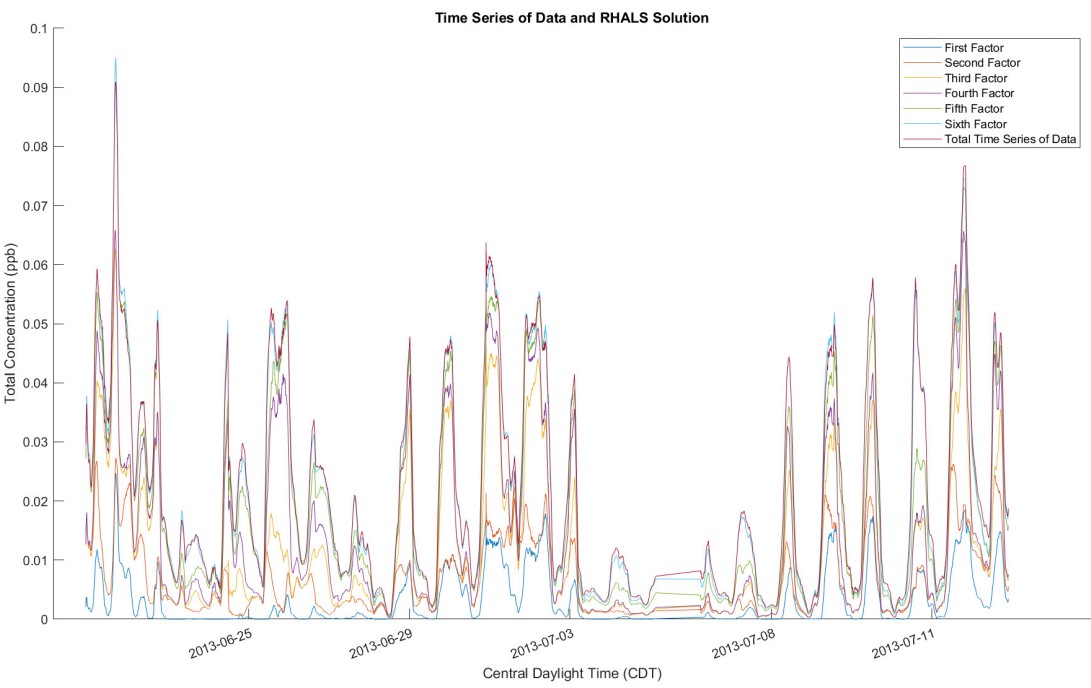

**Figure A1.** RHALS Time Series Factors laid out with total time series from data, plotted as a two hour rolling average.

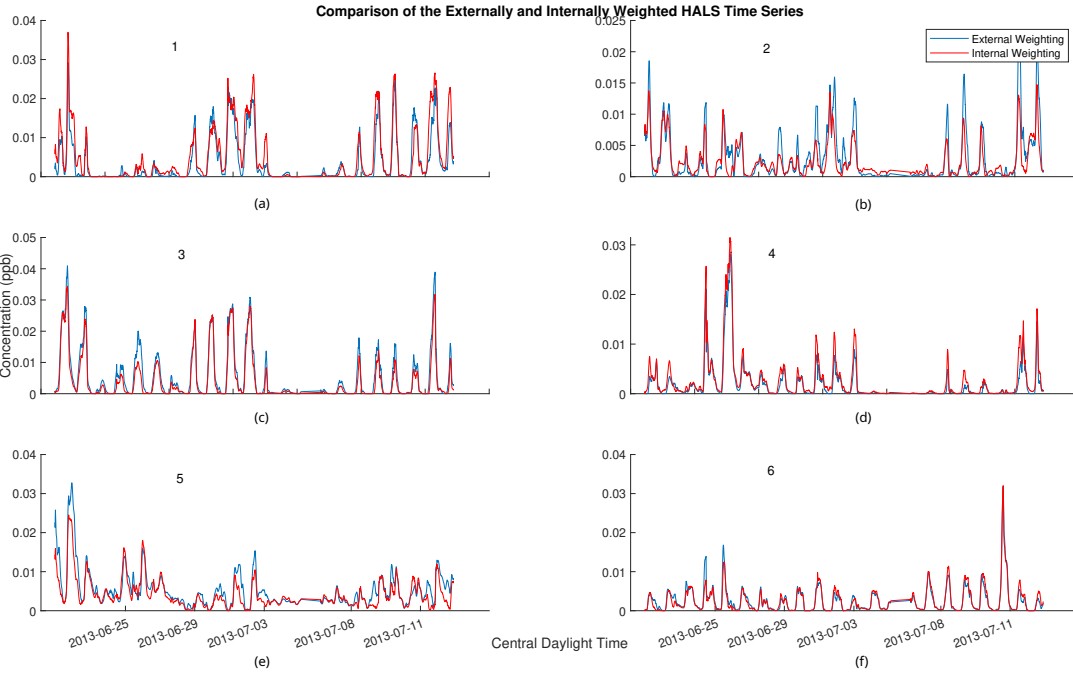

**Figure A2.** Comparison of two hour rolling average of time series for externally weighted and internally weighted ALS factors. Externally weighted error: $7.3666 \times 10^3$. Internally weighted error: $6.4768 \times 10^3$.

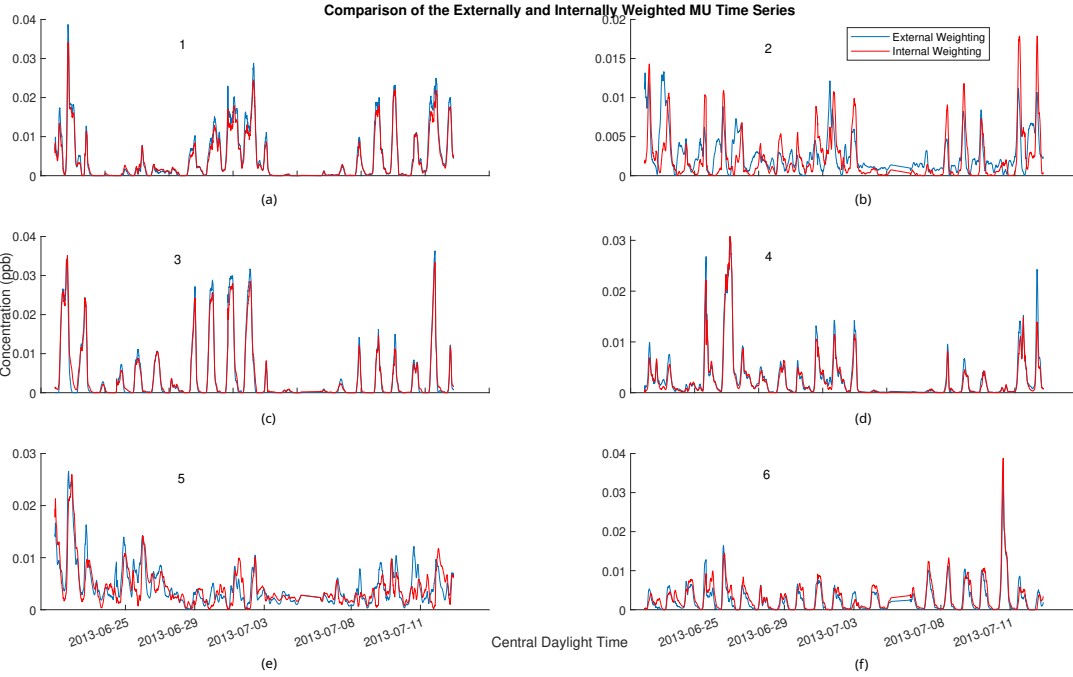

**Figure A3.** Comparison of two hour rolling average of time series for externally weighted and internally weighted MU factors. Externally weighted error: $6.9457 \times 10^3$. Internally weighted error: $6.4292 \times 10^3$.

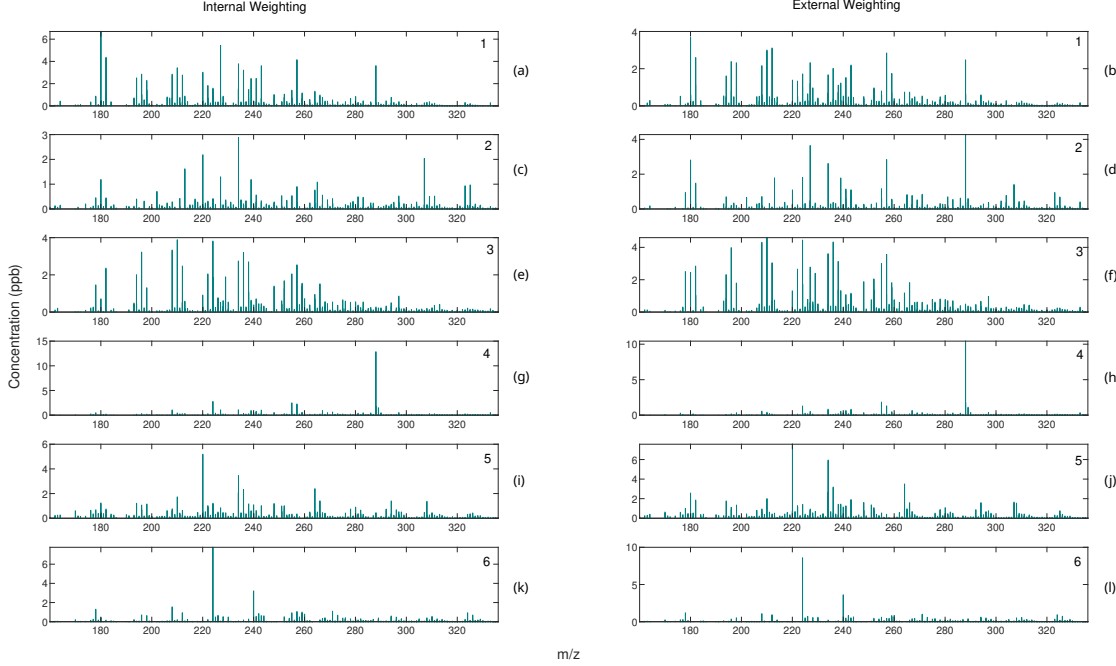

**Figure A4.** Comparison of mass spectra for internally weighted (on the left) and externally weighted (on the right) ALS factors.

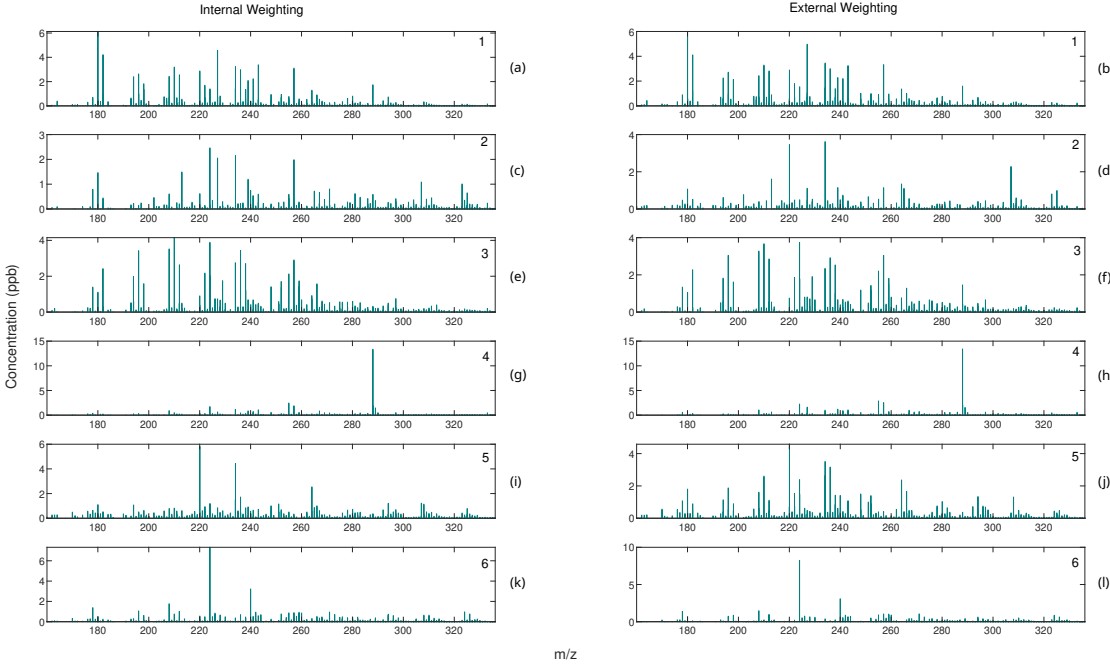

**Figure A5.** Comparison of mass spectra for internally weighted (on the left) and externally weighted (on the right) MU factors.

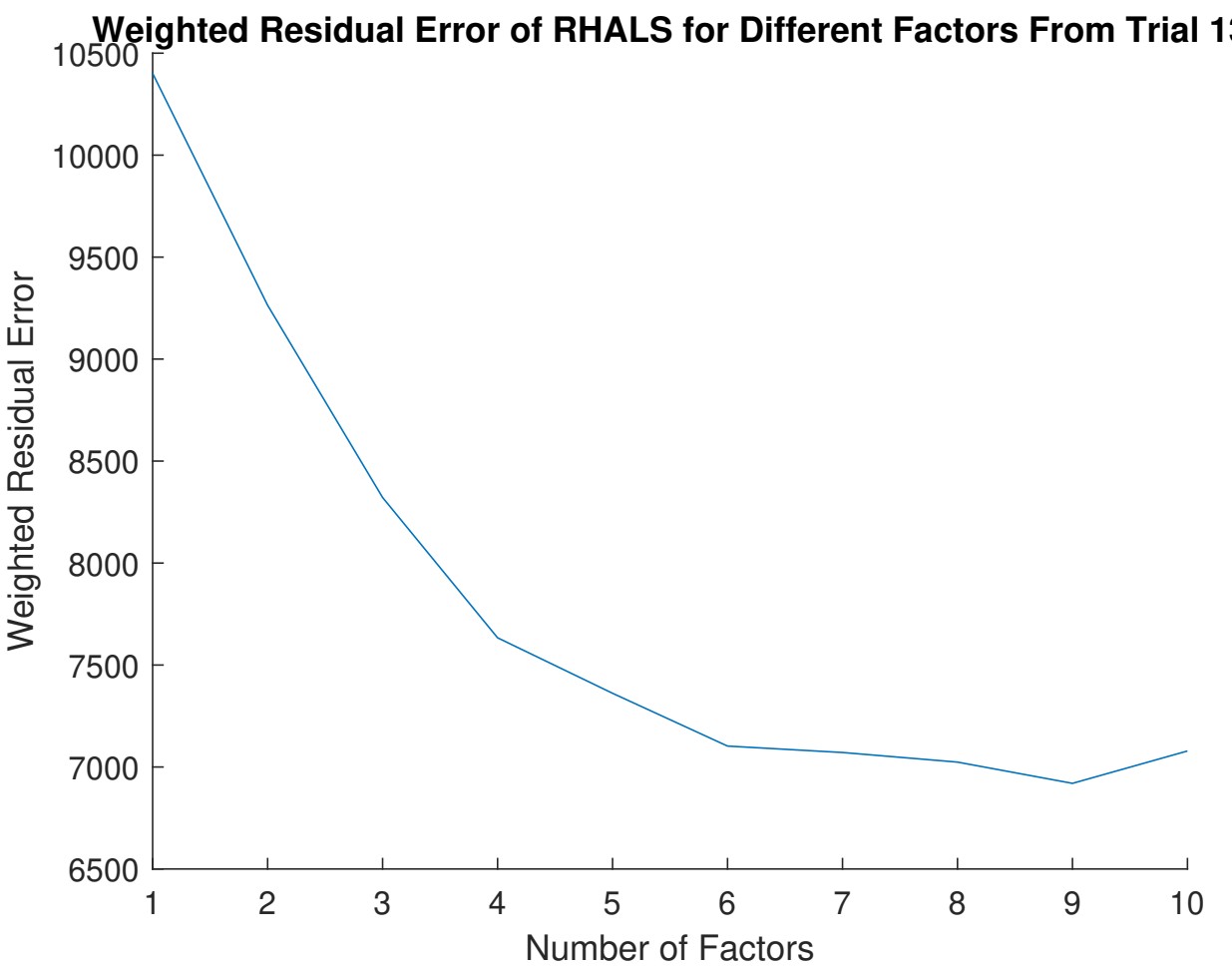

**Figure A6.** Number of factors versus weighted residual error for the RHALS algorithm.

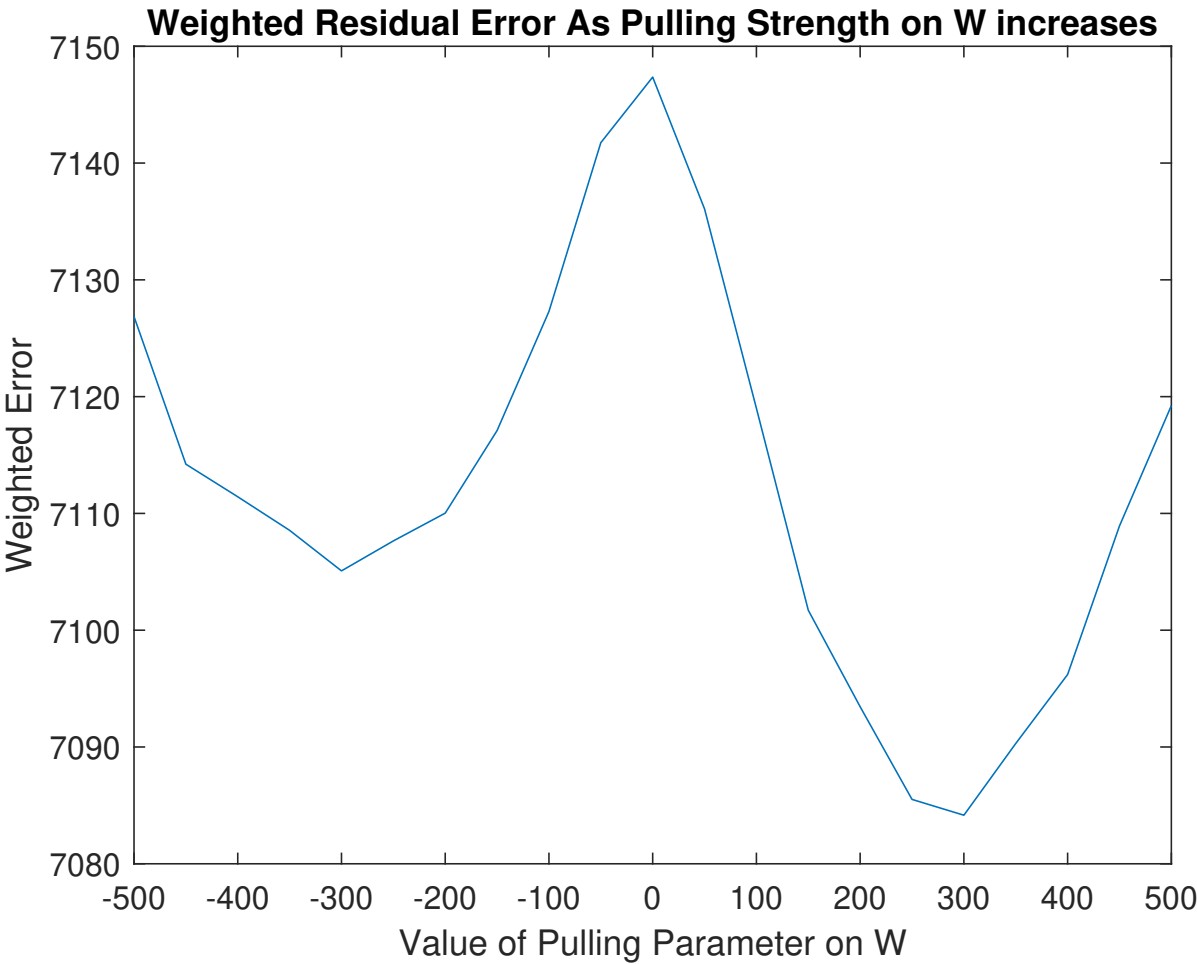

**Figure A7.** Weighted residual error of rotated solutions from the 11[th] trial of RHALS.

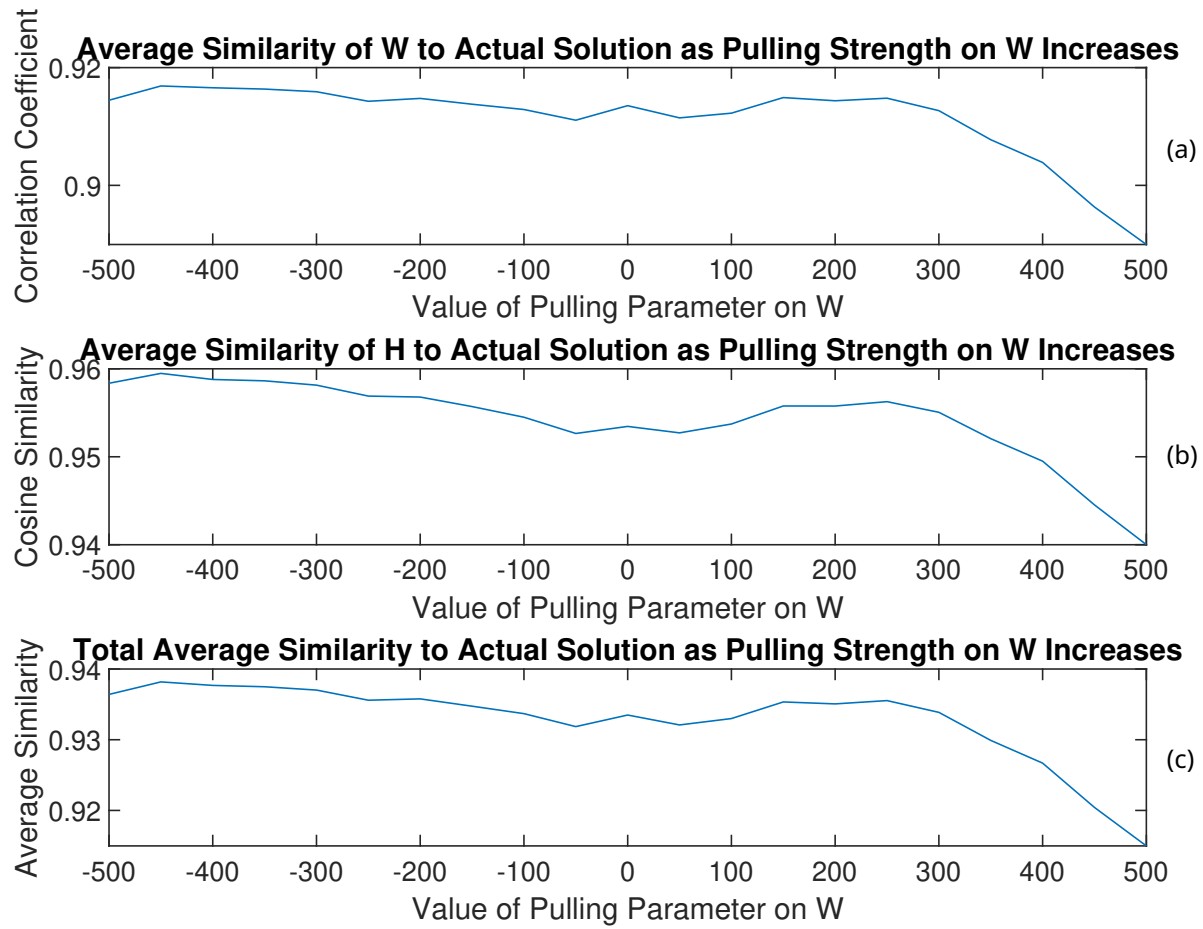

**Figure A8.** (a) **W**, (b) **H**, and (c) total average similarities of the rotated solutions to the PMF2 solution from the 11[th] trial of RHALS.

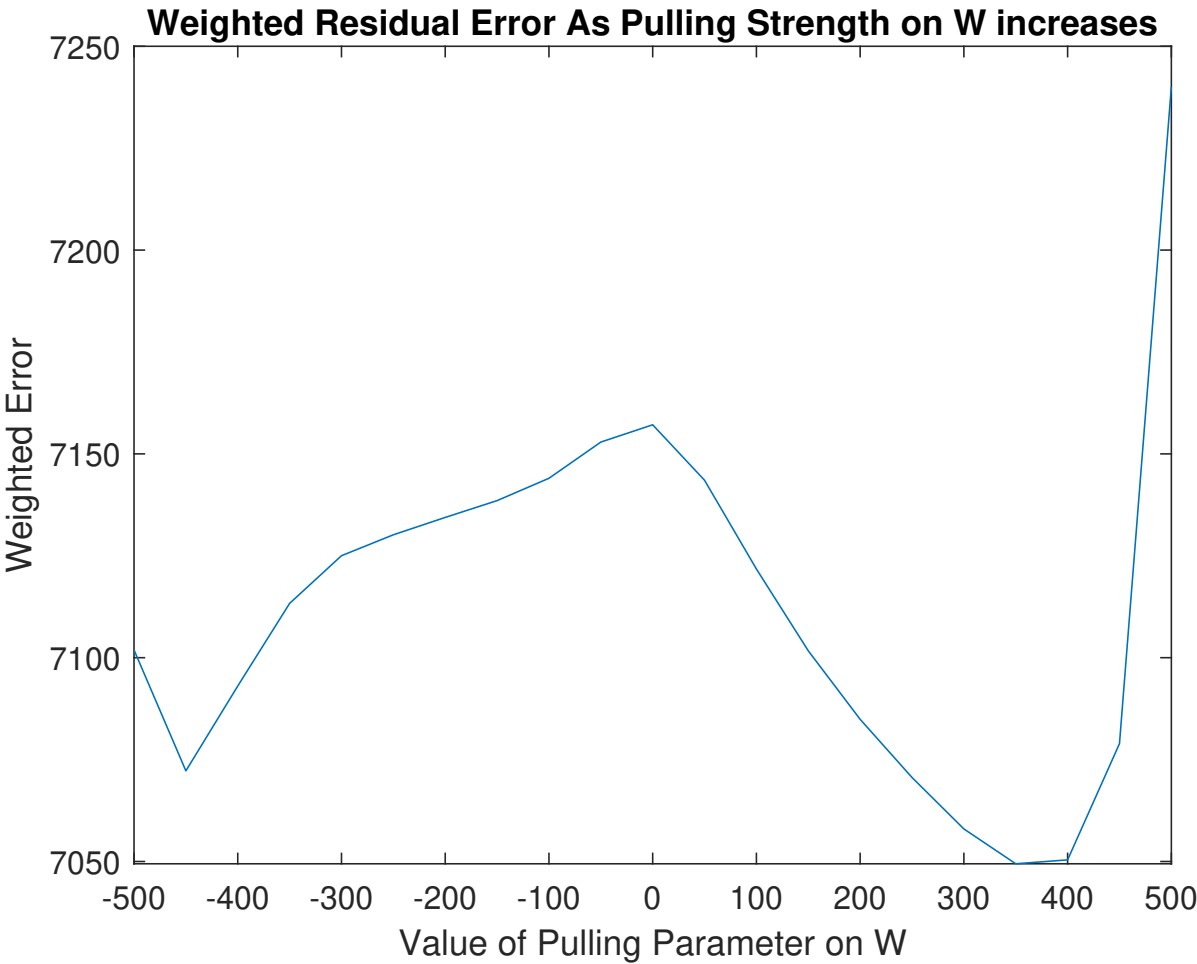

**Figure A9.** Weighted residual error of rotated solutions from the 17$^{\text{th}}$ trial of RHALS.

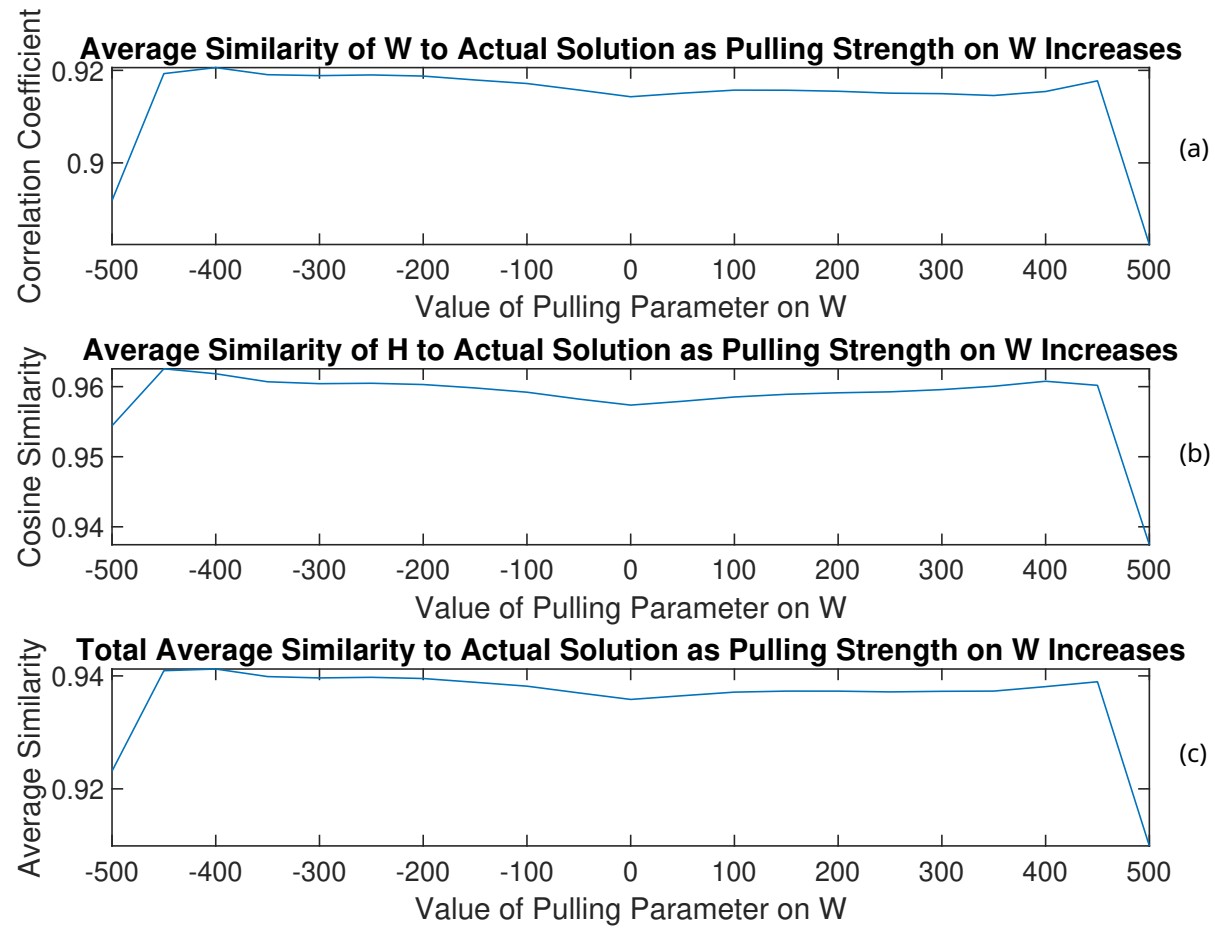

**Figure A10.** (a) **W**, (b) **H**, and (c) total average similarities of the rotated solutions to the PMF2 solution from the 17th trial of RHALS.

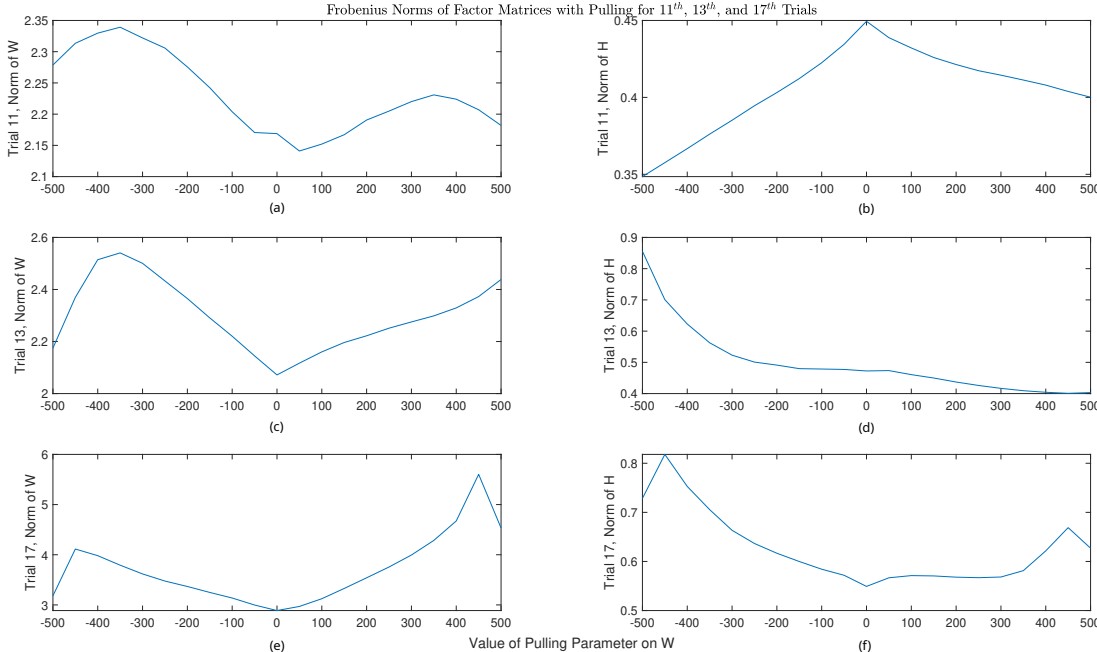

**Figure A11.** Frobenius norms of rotated factor matrices for the (a-b) 11<sup>th</sup>, (c-d) 13<sup>th</sup>, and (e-f) 17<sup>th</sup> trials.