# Peer review of "Positive Matrix Factorization of Large Real-Time Atmospheric Mass Spectrometry Datasets Using Error-Weighted Randomized Hierarchical Alternating Least Squares"

_EGUsphere, 2022_

## Referee Comment (RC2)

**Review of "Positive Matrix Factorization of Large Aerosol Mass Spectrometry Datasets Using Error-Weighted Randomized Hierarchical Alternating Least Squares"**

**Summary**

NMF is a widely recognized technique with many applications in data analytics. For a given data matrix $X$, the weighted NMF computates nonnegative factors $W, H$ that attempt to minimize the weighted Frobenius error $\|S \circ (X - WH)\|_F^2$ where $\circ$ is the Hadamard product and the elements $(S)_{ij}$ of $S$ are functions of the measurement uncertainties (this is not the authors' notation but see below). The main novelty is the design and incorporation of a form of weighting to existing algorithms for NMF (or PMF, as the authors opt to call them) and its application on atmospheric data. The experimental results indicate that the method is faster but sometimes less accurate than other methods. It is also shown that these (larger) errors are not seriously affecting intepretability. The authors also state that "We will not discuss the chemical interpretations of the data, rather just the mathematical results from running the RHALS algorithm." Therefore, the paper is more on the algorithmic and numerical issues than on the method's impact on the selected application.

RHALS and weighted RHALS have been discussed elsewhere in the literature, so the main novelty appears to be in the weighting scheme and the comparisons. As described, in the external weighting scheme, one computes unweighted factors, e.g. using standard RHALS on the scaled data matrix, then applies the weighting on their product and then computes new nonnegative factors. This latter step is accomplished using alternating least squares. So it seems that the original weighted NMF is solved first after scaling, and then a new NMF is computed. In particular, for this second phase, I assume that some NMF is used and not simply ALS, otherwise the factors will come out of mixed sign. Is this mentioned?. In any case, the idea is interesting but somwewhat ad hoc and a little circular and therefore deserves more discussion. See also the detailed comments for "[page 11], line 273".

The externally weighted algorithm turns out to be faster than existing "internally weighted" NMF methods, specifically MU and HALS. The authors also compare with the MU, ALS and HALS algorithms using the proposed external weighting scheme. A link to a MATLAB implementation is also provided. Overall, the scientific advance in the paper is incremental but can be of interest.

Some points of criticism:

1. The key point of the algorithm is that the combination of RHALS with the external weighting scheme is faster than competing schemes. However, to solidly back this claim, it is not sufficient to only provide runtimes, especially from unoptimized MATLAB implementations. For example, the computational costs of several formulas, e.g the updates on p9, are very sensitive to the order of the operations, the parenthesization, etc. MATLAB is very convenient, but does not attempt to optimize matrix expressions. Were these issues taken into account? The

authors also need to show that the "competing" codes have been implemented with runtime minimization in mind. Unfortunately, it is not sufficient to compare performance-oblivious implementations when the objective is to reduce runtimes.

2. The authors should also specify the costs of each method per iteration (preferably in terms of the matrix primitives used).

3. Comparisons are with weighted versions of ALS and MU. These are prone to slow convergence and or even failure to reach a reasonable local optimal solution. ALS, for example (see N. Gillis, p283 of https://doi.org/10.1137/1.9781611976410) "is easy to implement ... and sometimes provides reasonable solutions, typically for sparse matrices; ... However, it comes with no theoretical guarantee and in fact often diverges in practice, especially for dense input matrices; ... Therefore, ALS can be recommended only as a warm-start stage for theoretically better grounded approaches...." See chapter 8 of the aforementioned book and Table 8.2 for a comparison of methods: ALS and MU stand at the "low end" in several respects.

4. In Section 4.1, I am missing an analysis of the contribution of the number of factors to the cost. Indeed, in some cases the discussion is incomplete: For example on p11 it is stated that ".... the algorithm may take longer for a larger number of factors and in certain cases. More work is needed regarding the different convergence issues that may arise." In light of the fact that the external weighting scheme is a key contribution of this paper, the authors need to address these issues in greater detail.

5. It would also be useful to provide information on the cost of each major step and iteration rather than only comparing runtimes to convergence.

6. The paper makes a brief reference to a WNMF method for missing entries (Yahaya et al.) However the method is dismissed because of the cost of EM without further ado. The authors should address a) how they would handle missing entries, b) how does their method compare in view of related papers by the same team, some included in the publically available thesis of F. Yamaha. It is worth noting that the paper "Random Projection Streams for (Weighted) Nonnegative Matrix Factorization," doi: 10.1109/ICASSP39728.2021.9413496." discusses computational complexity, something I am missing in this manuscript as explained earlier.

7. I spotted some (trivial to correct but somewhat sloppy-type) errors. Some examples: Formula (7) and the formula for the quality of fit parameter (line 161) are incorrect. In the former, one must use the absolute values of the terms of the sum and in the latter it is the individual terms that must be squared, not the sum. Line 324 on p13 also has such an error. On p. 19, line 527, it is stated that "H is a k × n matrix that is assumed to be of full column rank". However, assuming that $k < n$, it cannot be full column rank, but at best, full row rank. There might be other errors as well.

**Detailed comments**

[page 1]: Please state whether the matrix is assumed to be strictly positive or nonnegative. If the latter, then why use PMF instead of NMF which is more general and the term used in some of the paper's major citations.

[page 2 and beyond]: The authors state: "In Eq. (1), the division by $\Sigma$ represents elementwise division". I strongly recommend against using fractional notation when dealing with matrices. They should use Hadamard division or Hadamard multipication with a variable that is defined to contain the inverses of the elements of $\Sigma$. In fact, on p265, the authors actually use the (MATLAB infix operator) for elementwise division. This is certainly better than the / symbol, but even better to use the "Hadamard notation".

[page 3]: Better include the section number when you refer to Sections, e.g. Methods (Section 2).

[page 4]: The use of "*" to denote matrix multiplication is inconsistent with the rest of the paper.

[page 4]: A proper and complete discussion on computational optimizations and parallelization would merit much more than the few statements in sections 1.3. I propose that the authors take this into account and eliminate section 1.3.2, incorporating the few comments in some other section.

[page 6]: Please clarify the statement of line 140. How does the incorporation of the weighting via the uncertainty matrix affect the costs?

[page 8]: Formula (10) is missing the opeining (left) parenthesis. Also in line 210 better say "denotes the trace of the matrix" rather than "takes the trace of the matrix". After formula (14) better say "Differentiating with respect..." instead of "Taking the derivative ...".

[page 9]: Careful in the parenthesization of Formula (16). Some parentheses are missing

[page 9]: In formula (20) and elsewhere, aren't the $\Sigma_j$ matrices diagonal? If so, then the transpose symbol is redundant. Also careful in the parenthesization of Formula (20). Some parethesizations can be more efficient than others.

[page 9]: In formula (20) and elsewhere: have you defined the maximum operator $[\cdot]_+$?

[page 10]: Some readers might question the characterization "rotation" for a non-orthogonal matrix. In the linear algebra literature, rotation matrices preserve the 2-norm and the Frobenius norm and are orthogonal. It might be worth clarifying this.

[page 11]: In line 273, better say "Here $W^\dagger$ and $H^\dagger$ denote the pseudoinverses ..." It is then stated that "Running this code in Matlab on a single CPU, we found the update rules using the pseudoinverses were less computationally expensive." Less expensive than what? How are the pseudoinverses computed? What is the computational cost of this part of the algorithm relative to the rest? How many iterations are necessary? Do they lead to convergence? In particular, looking at the code it seems that the method calls MATLAB's `pinv`. Is this necessary? What if the matrix is $m \times n$ with $n << m$? Also reading function `randomized_nnmf.m` one sees the comment: "NOTE: This code is not adapted from any research papers or pre existing code. The only motivation is that this code seems to produce feasible results and we haven't thought of any better alternatives. More testing needs to be done on the convergence of this method to feasible solutions." Has this been done?

[page 12]: In Figure 3, better say that MATLAB notation is used, in which the digits following the `e` symbol are the exponent to which

[page 13]: there is no nonnegative SVD, the authors mean something else (possibly the so called "nonnegative double SVD" initialization from ref. [1]).

[page 19:] In line 525, $W^\dagger$ must be bold according to the paper's notation. In any case, I suggest not to include Appendix B; its findings are well known linear algebra facts.

[page 21, Bibliography]: Line 543: (Burred) Incomplete reference: no date, publication information, etc.

[Bibliography] The authors should also take into account the WNMF work described in the well cited 2008 PhD KU Louvain thesis by N.D. Ho[1]. Another important reference (as noted earlier) is the book by N. Gillis[2].

[Bibliography] The entry "Tan, W., C. S. F. L. L. C. W. Z. and Cao, L...." is probably wrong. See 10.1145/3225058.3225096.

[page 27]: Fig4: Do you mean $\log n$ or $\log(n^2)$. Which base log? What are the exact sizes of the smallest and largest data sets?
* * *
[1]https://perso.uclouvain.be/paul.vandooren/ThesisHo.pdf.
[2]https://doi.org/10.1137/1.9781611976410

---

## Author Response (AR1)

**Author's Response**

Benjamin Sapper[1], Daven Henze[1], and Jose Jimenez[2]

[1]University of Colorado Boulder, 11 Engineering Dr, Boulder, CO 80309, United States
[2]Department of Chemistry and Cooperative Institute for Research in Environmental Sciences (CIRES), University of Colorado, Boulder, Colorado 80309, United States

**Correspondence:** Benjamin Sapper (bsapper77@gmail.com)

We thank the reviewers for their responses. We have considered both reviewers' comments and believe that we have greatly improved the manuscript from them.

In our response, red text is the original comment from the reviewer, black text is our response, and blue text is our updates to the manuscript. Text in italics is text from the original manuscript. All line numbers referenced in our response (black text) are from the revised manuscript.

**1 Review 1**

**1.1 Point 1**

Could the randomized strategy be explained as a stochastic minimization approach for imposing rank constraints on the NMF solution? Specifically, can it be demonstrated that the NMF solution combined with random projection minimizes a certain cost function? This information would help in evaluating the convergence properties of the proposed method.

The external weighting approach laid out in this paper does not minimize a single cost function, rather two in secession: first the expression

$$||A \oslash \Sigma - \tilde{W}\tilde{H}||_F^2 + \mathcal{L}(\tilde{W}, \tilde{H}) \tag{1}$$

where $\oslash$ is elementwise division, and $\mathcal{L}(\tilde{W}, \tilde{H})$ is a function of regularization terms, and then the expression

$$||(\tilde{W}\tilde{H}) \odot \Sigma - WH||_F^2 + \mathcal{L}(W, H) \tag{2}$$

where $\odot$ is elementwise multiplication. The first minimization finds linear factors of the standardized data, and the second minimization attempts to reweight those factors by the uncertainties of the data. The first minimization is a stochastic block coordinate descent (HALS) as laid out in Erichson et al. (2018), and the second minimization follows alternating least squares (ALS). HALS will converge to a stationary point if negative values are set to a value $\epsilon > 0$, while the latter nonnegative ALS method is not guaranteed to converge (Gillis, 2020). We choose not to add any comments to the manuscript about the theoretical convergence of these algorithms, as that is not the objective of this paper. We update the paper as follows on lines 527-529:

We see that different initializations can lead to different solutions, in terms of both similarity to the given PMF2 solution and weighted error, suggesting that convergence to a global minima isn't always achieved. This further emphasizes the importance of using multiple initializations in order to find an optimal solution.

**1.2 Point 2**

Regarding the potential drawback of imposing rank constraints on the NMF outputs, is it feasible to relax these constraints, perhaps by employing the nuclear norm?

Nuclear norm regularization is an increasingly popular tool in matrix factorization as it is the convex envelope to minimizing the rank function of a matrix approximation (Hu et al., 2013). Cai et al. (2010) showed that the unconstrained minimization problem

$$argmin_{\mathbf{X}} \frac{1}{2}||\mathbf{A} - \mathbf{X}||_F^2 + \tau||\mathbf{X}||_*^2 \tag{3}$$

where

$$||\mathbf{X}||_* = \sum_{i=1}^{n} \sigma_i(\mathbf{A})$$

with $\sigma_i(\mathbf{A})$ denoting the $i^{th}$ singular value of $\mathbf{A}$, is solved as $\mathbf{X} = \mathcal{D}_\tau(\mathbf{A})$, which is defined as

$$\mathcal{D}_\tau(\mathbf{A}) = \mathbf{U}(\mathbf{\Sigma} - \tau\mathbf{I})_+\mathbf{V}^T \tag{4}$$

where $\mathbf{U}\mathbf{\Sigma}\mathbf{V}^T$ is the singular value decomposition of $\mathbf{A}$, and $(\mathbf{M})_+$ projects all negative values in $\mathbf{M}$ to zero. Naturally, as $\tau$ increases, the solution $\mathbf{X} = \mathcal{D}_\tau(\mathbf{A})$ decreases in rank.

Two possible ways to apply nuclear norm regularization to weighted PMF are by the Alternating Direction Method of Multipliers (ADMM), as demonstrated in Sun and Mazumder (2013), and by reconstruction of the nuclear norm into a Frobenius norm, which can then be treated as L2 regularization (Fornasier et al., 2011). However, both these approaches involve key computations with the large low rank estimate $\mathbf{X}$, and thus may be computationally expensive to implement. Furthermore, both algorithms still involve some arbitrary choice of rank by requiring a preset amount of nuclear norm regularization $\tau$. In traditional PMF, factor profiles are optimized from a prechosen amount of factors, and thus the we centered our paper around this approach. We update the manuscript on lines 356-367 as:

An increasingly popular alternative to traditional regularization is nuclear norm regularization, which can be applied to matrix factorization without the need for rank constraints (Hu et al., 2013; Sun and Mazumder, 2013; Fornasier et al., 2011). The nuclear norm is defined as

$$||\mathbf{X}||_* = \sum_{i=1}^{n} \sigma_i(\mathbf{A})$$

where $\sigma_i(\mathbf{A})$ is the $i^{th}$ singular value of $\mathbf{A}$. It is the convex envelope to the rank function, and thus finding a PMF solution that minimizes the nuclear norm also has a minimal rank (Hu et al., 2013). Two possible ways to apply nuclear norm regularization

to weighted PMF are by the Alternating Direction Method of Multipliers (ADMM), as demonstrated in Sun and Mazumder (2013), and by reconstruction of the nuclear norm into a Frobenius norm, which can then be treated as L2 regularization (Fornasier et al., 2011). However, both these approaches involve key computations with the large low rank product **WH**, and thus may be computationally expensive to implement. Furthermore, both algorithms still involve some arbitrary choice of rank by requiring a preset amount of nuclear norm regularization. In traditional PMF, factor profiles are optimized from a prechosen amount of factors, and thus the we center our results around this approach.

**1.3 Point 3**

Lastly, the manuscript does not do a good job of citing the related state-of-the-art methods. For example, there has been a tremendous amount of work done in relation to randomized weighted NMF. Please see the following:

Yahaya, F., Puigt, M., Delmaire, G., Roussel, G. (2021, June). Random Projection Streams for (Weighted) Nonnegative Matrix Factorization. In IEEE ICASSP 2021.

Yahaya, F. (2021, November). Compressive informed (semi-) non-negative matrix factorization methods for incomplete and large-scale data: with application to mobile crowd-sensing data. Université du Littoral Côte d'Opale.

Yahaya, F., Puigt, M., Delmaire, G., Roussel, G. (2020). Gaussian Compression Stream: Principle and Preliminary Results. arXiv preprint arXiv:2011.0539.

The authors recognize that work done by Dr. Yahaya and their colleagues was understated in the paper. Thus, we have added two subsections to our manuscript, one in the External Weighting subsection (subsection 1.5) of the Background Section (lines 163-180), and one in the Results Section under subsection 4.3 (lines 482-519). We have also added Table 1 (Table 3 in the revised manuscript).

[revised manuscript text omitted]

**2 Review 2**

**2.1 Main Criticism 1**

The key point of the algorithm is that the combination of RHALS with the external weighting scheme is faster than competing schemes. However, to solidly back this claim, it is not sufficient to only provide runtimes, especially from unoptimized MATLAB implementations. For example, the computational costs of several formulas, e.g the updates on p9, are very sensitive to the order of the operations, the parenthesization, etc. MATLAB is very convenient, but does not attempt to optimize matrix expressions. Were these issues taken into account? The authors also need to show that the "competing" codes have been implemented with runtime minimization in mind. Unfortunately, it is not sufficient to compare performance-oblivious implementations when the objective is to reduce runtimes.

Both the MU and ALS algorithms listed in the paper have been formulated so that computing the relative matrix expressions is optimized. We chose not to explore implementing outside software for running these algorithms so we could specify certain diagnostics and features within the algorithm, such as regularization and the stopping condition. The basic HALS algorithm is translated from Python code referenced in Erichson et al. (2018), which references the function _update_cdnmf_fast in the *scikit-learn* Package in Python (Pedregosa et al., 2011). We argue that this is a "competitive" implementation of HALS.

We note that certain initial diagnostic calculations within these algorithms may slightly increase run times, such as processing the weighting matrix and allocating memory towards tracking the stopping condition, but we keep them within the algorithms for testing, as these are uniform across all algorithms and their contribution to run time is negligible.

MATLAB is well known to be computationally efficient in performing of matrix algebra, and we believe that the implementation of these algorithms in MATLAB yields a fair comparison of computational costs.

Nevertheless, we agree that the manuscript would be improved by adding the number of operations needed for each algorithm. Thus, we have inserted Table 2 into the manuscript.

We have also added the following text to the manuscript in Section 4.1 (lines 404-413)

Table 2 shows the number of operations required at each step (update of $\mathbf{W}$ and $\mathbf{H}$). Preprocessing steps such as finding $\mathbf{A} \oslash \mathbf{\Sigma}$ and $\mathbf{\Sigma} \odot \mathbf{\Sigma}$ are excluded from the table. External Weighting eliminates almost all elementwise operations needed, which is a slow memory bound operation (Jia et al., 2020). However, systems with an abundance of free memory and/or GPUs may find that internal weighting methods are sufficiently quick for small or medium size problems. Furthermore, the performance of matrix operations within these algorithms may vary between programming languages and libraries, such as between the numpy, numexpr, and Theano libraries in Python (Bergstra et al., 2010).

Analyzing Table 2 allows us to see why, in practice, it takes longer for internally weighted ALS and HALS to converge than internally weighted MU, as the former will have more matrix multiplication operations and elementwise operations for our

**Table 2.** Comparison of the number of operations per step for each algorithm used. We also track the number of elementwise multiplications and divisions (implemented in Matlab as ".*" and "./") in the third column.

| Comparison of Cost/Step for Different Algorithms | | |
|---|---|---|
| Algorithm | Number of Operations Using Matrix Multiplication | Number of Elementwise Multiplications/Divisions |
| ALS (IW) | $\mathcal{O}(mnk^2)$ | $2mnk$ |
| ALS (EW) | $\mathcal{O}(mnk)$ | $0$ |
| MU (IW) | $\mathcal{O}(mnk)$ | $4mn + 4(m+n)k$ |
| MU (EW) | $\mathcal{O}(mnk)$ | $4(m+n)k$ |
| MU (Randomized) | $\mathcal{O}((k+l)nk)$ | $4((k+l)+n)k$ |
| HALS (IW) | $\mathcal{O}(mnk^2)$ | $2mnk$ |
| HALS (EW) | $\mathcal{O}(mnk)$ | $0$ |
| RHALS | $\mathcal{O}((k+l)nk)$ | $0$ |
| Post Processing | $\mathcal{O}(mnk)$ | $0$ |

target rank, $k = 6$. However, when the algorithms are externally weighted, ALS and HALS become much more computationally efficient, allowing runtime to diminish as well.

**2.2 Main Criticism 2**

The authors should also specify the costs of each method per iteration (preferably in terms of the matrix primitives used).

See above.

**2.3 Main Criticism 3**

Comparisons are with weighted versions of ALS and MU. These are prone to slow convergence and or even failure to reach a reasonable local optimal solution. ALS, for example (see N. Gillis, p283 of https://doi.org/10.1137/1.9781611976410) "is easy to implement ... and sometimes provides reasonable solutions, typically for sparse matrices; ... However, it comes with no theoretical guarantee and in fact often diverges in practice, especially for dense input matrices; ... Therefore, ALS can be recommended only as a warm-start stage for theoretically better grounded approaches...." See chapter 8 of the aforementioned book and Table 8.2 for a comparison of methods: ALS and MU stand at the "low end" in several respects.

We recognize that ALS has been deemed a suboptimal choice for PMF problems. However, with the data used in the paper, we found that ALS almost always converged to reasonable solutions, albeit at slower run times. The additional algorithms Table 8.2 in Gillis (2020) lists include Projected Gradient Descent (PGM), Alternating Nonnegative Least Squares (ANLS), and Alternating Direction Method of Multipliers (ADMM), as well as techniques to speed up convergence of any NMF algo-

185 rithm, such as updating **H** several times before updating **W**. Since the main purpose of testing several algorithms is to show the differences between using internal and external weighting, as well as randomization, we choose not to add the implementation of these alternative algorithms. However, we add the following sentences to the Background in the manuscript:

In Section 1.2.2 (lines 67-69)

190 It is possible to perform NMF using other forms of gradient descent - for example, the projected gradient method (PGM) sets the step size to the inverse of the maximum eigenvalue of Hessian of the cost function, and may lead to faster convergence than MU (Gillis, 2020). However, we choose to only test MU, due to its widespread use and flexibility (Gillis, 2020).

In Section 1.2.3 (lines 75-82)

195 Nonnegative ALS has no theoretical convergence guarentee and, in some problems, may fail to converge to a feasible so-lution (Gillis, 2020). For this reason, Alternating Nonnegative Least Squares (ANLS) and Alternating Direction Method of Multipliers (ADMM) are interesting alternatives. In ANLS, indices of an "active set" are set to zero, and the rest are updated via an unconstrained optimization (Kim and Park, 2011). The active set is then updated to contain the indices with the new negative factor elements. In ADMM, an auxiliary factor matrix **Y** is formed, and an additional term is added to the cost function

200 which penalizes the distance between the target factor matrix **W** or **H** and **Y** (Gillis, 2020). Both of these methods may lead to faster and better convergence than nonnegative ALS, and are preferred by Gillis (2020). However, we find that the simple nonnegative ALS almost always converges to feasible solutions for our dataset, and we do not explore these alternative methods.

We have also considered these comments for the convergence of the external weighting step – see our response to Main

205 Criticism 4.

**2.4   Main Criticism 4**

In Section 4.1, I am missing an analysis of the contribution of the number of factors to the cost. Indeed, in some cases the discussion is incomplete: For example on p11 it is stated that ".... the algorithm may take longer for a larger number of factors and in certain cases. More work is needed regarding the different convergence issues that may arise." In light of the fact that

210 the external weighting scheme is a key contribution of this paper, the authors need to address these issues in greater detail.

See Point 1 for the discussion of the number of factors and computational costs.

We omit numerical results of the number of factors vs runtime, as we have already listed the computational complexity in our response to point 1.

215 More recent research has found that adding L2 regularization to the post processing step of the External Weighted algorithms improves convergence. Thus, we have deleted the following in Section 2.3:

*However, the algorithm may take longer for a larger number of factors and in certain cases. More work is needed regarding the different convergence issues that may arise.*

and have added (lines 320-326):

In practice, one can use L2 regularization in the external weighting steps, equal to $0.01$ for our data, as the least squares method may become increasingly ill-posed as the number of factors increases. This value may need to be altered based on the magnitude of the values in the data as well as the number of factors. However, we found that adding L2 regularization lowered the similarity of the factors to the given factors from the solution using PMF2.

We used this method for all externally weighted algorithms tested in Section 4. However, theoretically, any nonnegative matrix factorization algorithm could be used for the post processing step. This may become relevant, since as noted in Section 2.2.3, the nonnegative ALS method described above may have convergence issues for certain factorization problems (Gillis, 2020).

**2.5   Main Criticism 5**

It would also be useful to provide information on the cost of each major step and iteration rather than only comparing runtimes to convergence.

See Point 1 for the discussion about computational complexity per step. As we already have a discussion of the percentage of each step of the RHALS algorithm (randomization, main algorithm, and post processing step) in Section 4.1 of the manuscript, we do not add specific numerical results on these values.

**2.6   Main Criticism 6**

The paper makes a brief reference to a WNMF method for missing entries (Yahaya et al.) However the method is dismissed because of the cost of EM without further ado. The authors should address a) how they would handle missing entries, b) how does their method compare in view of related papers by the same team, some included in the publically available thesis of F. Yamaha. It is worth noting that the paper "Random Projection Streams for (Weighted) Nonnegative Matrix Factorization," doi: 10.1109/ICASSP39728.2021.9413496." discusses computational complexity, something I am missing in this manuscript as explained earlier.

a) External Weighting can only apply to continuous weights, as the pre and post processing steps can only take nonzero real entries. We have clarified this by adding the following in Section 2.3 (lines 293-296) in the manuscript:

We note that the uncertainty matrix $\Sigma$ must only contain nonzero real entries. Thus, it can not handle problems with binary weights, such as PMF problems with missing entries. One approach to PMF with binary weights is the Expectation-

Maximization (EM) approach detailed in Section 1.5.1 (Yahaya et al., 2021; Zhang et al., 2006).

b) See Response to Review 1 for our discussion on Dr. Yamaha's results on the Expectation-Maximization approach to weighted PMF , which includes the addition of Section 1.5.1 as mentioned above, as well as Section 4.3.2. We have also deleted the following from Section 2.3:

*Furthermore, how would the uncertainties be included after the dimension of the data is reduced by a randomization step? One way to address the weighting problem introduced by dimension reduction is through an expectation maximization step layed out in (Yahaya et al., 2019). At each iteration, the data matrix A is scaled based on the uncertainties and current factors, and then compressed again into a lower dimension. However, this method is still computationally expensive – in fact, the computationally expensive step of expectation maximization eliminates any of the time benefits gained by randomization.*

**2.7 Main Criticism 7**

I spotted some (trivial to correct but somewhat sloppy-type) errors. Some examples: Formula (7) and the formula for the quality of fit parameter (line 161) are incorrect. In the former, one must use the absolute values of the terms of the sum and in the latter it is the individual terms that must be squared, not the sum. Line 324 on p13 also has such an error. On p. 19, line 527, it is stated that "H is a k × n matrix that is assumed to be of full column rank". However, assuming that k < n, it cannot be full column rank, but at best, full row rank. There might be other errors as well.

These have been corrected: Formula (7) is now

$$Q = \sum_{i=1}^{m} \sum_{j=1}^{n} \left( \frac{\mathbf{A}_{ij} - \sum_{d=1}^{k} \mathbf{W}_{id} \mathbf{H}_{dj}}{\sigma_{ij}} \right)^2$$

and the equation for the quality of fit parameter is now

$$Q = \sum_i \sum_j \left( \frac{\mathbf{A}_{ij} - \sum_k \mathbf{W}_{ik} \mathbf{H}_{kj}}{\sigma_{ij}} \right)^2$$

On p. 13, the weighted squared error formula has been corrected to

$$\sqrt{\sum_i \sum_j \left( \frac{\mathbf{A}_{ij} - \sum_k \mathbf{W}_{ik} \mathbf{H}_{kj}}{\sigma_{ij}} \right)^2}$$

Also, in the Appendix, the phrase

*H is a k ×× n matrix that is assumed to be of full column rank*

has been replaced by

**H** is a k × n matrix that is assumed to be of full row rank.

**2.8 Other Comments**

[page 1]: Please state whether the matrix is assumed to be strictly positive or nonnegative. If the latter, then why use PMF instead of NMF which is more general and the term used in some of the paper's major citations.

285    We see the terms "Positive Matrix Factorization" (PMF) and "Nonnegative Matrix Factorization" (NMF) to be interchangable. For example, Paatero et. al uses the terms "positive" and "nonnegative" in the title of Paatero and Tapper (1994). Eq. 1 on line 35 of the manuscript already specifies that the factor matrices $\mathbf{W}$ and $\mathbf{H}$ can take the value of $0$, and thus we see no need to clarify this further.

290    [page 2 and beyond]: The authors state: "In Eq. (1), the division by $\Sigma$ represents elementwise division". I strongly recommend against using fractional notation when dealing with matrices. They should use Hadamard division or Hadamard multipication with a variable that is defined to contain the inverses of the elements of $\Sigma$. In fact, on p265, the authors actually use the (MATLAB infix operator) for elementwise division. This is certainly better than the / symbol, but even better to use the "Hadamard notation".

295

We have replaced the notation of / for elementwise division with $\oslash$.

[page 3]: Better include the section number when you refer to Sections, e.g. Methods (Section 2)

300    We have rewrote the phrase *the Methods Section* on line 89 as Section 2.

[page 4]: The use of "*" to denote matrix multiplication is inconsistent with the rest of the paper.

We have replaced "$*$" on line 112 with "$\cdot$".

305

[page 4]: A proper and complete discussion on computational optimizations and parallelization would merit much more than the few statements in sections 1.3. I propose that the authors take this into account and eliminate section 1.3.2, incorporating the few comments in some other section.

310    We have gotten rid of section 1.3.2. Section 1.3 is now just titled "Random Projections." We have also clarified the use of GPUs by changing *can be parallelized* to can be parallelized using Graphics Processing Units (GPUs) on line 425.

[page 6]: Please clarify the statement of line 140. How does the incorporation of the weighting via the uncertainty matrix affect the costs?

315

As mentioned earlier in the response to Main Criticism 1, elementwise operations are slow memory bound processes for large matrices. We have updated the manuscript on lines 145-148 as follows:

However, doing this is very computationally expensive due elementwise operations with the uncertainty matrix $\Sigma$. Elemen-
320 twise operations of large arrays are inefficient processes compared to other operations of the same computational complexity, such as matrix-vector multiplication, due to the large allocation of memory towards intermediary results (Jia et al., 2020).

[page 8]: Formula (10) is missing the opeining (left) parenthesis. Also in line 210 better say "denotes the trace of the matrix" rather than "takes the trace of the matrix". After formula (14) better say "Differentiating with respect..." instead of "Taking the
325 derivative ...".

We have fixed these minor inaccuracies.

[page 9]: Careful in the parenthesization of Formula (16). Some parentheses are missing

330

The only issue with Eq. 16 is that we mistakenly returned to elementwise notation while the uncertainty matrices $\Sigma_i$ and $\Sigma_p$ were still diagonal. We replace $\Sigma_i$ and $\Sigma_p$ with $\Sigma(i,:)$ and $\Sigma(:,p)$, the $i^{th}$ row and $p^{th}$ column of $\Sigma$, respectively, in Eqs. 19-24 (previously 16-21).

335 [page 9]: In formula (20) and elsewhere, aren't the $\Sigma_j$ matrices diagonal? If so, then the transpose symbol is redundant. Also careful in the parenthesization of Formula (20). Some parethesizations can be more efficient than others.

See above for the issue with $\Sigma_i$ and $\Sigma_p$. In Eq. 20, the algorithm is made more computationally efficient by finding $\mathbf{H}(\mathbf{H}^T(j,:) \oslash (\Sigma^T(i,:) \odot \Sigma^T(i,:)))$ before multiplying on the left by $\mathbf{W}$ (and similarly in Eq. 21). We also note that the product
340 $\mathbf{WH}$ ($\mathbf{H}^T\mathbf{W}^T$ in Eq. 21) is not preallocated before the update, the same implementation as Erichson et al. (2018). We have updated the manuscript on lines 258-260 as follows.

In Eqs. 20 and 21, the Hessians $\mathbf{H}(\mathbf{H}^T(j,:) \oslash (\Sigma^T(i,:) \odot \Sigma^T(i,:)))$ and $\mathbf{W}^T(\mathbf{W}(:,j) \oslash (\Sigma(:,p) \odot \Sigma(:,p)))$ should be found prior to multiplication by $\mathbf{W}(i,:)$ and $\mathbf{H}^T(:,p)$, respectively, to minimize computational costs. We do not preallocate the prod-
345 ucts $\mathbf{WH}$ and $\mathbf{H}^T\mathbf{W}^T$, which is the same implementation as in Erichson et al. (2018).

 In formula (20) and elsewhere: have you defined the maximum operator $[\cdot]_+$?

We have updated the manuscript on line 248 as:

where $[\boldsymbol{v}]_+$ projects all negative values of $\boldsymbol{v}$ to $0$.

 Some readers might question the characterization "rotation" for a non-orthogonal matrix. In the linear algebra literature, rotation matrices preserve the 2-norm and the Frobenius norm and are orthogonal. It might be worth clarifying this.

We agree with the reviewer that it is best to clarify this. We add to Section 1.4 (lines 118-119):

We note that $\mathbf{T}$ does not necessarily represent a true rotation in a mathematical form, which would require $\mathbf{T}$ to be orthogonal.

and to Section 2.2 (lines 262-263)

As mentioned in Section 1.4, we do not attempt to constrain these "rotations" to be norm preserving. However, it is possible to find approximate rotations.

We also put quotation marks around the first word "rotational" in Section 1.4 (line 117). Additionally, we cite Paatero et al. (2002) on line 123.

 In line 273, better say "Here $W^\dagger$ and $H^\dagger$ denote the pseudoinverses ..." It is then stated that "Running this code in Matlab on a single CPU, we found the update rules using the pseudoinverses were less computationally expensive." Less expensive than what? How are the pseudoinverses computed? What is the computational cost of this part of the algorithm relative to the rest? How many iterations are necessary? Do they lead to convergence? In particular, looking at the code it seems that the method calls MATLAB's pinv. Is this necessary? What if the matrix is m×n with n $<<$ m? Also reading function randomized nnmf.m one sees the comment: "NOTE: This code is not adapted from any research papers or pre existing code. The only motivation is that this code seems to produce feasible results and we haven't thought of any better alternatives. More testing needs to be done on the convergence of this method to feasible solutions." Has this been done?

We have replaced

*Here, $\boldsymbol{W}^\dagger$ denotes the pseudoinverses of $\boldsymbol{W}$*

with (line 304)

Here, $\mathbf{W}^\dagger$ and $\mathbf{H}^\dagger$ denotes the pseudoinverses of $\mathbf{W}$ and $\mathbf{H}$

We have addressed the Computational Cost in 2 in response to Main Criticism 1. Both proposed post processing steps are $\mathcal{O}(mnk)$, and we found experimentally that using MATLAB's pinv, a Krylov subspace method (Feng et al., 2018), generated slightly faster results. As mentioned on page 11 on line 287, the algorithm takes 20-40 iterations to converge. As mentioned in response to Main Criticism 4, regularization can be added to improve the convergence, especially if a large number of factors are chosen. As noted in the Appendix of the manuscript, using the pseudoinverse is equivalent to using least squares, as long as $\mathbf{W}$ and $\mathbf{H}$ are full rank. If n ≪ m, the update of $\mathbf{W}$ using $\mathbf{H}^\dagger$ is a less overdetermined problem compared to the update of $\mathbf{H}$. However, since $\mathbf{W}$ and $\mathbf{H}$ are alternatingly updated, it is not clear as to whether the algorithm would be affected by changing the ratio of $m$ to $n$. That being said, further improvements in the external weighting algorithm are a subject for future research.

We have updated the manuscript on lines 309-310:

In Matlab, a Krylov subspace algorithm is used for calculating the pseudoinverses, and both update rules are $\mathcal{O}(mnk)$ (Feng et al., 2018).

Additionally, we replaced

*Running this code in Matlab on a single CPU, we found the update rules using the pseudoinverses were less computationally expensive.*

with (lines 310-311)

Running this code in Matlab on a single CPU, we found the update rules using the pseudoinverses were faster.

We also removed the above comment in the code for randomized_nnmf.m. The new doi for the algorithm code is https://zenodo.org/badge/latestdoi/537111580.

[page 12]: In Figure 3, better say that MATLAB notation is used, in which the digits following the e symbol are the exponent to which

Since many readers may not be familiar with MATLAB notation, we keep the description of "e" as is. However, we add to line 344

and the number to the right is the exponent (i.e. e03=$10^3$).

[page 13]: there is no nonnegative SVD, the authors mean something else (possibly the so called "nonnegative double SVD" initialization from ref. [1]).

420    We have corrected lines 372-373 to read the nonnegative double SVD (NNDSVD).

[page 19:] In line 525, $\mathbf{W}^{\dagger}$ must be bold according to the paper's notation. In any case, I suggest not to include Appendix B; its findings are well known linear algebra facts.

425    This has been corrected. We keep Appendix B, as some readers may have a background primarily in atmospheric chemistry, and may be interested as to why the external weighting algorithms are identical mathematically.

[page 21, Bibliography]: Line 543: (Burred) Incomplete reference: no date, publication information, etc.

430    We have added the following peer reviewed reference to line 61 (where Burred is orginally cited): (Gillis, 2020). We have also added a more complete reference for the original source – see Burred (2017).

[Bibliography] The authors should also take into account the WNMF work described in the well cited 2008 PhD KU Louvain thesis by N.D. Ho . Another important reference (as noted earlier) is the book by N. Gillis.

435

We have added in the citation of N.D. Ho's thesis as an alternative derivation of weighted HALS into Section 2.1 on line 216.

Another derivation of weighted HALS is given in Ho (2014).

440    [Bibliography] The entry "Tan, W., C. S. F. L. L. C. W. Z. and Cao, L...." is probably wrong. See 10.1145/3225058.3225096.

See Tan et al. (2018).

[page 27]: Fig4: Do you mean log n or $log(n^2)$. Which base log? What are the exact sizes of the smallest and largest data sets?

The logarithm used is the base 10 log, and it is taken of $n$. The caption to Figure 4 has been updated as

Size of data matrix vs run time (in seconds) in RHALS algorithm, performed with three different random seeds (containing the absolute value of random normal variables). The x-axis shows the base 10 log of $n$, and the y-axis plots the base 10 log of run time. The median run time is plotted, with the error bars plotting the maximum and minimum run times. Note that run

time initially decreases due to a coincidental reduction in the number of steps to convergence, but problems of environmental interest are generally located on the right of the graph or beyond its right edge.

**3  Additional Changes**

In section 1.3, we have replaced $\mathbf{Q}$ with notation $\mathbf{P}$ to make the difference between the randomized projection and the cost function and quality of fit parameters clear.

We changed $A$ to $\mathbf{A}$ on line 107. Additionally, we changed the sentence (lines 107-108)

*Each vector $\boldsymbol{y}$ is slightly pushed out of the column space by the term $\mathbf{E}\boldsymbol{\omega}$.*

to

Each vector $\boldsymbol{y}$ is slightly pushed out of the column space of $\mathbf{B}$ by the term $\mathbf{E}\boldsymbol{\omega}$.

We replaced

*The authors concluded that a significant portion of the Secondary Organic Aerosol (SOA) was anthropogenic in origin.*

with (lines 202-203)

The authors concluded that a significant portion of the Secondary Organic Aerosol (SOA) was the result of interactions between biogenic and anthropogenic emissions.

We replaced

*Suppose a factor is interpretable by a scientist running a PMF algorithm, but a particular component is lacking or overrepresented. The scientist might be interested in a rotated solution that includes the complete factor.*

with (lines 262-264)

Suppose that a solution is mostly interpretable by a scientist running a PMF algorithm, but some aspects of the solution appear unrealistic, e.g. a factor is zero during a period in which it is expected to be present, or 2 factors appear mixed in their

time series and/or spectra. The scientist might be interested in a rotated solution that adjusts this.

To clarify the addition of regularization into the cost function, we changed

*while L2 regularization is added to control the norms of the factors as well as avoid overfitting*

to (lines 331-332)

while L2 regularization is added to control the Euclidean norms of the factors as well as avoid overfitting ill-posed problems (Erichson et al., 2018; Hansen and O'Leary, 1993).

Additionally, we clarify that regularization is not necessarily crucial for our specific problem, by adding (lines 348-351)

We note that choosing much smaller regularization values does not drastically increase the norms of the solution, suggesting that regularization is not especially necessary for this factorization. However, for more ill-posed problems, the L1 and L2 norms may become extremely large as the amount of regularization tends to zero (Hansen and O'Leary, 1993).

In line 421, *proportional* is changed to linear.

To emphasize the possibility that other solutions rotated away from PMF2 can be interpreted as valid factors of the data, we add (lines 538-539) This rate could vary depending on the number of factors and the rotational ambiguity of the problem, as in the "bad" trials RHALS may simply be finding a rotated version of the "true" solution.

We had incorrectly labeled the the mass spectra (m/z) in Figures 1, 8, 12, A4, and A5 as the index, and the x axis have been changed to reflect the actual mass to charge ratio (m/z). We also replotted Figures 7, A2, and A3 to show the comparison between the externally and internally weighted time series, as we could not find the random seed initially used to generate the original plots. The weighted errors of the externally weighted HALS, ALS, and MU algorithms have been changed from *7.1120* $\times 10^3$*, 7.2465*$\times 10^3$*, and 7.0239*$\times 10^3$ to $7.1321 \times 10^3$, $7.3666 \times 10^3$, and $6.9457 \times 10^3$, respectively, and the weighted errors of the internally weighted HALS, ALS, and MU algorithms have been changed from *6.4657* $\times 10^3$*, 6.4853*$\times 10^3$*, and 6.3900*$\times 10^3$ to $6.4841 \times 10^3$, $6.4768 \times 10^3$, and $6.4292 \times 10^3$, respectively. The interpretation of these plots does not change.

A legend was added to each subplot in Figure 3 detailing the regularization and magnitude of the data for each L curve, and the y axes of Figures 5, 6, 9, 14, A7, and A9 were adjusted for clearer interpretation.

The phrase *Actual Solution* was changed to PMF2 Solution in the title of Figures 15, A8, and A10.

520    The line but problems of environmental interest are generally located on the right of the graph or beyond its right edge was added to the end of the caption of Figure 4, and the values in the captions of Figures 5 and 9 were rounded to the appropriate amount of significant figures.

We have corrected each reference as parenthetical or textual, and we also reformatted Eqs. 2 and 38 (previously 35) to 525    remove spaces Additionally, other spelling and grammar mistakes were corrected.

**References**

[revised manuscript text omitted]

---

## Author Response (AR2)

**Response to July 2023 Review**

Sean Youn

July 2024

This document addresses comments from reviewers for the article "Positive Matrix Factorization of Large Real-Time Atmospheric Mass Spectrometry Datasets Using Error-Weighted Randomized Hierarchical Alternating Least Squares" (Sapper et al.). Note the change in the title (previously "Positive Matrix Factorization of Large Aerosol Mass Spectrometry Datasets Using Error-Weighted Randomized Hierarchical Alternating Least Squares") in order to be more precise with regards to the principal dataset used in the presented analysis.

The original comments by the reviewer are presented in red text, our responses are shown in black text, and significant additions to the manuscript are presented in blue text. In the time since the last submission of this manuscript, we have added two new co-authors (Youn and Jimenez). Some additional changes only affecting readability of the manuscript have been made as well.

**1 Reviewer 1**

No comments or criticisms to address

**2 Reviewer 2**

**2.1 Paragraph 1**

In the Appendix there was a statement that H is a kxn matrix that is assumed to be full column rank (even though k was assumed to be smaller than n). In their response, the authors say that this (erroneous) statement was corrected with "H is a kxn matrix that is assumed to be full row rank". However, the revised document (line 666) still has the wrong statement.

The text in question is present in line 630 in the updated manuscript and has been modified appropriately.

However, there seem to be cases where this [Hadamard notation] was not applied, e.g. lines 158 and 161 (there might be others)

Equations 1, 3, 11, 19, 20, 21, 22, 23, 24, 38, and A1 and lines 75, 89, 259, 293, and 405 have been updated to incorporate Hadamard notation for element-wise division in particular. Note that line numbers have changed between the version

viewed by the reviewer and the current manuscript due to other changes in the main text. We believe all presented equations now make full use of Hadamard notation when appropriate as suggested by the reviewer.

**2.2   Paragraph 2**

The authors claim in the abstract that their algorithm results in computational speedups of 38, 67 and 634 compared to other algorithms. The statement is not justified by the methodology adopted in this paper. In particular, there is no detailed performance evaluation, neither a careful optimization of the methods. A modest statement, e.g. "numerical experiments with the proposed method indicate that the method is faster than competing algorithms" would be more appropriate

In order to address this comment and other comments in 2.3, the abstract text will be changed to the following:

"Weighted positive matrix factorization (PMF) has been used by scientists to find small sets of underlying factors in environmental data. However, as the size of the data has grown, increasing computational costs have made it impractical to use traditional methods for this factorization. In this paper, we present a new external weighting method to dramatically decrease computational costs for these traditional algorithms. The external weighting scheme, along with the Randomized Hierarchical Alternating Least Squares (RHALS) algorithm, was applied to the Southern Oxidant and Aerosol Study (SOAS 2013) dataset of gaseous highly oxidized multifunctional molecules (HOMs). The modified RHALS algorithm successfully reproduced six previously-identified, interpretable factors with the total computation time of the non-optimized code showing potential improvements on the order of one to two orders of magnitude compared to competing algorithms. We also investigate rotational ambiguity in the solution, and present a simple "pulling" method to rotate a set of factors. This method is shown to find alternative solutions, and in some cases, lower the weighted residual error of the algorithm."

**2.3   Paragraph 3**

This paper essentially uses a modification of the NMF to target a specific dataset (analyzed extensively by Massoli et al.) for which k=6 factors have been found to be appropriate. The paper makes no effort to consider different combinations of (m,n,k). It seems to me that this makes the paper very application and data specific and possibly not as interesting. In any case, the authors should declare this early in the paper so that the readers are not misled regarding the generality of the method.

Text and wording to clarify the main application/study of the algorithm to SOAS (2013) have been included in the revised abstract already shown in 2.2. However, experimentation on matrices of different sizes, the predominant variable between real-time datasets obtained with different mass spectrometers, was done in this study (see Section 4.2 (Simple Case) and Figure 4) and is

discussed accordingly. Analysis of the operations per step for each algorithm is also presented in Table 1 for general values of $m, n$, and $k$. In addition, five or six factors are typical in real-time atmospheric and aerosol mass spectrometry data (Ulbrich et al. (2009); Massoli et al. (2018); Zhang et al. (2011)). Though variation in the optimal number of factors can theoretically occur depending on the dataset, significant deviation from five or six factors is rare in practice.

Real-time mass spectrometry data sets are variable in terms of the matrix size, and we believe this constitutes enough diversity to be useful to the general atmospheric/aerosol mass spectrometry community. To reflect some of the reviewer's concerns, the following addition will be made to Section 1.7 (Data):

"We use this six factor solution as a reference solution, and test whether the RHALS algorithm can recreate formulated factors as well as those found from PMF2. Analysis of results for different numbers of factors (other than the original six identified in Massoli et al. (2018)) were not considered in order to maintain interpretability of the algorithm output. For reference, the PMF2 factor mass spectra and the time trends over all of the data are shown in Figure 1. The factor time series, as well as the time series of the total mass concentration is also shown in Figure 2. Both plots show total concentration amounts over the entire time series and mass spectra respectively."

**2.4   Paragraph 4**

The authors insist on using the term PMF instead of NMF (because of the use of the term by Paatero et al. However, the matrices used in the paper are not positive, they are nonnegative. Calling a matrix with some zero elements positive is mathematically incorrect, with all due respect to the pioneering work of Paatero et al. Indeed, there are results that hold for positive matrices but do not hold for nonnegative ones unless extra conditions are imposed (e.g. the Perron theorem

We acknowledge that the reviewer is entirely correct in their assertion that nonnegative matrix factorization is a more precise term than positive matrix factorization from a mathematical perspective (and that the two are not necessarily always equivalent). However, positive matrix factorization remains the dominant nomenclature, especially in the community most relevant to this study due to the explicit use of the term PMF by the EPA for their model/software (by Paatero). As of this writing, a quick search in Google Scholar lists about 506,000 results for "positive matrix factorization" and about 202,000 results for "nonnegative matrix factorization." Furthermore, a search of "positive matrix factorization aerosol" yields 16,800 results compared to 5,170 from "nonnegative matrix factorization aerosol."

Nevertheless, to avoid any further confusion, we have made the following modification to line 36 in Section 1.1 (Problem Statement):

"In Eq.(1), $\oslash$ represents elementwise division, the norm $||\cdot||_F$ is the Frobenius norm, and all elements of $\mathbf{W}$ and $\mathbf{H}$ are constrained to be nonnegative. Further, we note that for consistency with nomenclature in the literature related to use of this algorithm for factorization of aerosol mass spectrometry datasets, we refer

to this approach as "positive" matrix factorization (i.e., PMF) while recognizing that a more precise name would be nonnegative matrix factorization."

**2.5 Paragraph 5**

The statement in the authors' response "We have addressed the Computational Cost in 2 in response to Main Criticism 1. Both proposed postprocessing steps are O(mnk), and we found experimentally that using MATLAB's pinv, a Krylov subspace method (Feng et al.,2018), generated slightly faster results." is wrong, at least regarding the MATLAB pinv function. It is likely that the authors were misled by the discussion in the reference by Feng et al. which argues (rightly) for using the lansvd of PROPACK instead of the Mathworks svds for performing large scale truncated svd. On the other hand, the native MATLAB pinv does not use svds but the native MATLAB svd function which is certainly not a Krylov method. Incidentally, this is another example where the choice of citation matters. This affects lines 305-311 of the text and possibly others.

The section in question is in Section 2.3 (External Weighting). In order to avoid confusion regarding the issue raised by the reviewer, reference to MATLAB's pinv function as a Krylov subspace method was removed from the text as it is not critical to understanding the main mechanisms of the presented algorithms.

**2.6 Paragraph 6**

Throughout the paper, the authors use the symbol k sometimes as index and other times as the rank of the sought factorization which is confusing. Also, in some summation formulas (e.g. 2), the authors use summation (e.g. over k) without specifying the range while in others (e.g. 7) they do.

Notation has been standardized to refer to $m$, $n$, and $k$ as the number of rows, columns, and factors respectively and to to use $i$, $j$, and $l$ to refer to indices of the rows, columns, and factors respectively (as is needed in summation notation, among other cases). Summations have also been modified to explicitly state the ranges of summation in all cases.

If $Q_j$ is defined as stated (minimizing) then the min operator on the left-hand side of relation (11) is redundant.

The true cost function is defined as follows:

$$Q_j = ||(\mathbf{R}_j - \mathbf{W}(:, j)\mathbf{H}(j, :)) \oslash \boldsymbol{\Sigma}||_F^2 \tag{1}$$

Minimizing the cost-function is purpose of the algorithm, but the cost function itself is not the minimization of the equation as currently presented. References to the cost function have been modified to match the equation in 1 and the wording/equation in Section 2.1 (HALS Algorithm)) has been modified to the following:

"The HALS algorithm applies block coordinate descent methods in order to minimize the cost function $Q_j$ by minimizing a "block," or outer product of

individual factors, of $\mathbf{W}$ and $\mathbf{H}$ at a time while keeping the other factors fixed. (Erichson et al. (2018))."

$$Q_j = ||(\mathbf{R}_j - \mathbf{W}_{(:,j)}\mathbf{H}_{(j,:)}) \oslash \mathbf{\Sigma}||_F^2 \qquad (2)$$

Using both $Q_j^i$ and $Q_j^P$ (defined differently) is confusing.

$Q_j^i$ and $Q_j^p$ are first introduced and defined in Equations 13 and 14 respectively. Lines 228 to 230 (immediately preceding these equations) are modified to the following to make clear we are differentiating between rows and columns of the matrices:

"To derive update rules for HALS, partial derivatives of Eq. (1) are taken with respect to the factors $\mathbf{W}_{(:,j)}$ and $\mathbf{H}_{(j,:)}$. With $\mathbf{\Sigma}$ present, this can become tricky, so we present a variation on the derivation presented in Erichson et al. (2018) by considering just a row $(i)$ and column $(p)$ of the weighted residual."

The last comma between W(:,j) and H(i,:) should be eliminated.

If this comment is in reference to Eq. 12 in Section 2.1 (HALS Algorithm), the comma has been removed.

The sizes of the matrices Sigmai, Sigmap and Sigma should be explicitly specified.

To define the sizes of the matrices, the following changes will be made:

Line 30 will be changed to:

"...suppose that accompanying the dataset $\mathbf{A}$ is an equally sized $(m \times n)$ matrix $\mathbf{\Sigma}$ ..."

Line 233 will be changed to:

"In Eq. (13) and Eq. (14), $\mathbf{\Sigma}_i$ are $\mathbf{\Sigma}_p$ are diagonal matrices (of size $m$ and $n$ respectively) with the diagonal elements corresponding to the elements of the $i^{th}$ row (for $\mathbf{\Sigma}_i$) and $p^{th}$ column (for $\mathbf{\Sigma}_p$) of $\mathbf{\Sigma}$."

The use of italics for Tr and arg is not consistent with the use of romans for min and max. In mathematical typesetting it is better to keep these non-italicized/sans serif.

Notation for trace has been corrected from $Tr$ to Tr in lines 233, 609, 617 and Equations 15, 16, 17, 18, A4, A5. $argmax_{\mathbf{WH}}$ in Equations 8 and 10 are now presented as $\text{argmax}_{\mathbf{WH}}$. The use of $Q_{aux}$ to denote auxiliary terms in section 2.2 (Rotational Considerations) has been modified to $Q_{\text{aux}}$. In section 4.2 (Simple Case), the use of "test" to in matrix subscripts is also converted to non-italicized text (i.e. $\mathbf{A}_{\text{test}}$)

Further similar changes beyond those listed above have been made to ensure the following of appropriate typesetting conventions in the manuscript.

Notation becomes quite confusing in (27) and beyond.

Equations 32 and 33 now precede Equations 30 and 31 in order to maintain consistency with the ordering of equations 28 and 29 (from which equations

30 to 33 were derived). Further changes to maintain notation consistency and increase clarity in equations throughout the manuscript are addressed in other comments.

On p11, last line, it should be "Thus, the algorithm cannot handle ..." to make it clear that you are referring to the algorithm anot not to the uncertainty matrix.

The comment refers to the second sentence of Section 2.3 (External Weighting). The wording has been modified as suggested.

On p12, "denotes the pseudoinverses" should be "denote the pseudoinverses".

The line in question in also in Section 2.3 (External Weighting) and has been modified as suggested.

It is stated that "Mathematically, these methods ..." - state which are these methods.

This comment is in reference to a sentence (line 305) in Section 2.3 (External Weighting). The sentence in question is referring to two sets of equations presented in Equations 35 and 36. The sentence will be modified to the following for clarity:

"Mathematically, the two methods for calculating $\mathbf{W}$ and $\mathbf{H}$ detailed in Equations 35 and 36 respectively are identical, as long as the rank of the factor matrices is equal to $k$ ..."

On p12, line 312 and beyond, the initialization of H is not clear (also better say "initialize" rather than "initiate"). In particular, formula (37) shows $H_0$ on the left side and H on the right. How is H chosen on the right side?

The wording has been modified from "initiate" to "initialize" as suggested. Equation 37 and the text above it describing the appropriate scaling factor have been modified to the following to remove use of $mean()$ to denote the element-mean of a matrix and also to include $\hat{\mathbf{H}}$ on the right hand side of the equation (as opposed to $\mathbf{H}$ in previous versions of the manuscript):

$$\mathbf{H}_0 = \left( \sqrt{\frac{\bar{\mathbf{A}}}{k \bar{\mathbf{H}}^2}} \right) \hat{\mathbf{H}} \tag{3}$$

In Eq. (3), $\bar{\mathbf{A}}$ denotes the element-mean of matrix $\mathbf{A}$ and $\bar{\mathbf{H}}$ denotes the element-mean of $\hat{\mathbf{H}}$. As $\hat{\mathbf{H}}$ is defined earlier in line 297, this should be sufficient to clarify the initialization of $\mathbf{H}_0$.

On p12, line 319 and possibly elsewhere, the font for H and W is inconsistent (non-bold) with the (bold) font for the same variables elsewhere in the text.

Notation has been standardized to utilize bold font for all matrices and bold, italic font for vectors.

When defining norms it is better to use some space, a variable symbol or a dot instead of using the two double vertical bars next to each other.

Matrix norms (previously denoted $||||_F$ in the case of the Frobenius norm, for example) have been changed to $|| \cdot ||_F$

In line 377, it is stated that MU is initialized with random numbers. I assume this means that the W and H factors in MU are initialized with nonnegative random values. If this is so, it needs to be said. The distribution also needs to be mentioned (e.g. are they uniformly distributed pseudorandom numbers in [0,1]?)

This comment refers to the second paragraph of the Results section. The paragraph will be modified as follows:

"In the MU algorithm, the elements of $\mathbf{W}$ and $\mathbf{H}$ are initialized by taking the absolute value of random numbers drawn from a standard normal distribution. These values are then scaled by a factor of $\bar{\mathbf{A}}^2$. The ALS and HALS algorithms are initialized by the nonnegative double SVD (NNDSVD) approach detailed in Boutsidis and Gallopoulos (2008)."

The quantity $\bar{\mathbf{A}}$ is previously-defined earlier in the manuscript to represent the element-mean of the matrix $\mathbf{A}$.

The authors still make the error and use the term "non-negative SVD" in line 379, in contrast with the term used for NNDSVD in line 378. See also the comment labeled [page13] in my original review.

The instance of "nonnegative SVD" mentioned by the reviewer has been corrected to "NNDSVD" (previously defined as nonnegative double SVD).

It is also hard not to notice that the authors write "nonnegative" in lines 377-8 and "non-negative" in line 379.

Instances of words prefaced by the prefix "non" (e.g. "non-negative") have been changed such that they are unhyphenated (e.g. "nonnegative"). It is possible though that LaTeX is hyphenating the words if they wrap around to a new line.

On p. 8, the W(:,j), H(j,:) are introduced as the j-th column and row of W and H, while in line 394, the notation becomes $W_j$ and $H_j$. Incidentally, if you follow the parenthesized notation, then element at row-i, column-j should be denoted by W(i,j) and not $W_{ij}$.

All equations have been modified to utilize the following notational conventions:

- $\mathbf{W}_{(:,j)}$ and $\mathbf{H}_{(j,:)}$ denote the $j^{th}$ column and row of matrices $\mathbf{W}$ and $\mathbf{H}$ respectively

- When necessary, $\mathbf{R}_{j(i,:)}$ is used to denote the $i^{th}$ row of the $j^{th}$ residual matrix ($\mathbf{R}$)

- $\mathbf{W}_{ij}$ denotes the element in row $i$ and column $j$ in matrix $\mathbf{W}$. This is chosen in contrast with the reviewer's suggestion that $\mathbf{W}_{(i,j)}$ be used to denote the element of matrix $\mathbf{W}$ in order to reduce clutter within equations (and $\mathbf{W}_{ij}$ is common, accepted notation)

**2.7   Paragraph 7 and 8**

Concerning the bibliography: Some of the references are not suitable. For example: Why refer to Tan et al. 2018 and Takacs and Tikk for HALS? Have these papers introduced HALS? To my understanding, they are only using some version of HALS for some application. Why not simply refer to the primary sources for HALS or even better references that provide a solid foundational discussion, e.g. the monograph by Gillis or the book by Cichocki, Zdunek et al. I would also assume that there must be some specific journal bibliography style.

These references have been removed in favor of the book by Cichocki et al. (2009) as suggested by the reviewer. Formatting in the bibliography has also been addressed.

Several references are incomplete or shown in an inconsistent manner. For example: The Ph.D. thesis by Yahaya does not show the institution. Some wordings are capitalized for no good reason (the institution for Ho's thesis). Using url citations (e.g. Burred J.) that have not undergone proper peer review or at least are posted on a reputable archive (e.g. ArXiv) is not good practice. For this particular one, either the Gillis monograph or the aforementioned monograph by Cichocki, Zdunek et al. noted earlier would be sufficient and more appropriate.

Formatting and content of the references have been updated for completeness and presentation. The reference by Burred has been removed and replaced with peer-reviewed sources as suggested by the reviewer.

**References**

Boutsidis, C. and Gallopoulos, E.: SVD based initialization: A head start for nonnegative matrix factorization, Pattern Recognition, 41, 1350–1362, https://doi.org/10.1016/j.patcog.2007.09.010, 2008.

Cichocki, A., Zdunek, R., Phan, A. H., and Amari, S.: Alternating Least Squares and Related Algorithms for NMF and SCA Problems, chap. 4, pp. 203–266, John Wiley  Sons, Ltd, ISBN 9780470747278, https://doi.org/10.1002/9780470747278.ch4, 2009.

Erichson, N. B., Mendible, A., Wihlborn, S., and Kutz, J. N.: Randomized Nonnegative Matrix Factorization, Pattern Recognition Letters, 104, 1–7, https://doi.org/10.1016/j.patrec.2018.01.007, 2018.

Massoli, P., Stark, H., Canagaratna, M. R., Krechmer, J. E., Xu, L., Ng, N. L., Mauldin, R. L., Yan, C., Kimmel, J., Misztal, P. K., Jimenez, J. L.,

Jayne, J. T., and Worsnop, D. R.: Ambient Measurements of Highly Oxidized Gas-Phase Molecules during the Southern Oxidant and Aerosol Study (SOAS) 2013, ACS Earth and Space Chemistry, 2, 653–672, https://doi.org/10.1021/acsearthspacechem.8b00028, 2018.

Ulbrich, I. M., Canagaratna, M. R., Zhang, Q., Worsnop, D. R., and Jimenez, J. L.: Interpretation of organic components from Positive Matrix Factorization of aerosol mass spectrometric data, Atmospheric Chemistry and Physics, 9, 2891–2918, https://doi.org/10.5194/acp-9-2891-2009, 2009.

Zhang, Q., Jimenez, J. L., Canagaratna, M. R., Ulbrich, I. M., Ng, N. L., Worsnop, D. R., and Sun, Y.: Understanding Atmospheric Organic Aerosols via Factor Analysis of Aerosol Mass Spectrometry: A Review, Analytical and Bioanalytical Chemistry, 401, 3045 – 3067, 2011.

---

## Author Response (AR3)

**Response to February 2025 Review**

Sean Youn

February 2025

This document addresses comments from reviewers for the article "Positive Matrix Factorization of Large Real-Time Atmospheric Mass Spectrometry Datasets Using Error-Weighted Randomized Hierarchical Alternating Least Squares" (Sapper et al.).

The original comments by the reviewer are presented in red text, our responses are shown in black text, and significant additions to the manuscript are presented in blue text. References to line numbers below correspond to the manuscript version submitted August 2024 for review.

**1 Reviewer 3 Comments**

**1.1**

Since the article was submitted in 2022, the latest references are from 2021. I suggest that the authors perform a short review of some recent literature to incorporate some of the latest developments.

The following additions have been made to Section 1.1 to acknowledge recent literature in the field of positive/nonnegative matrix factorization (particularly for large datasets and with application to atmospheric composition analysis).

**Line 40:** "Traditional factor analysis methods are known to be computationally expensive. Steps to speed up factor analysis have been explored, such as randomization and the use of graphical processing units (GPUs) (Halko et al. (2011); Tan et al. (2018)). Developing efficient algorithms is especially critical in atmospheric mass spectrometry, as improvements in instrumentation and increases in the duration of their use in field campaigns has led to intractably large datasets. Currently, analysis of these datasets requires sacrificing data resolution or extensive manual preprocessing to operate within existing PMF software tools, and full analysis can routinely take days or weeks of computation time (Hopke et al. (2023)). As a result, a variety of approaches have emerged for efficient source apportionment of atmospheric mass spectrometry data. Algorithms to solve the nonconvex optimization posed by PMF range from gradient descent, block coordinate descent, and projected gradient methods (Guo et al. (2024)). Attempts at using supervised, ensemble machine learning approaches

have been shown to be capable of replicating results from traditional (unsupervised) factorization methods while reducing computation time (Zhang et al. (2025)). Recently, Erichson et al. (2018) applied randomization to PMF and introduced a new method, randomized hierarchical alternating least squares (RHALS), to solve the unweighted PMF problem. In this paper, we test the application of RHALS to atmospheric concentration data that contain uncertainties. Accounting for these uncertainties as regression weights, we introduce a method of externally weighting and unweighting the data, which to our knowledge is novel in its application to RHALS. We consider the accuracy and the reduced computational costs compared to other PMF algorithms commonly used in the field of atmospheric science."

**Line 136:** "It is not feasible to span all possible variants that $\mathbf{T}$ can take. Thus, the problem is often simplified to considering only positive rotations (values of $\mathbf{T}$ greater than zero) and negative rotations (values of $\mathbf{T}$ less than zero). A rotational program in PMF2 called FPEAK uses the parameter $\phi$ to denote the rotation strength, with positive values leading to positive rotations in $\mathbf{W}$ (Paatero (1997)). Paatero further improved this method in the Multilinear Engine (ME) algorithm, where the strength of rotation is allowed to vary between factors (Paatero and Hopke (2009)). The pulling algorithm presented in Paatero and Hopke (2009) is a sophisticated rotational method; more rudimentary pulling methods that mimic varying the regularization of the factor matrices are presented in Paatero (1997) and Paatero et al. (2002). Recent attempts at controlling for rotational ambiguity have involved additional factorization of the time-series matrix $\mathbf{W}$ into a matrix incorporating shape regularization to reflect known diurnal patterns of factors and a diagonal scaling matrix (Nanra et al. (2024))."

**1.2**

Some aesthetic changes - Authors can work on improving the plots and figures presented in the paper and try to make them uniform across all the figures presented in the paper. Numbers and lines in some of plots are too small to be properly visible when printed out.

While the authors agree in spirit with the reviewer's comment, the untimely passing of the fist author, loss of access to his original figures, and the inherently stochastic nature of the methodology means that we are unable to reproduce and make stylistic improvements to the figures in the manuscript. However, all figures are vector graphics and can therefore be zoomed in on digital versions with high fidelity.

**1.3**

While not relevant for publication, the authors should consider making detailed documentation of the code written in MATLAB available for others to use with

ease. This will enable a larger adoption of the proposed RHALS method.

The authors appreciate and agree with the suggestion. For the reasons mentioned in Section 1.2, we are unable to make changes to the GitHub repository containing the EW-RHALS MATLAB code. However, as the goal is to make EW-RHALS open source and widely available, we are currently porting the code from MATLAB to Python. At the time when we are ready to publish the Python code, we will ensure there is detailed documentation of the code and algorithms.

**2    Topic Editor Comments**

**2.1**

Please consider the few points raised in the last review. Regarding the figures, please also label all panels with (a),(b),...

All figures have been amended to include panel labels as needed.

**References**

Erichson, N. B., Mendible, A., Wihlborn, S., and Kutz, J. N.: Randomized Nonnegative Matrix Factorization, Pattern Recognition Letters, 104, 1–7, https://doi.org/10.1016/j.patrec.2018.01.007, 2018.

Guo, Y.-T., Li, Q.-Q., and Liang, C.-S.: The rise of nonnegative matrix factorization: Algorithms and applications, Information Systems, 123, 102 379, https://doi.org/https://doi.org/10.1016/j.is.2024.102379, 2024.

Halko, N., Martinsson, P. G., and Tropp, J. A.: Finding Structure with Randomness: Probabilistic Algorithms for Constructing Approximate Matrix Decompositions, SIAM Review, 53, 217–288, https://doi.org/10.1137/090771806, 2011.

Hopke, P. K., Chen, Y., Rich, D. Q., Mooibroek, D., and Sofowote, U. M.: The application of positive matrix factorization with diagnostics to BIG DATA, Chemometrics and Intelligent Laboratory Systems, 240, 104 885, 2023.

Nanra, M., Saha, S., Shukla, A., Tripathi, S., and Kar, P.: Robust Shape-regularized Non-negative Matrix Factorization for Real-time Source Apportionment, pp. 192–201, https://doi.org/10.1145/3632410.3632457, 2024.

Paatero, P.: Least squares formulation of robust non-negative factor analysis, Chemometrics and Intelligent Laboratory Systems, 37, 23–35, https://doi.org/10.1016/S0169-7439(96)00044-5, 1997.

Paatero, P. and Hopke, P. K.: Rotational tools for factor analytic models, Journal of Chemometrics, 23, 91–100, https://doi.org/10.1002/cem.1197, 2009.

Paatero, P., Hopke, P. K., Song, X.-H., and Ramadan, Z.: Understanding and controlling rotations in factor analytic models, Chemometrics and Intelligent Laboratory Systems, 60, 253–264, https://doi.org/10.1016/S0169-7439(01)00200-3, 2002.

Tan, W., Chang, S., Fong, L., Li, C., Wang, Z., and Cao, L.: Matrix Factorization on GPUs with Memory Optimization and Approximate Computing, Proceedings of the 47th International Conference on Parallel Processing, 26, 1–10, https://doi.org/10.1145/3225058.3225096, 2018.

Zhang, Y., Fang, J., Meng, Q., Ge, X., Chebaicheb, H., Favez, O., and Petit, J.-E.: An Ensemble Machine Learning Approach for Predicting Sources of Organic Aerosols Measured by Aerosol Mass Spectrometry, ACS ES&T Air, https://doi.org/10.1021/acsestair.4c00262, 2025.